# Quantitative characterization of cell niches in spatially resolved omics data

Sebastian Birk [1,2,3,4], Irene Bonafonte-Pardàs[5,6], Adib Miraki Feriz[4], Adam Boxall [4], Eneritz Agirre [7], Fani Memi[4], Anna Maguza[3,8], Anamika Yadav[4], Erick Armingol [4], Rong Fan [9,10,11,12], Gonçalo Castelo-Branco [7,13], Fabian J. Theis [2,4,5,14], Omer Ali Bayraktar [4], Carlos Talavera-López [3,8] ✉ & Mohammad Lotfollahi [4,5] ✉

Spatial omics enable the characterization of colocalized cell communities that coordinate specific functions within tissues. These communities, or niches, are shaped by interactions between neighboring cells, yet existing computational methods rarely leverage such interactions for their identification and characterization. To address this gap, here we introduce NicheCompass, a graph deep-learning method that models cellular communication to learn interpretable cell embeddings that encode signaling events, enabling the identification of niches and their underlying processes. Unlike existing methods, NicheCompass quantitatively characterizes niches based on communication pathways and consistently outperforms alternatives. We show its versatility by mapping tissue architecture during mouse embryonic development and delineating tumor niches in human cancers, including a spatial reference mapping application. Finally, we extend its capabilities to spatial multi-omics, demonstrate cross-technology integration with datasets from different sequencing platforms and construct a whole mouse brain spatial atlas comprising 8.4 million cells, highlighting NicheCompass' scalability. Overall, NicheCompass provides a scalable framework for identifying and analyzing niches through signaling events.

Cell interactions are crucial for tissue formation, shaping small, diverse building blocks called niches—communities of spatially colocalized cells with coordinated functions[1,2]. Reflected in spatial gene expression patterns[3-5], these interactions provide a basis for identifying niches and analyzing their roles in health, development and disease, offering insights into tissue architecture and biomarkers to advance diagnostics, drug discovery and targeted therapies[6,7].

Recent developments in spatial genomics enable the comprehensive resolution of niches through imaging-based[8-11] and sequencing-based[12-16] spatial transcriptomics and multi-omics technologies[17], facilitating the construction of whole-organ spatial atlases spanning millions of cells[18,19]. Although these atlases provide

a foundation to study niches and cellular communication, computational approaches to identify and characterize niches based on their underlying cell interactions are lacking. Existing approaches identify niches by grouping cells based on histology or spatial gene expression[20-32] but often overlook key cellular processes, limiting biological insights. Signaling-based niche characterization can deepen our understanding of tissue hierarchies, spatially localized cellular processes and niche adaptation to homeostatic changes.

Here, we present NicheCompass (Niche Identification based on Cellular grapH Embeddings of COMmunication Programs Aligned across Spatial Samples), a graph deep-learning approach to identify and quantitatively characterize niches by learning cell

embeddings encoding signaling events as spatial gene program activities. NicheCompass explicitly models cellular communication by predicting the molecular profiles of cells and their neighbors in relation to specific signaling events, enabling pathway usage scoring in microenvironments and facilitating niche identification and characterization. Although existing methods address tasks[33] such as integration[20–28] and cell–cell communication inference[34,35], they differ from NicheCompass in at least two features in addition to its unique signaling-based approach: (1) they rely on single-cell data integration methods, leading to suboptimal niche recovery[22,28]; (2) they lack scalability[20,26]; (3) they cannot model spatial multi-omics[20,23–25,26,28]; or (4) they fail to map query data onto existing reference atlases[20,22–25,26,28].

We demonstrate the utility of NicheCompass across simulated and real data spanning varying species, conditions, technologies and modalities. In mouse organogenesis, NicheCompass reveals a hierarchy of highly resolved functional niches with niche-specific gene programs, consistent across embryos. Benchmarks show accurate niche recovery, gene program inference and batch effect removal. In human breast and lung cancer, NicheCompass decodes the tumor microenvironment, capturing donor-specific spatial organization and cellular processes, and enables spatial reference mapping, contextualizing query datasets with a reference to identify novel niches and contrast cellular processes. In a multimodal mouse brain dataset, it comprehensively characterizes niches based on multimodal programs. Finally, we demonstrate its scalability and cross-technology applicability by constructing spatial atlases across millions of cells.

## Results

### NicheCompass enables signaling-based niche characterization
NicheCompass processes cell-level or spot-level resolution spatial omics data by constructing a spatial neighborhood graph in which nodes represent cells or spots and edges indicate spatial proximity (Fig. 1a). Each node contains an omics feature vector (gene expression in unimodal data or paired gene expression and chromatin accessibility in multimodal data) and covariates (for example, sample) to account for confounders. A graph neural network encoder generates cell embeddings by jointly encoding features of nodes and their neighbors, capturing cellular microenvironments (Fig. 1b). A separate module removes batch effects through covariate embeddings[36]. To make embeddings interpretable, NicheCompass incorporates domain knowledge of intercellular and intracellular interaction pathways[37–42] to define spatial gene programs, with each embedding dimension incentivized to represent the activity of a specific program[43] (Fig. 1c). To overcome domain knowledge limitations (for example, quality issues, incompleteness or absence of niche-relevant features such as morphogen spatial gradients[44]), NicheCompass learns spatial de novo programs, capturing spatially co-expressed genes absent from prior knowledge (Fig. 1c).

To model intercellular interactions, programs are divided into self components and neighborhood components (Fig. 1d). The neighborhood component includes pathway genes associated with the source of intercellular interactions, modeling the microenvironment as a signaling source. The self component includes pathway genes related to the target of intercellular or intracellular interactions, modeling a cell or spot as a signaling receiver and responder. Prior programs are categorized into cell–cell communication, transcriptional regulation or combined interaction programs (Fig. 1d, Supplementary Fig. 1 and Supplementary Note 1). In multimodal scenarios, peaks are linked to genes if they lie within the gene body or promoter region[45]. NicheCompass provides default programs for each category through database application programming interfaces (APIs)[37–40] while allowing customization.

Embeddings are decoded to jointly reconstruct spatial and molecular information (Fig. 1e). A graph decoder computes sample-specific embedding similarities to reconstruct the neighborhood graph using an edge reconstruction loss, encouraging similar embeddings for neighboring nodes. Two masked linear omics decoders reconstruct features specific to each program, disentangling variation and enabling interpretability[43,46]: one reconstructs neighborhood omics features, obtained by aggregation across neighbors; the other reconstructs the node's own omics features. For instance, a ligand-encoding gene is reconstructed in the neighborhood, while its corresponding receptor-encoding and target genes are reconstructed in the node. Redundancy in programs is addressed by prioritizing informative ones with a pruning mechanism while applying selective regularization to promote gene sparsity within programs (Methods).

The complete architecture of NicheCompass is a multimodal conditional variational graph autoencoder[47,48]. This design enables a quantitative signaling-based niche characterization and provides an end-to-end framework for spatial omics analysis (Fig. 1f and Supplementary Note 2).

### NicheCompass elucidates tissue architecture across embryos
We applied NicheCompass to a sequential fluorescence in situ hybridization (seqFISH) mouse organogenesis dataset[49] comprising three spatially disparate embryo tissues (Supplementary Fig. 2a). After integration and clustering of embeddings, we annotated clusters with niche labels based on two characterizing programs (Methods), anatomical locations and cell type compositions (Fig. 2a and Supplementary Fig. 2b). Niches were spatially contiguous and exhibited distinct cell type composition patterns (Fig. 2a,b), including homogeneous populations characteristic of organogenesis[49] and heterogeneous populations (Supplementary Fig. 3), highlighting the value of spatial information. NicheCompass revealed clearly segregated central nervous system (CNS) niches, previously labeled collectively, and identified an additional floor plate niche enriched in the Shh combined interaction program, consistent with Shh secretion and marker expression[50] (Fig. 2a and Supplementary Fig. 4a). Integration across embryos was successful (Fig. 2c), with most niches present in all embryos and absences explained by sample-specific tissue architecture (Supplementary Fig. 5).

To assess global spatial organization, we applied hierarchical clustering, grouping niches into higher-order functional components (Fig. 2d). CNS niches (midbrain, forebrain, floor plate, hindbrain, spinal cord) formed one cluster, while dorsal and ventral gut niches constituted another, consistent with anatomy. Characterizing program activities supported this hierarchy and distinguished individual niches (Fig. 2e). Niches within the same cluster exhibited similar cell type composition, reflecting meaningful molecular integration (Fig. 2f).

We analyzed program activities in gut and brain niches to investigate interactions driving niche identity. Each niche showed enriched activity of specific programs (Fig. 2g,h, Extended Data Fig. 1 and Supplementary Note 3). In the ventral gut niche, the Spint1 combined interaction program showed the highest activity (Fig. 2g). Based on gene importances (Methods), this program was driven by *Spint1* and *St14*, encoding the ligand HAI-1 and receptor matriptase, respectively, whose interaction regulates intestinal epithelial barrier integrity[51,52]. In the dorsal gut niche, the Cthrc1 combined interaction program was upregulated (Fig. 2g), driven by the ligand-encoding and receptor-encoding genes *Cthrc1* and *Fzd3* and localized to the notochord[53], validated by *Nog* marker expression[54] (Supplementary Fig. 4b). Cthrc1–Fzd3 binding is implicated in the Wnt planar cell polarity pathway during mouse embryo development[53]. In the hindbrain niche, the Fgf3 combined interaction program was upregulated (Fig. 2h), driven by the ligand-encoding and receptor-encoding genes *Fgf3* and *Fgfr1* (ref. 55). Fgf3 signaling is essential for neuronal development and establishment of hindbrain compartment boundaries[56,57]. The floor plate niche was demarcated by the Calca combined interaction program (Fig. 2h), driven by *Calca*, which is important in glutamatergic neurons at the midbrain–hindbrain junction[58]. In the midbrain niche, we identified enriched activity of the Fgf17 combined interaction program (Fig. 2h), driven by the ligand-encoding and receptor-encoding genes *Fgf17* and

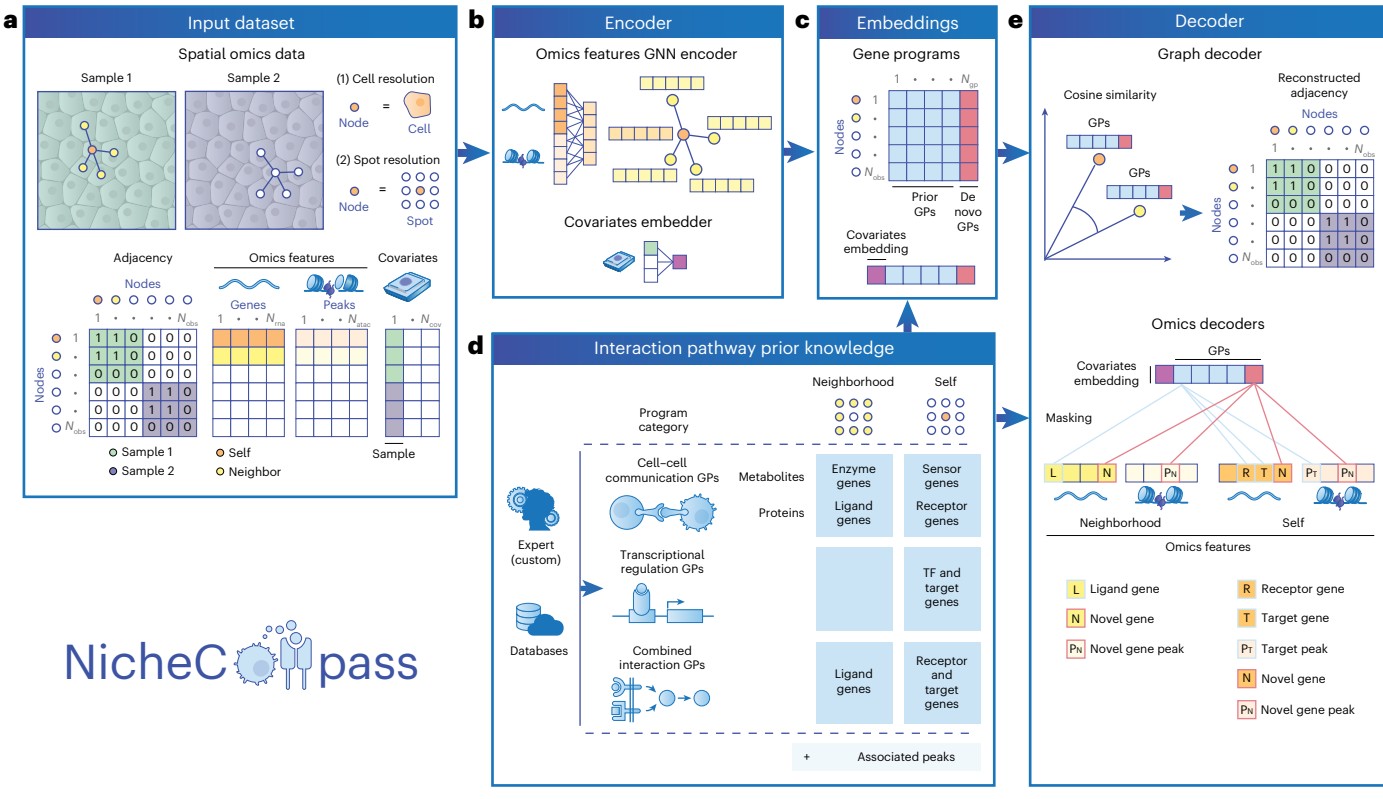

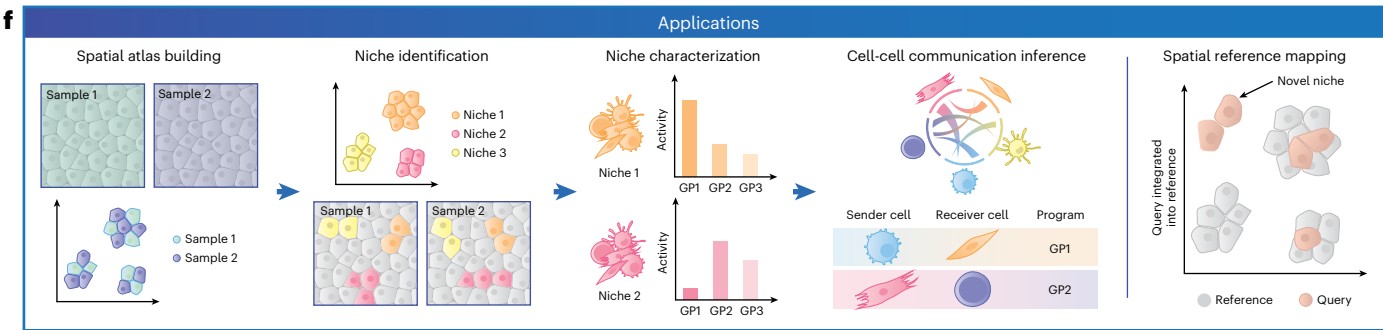

**Fig. 1 | Overview of NicheCompass. a**, NicheCompass takes single-sample or multi-sample spatial omics data with cell-level or spot-level observations as input. Using the 2D coordinates, it constructs a spatial neighborhood graph (represented with a binary adjacency matrix), with each cell or spot representing a node. Each observation includes omics features (gene expression and optionally paired chromatin accessibility) and covariates to account for confounders (for example, sample). **b**, A graph neural network (GNN) encoder generates cell embeddings, with covariates embedded for removal of confounding effects. **c**, The model is incentivized to learn an embedding in which each feature represents the activity of a spatially localized interaction pathway retrieved from domain knowledge, represented as a prior program. In addition to prior programs, the model can discover de novo programs, which learn a set of spatially co-occurring genes and peaks. GPs, gene programs. **d**, GPs, derived from databases or experts, are classified into three categories and comprise neighborhood components and self components to reflect intercellular and intracellular interactions. The neighborhood component contains genes linked to the interaction source of intercellular interactions, and the self component contains genes linked to the interaction target of intercellular interactions and genes linked to intracellular interactions. Peaks are associated with genes if locationally proximal. TF, transcription factor. **e**, Decoders reconstruct spatial and molecular information while constraining embedding features to represent the activity of a specific program: a graph decoder reconstructs sample-specific input adjacencies, and omics decoders reconstruct a node's omics counts and aggregated counts of its neighborhood. Omics decoders are linear and masked based on programs, thus enabling interpretability (exemplified by a combined interaction program). **f**, NicheCompass facilitates critical downstream applications in spatial omics data analysis. Illustrations of cells were created with BioRender.com.

*Fgfr2*. This pathway is crucial for vertebrate midbrain patterning[59,60]. Lastly, in the forebrain niche, the Dkk1 ligand–receptor program showed distinctive activity (Fig. 2h), with Dkk1 promoting forebrain neuron precursor formation[61,62].

To validate the integrity of the learned program activities, we compared the expression of ligand-encoding and receptor-encoding genes with their reconstructed expression, finding strong congruence (Extended Data Fig. 1). To assess reproducibility and robustness of the identified niches and inferred programs, we trained additional models with different seeds and neighborhood graphs, observing high alignment (Extended Data Fig. 2). We further evaluated the generalizability in leave-one-out scenarios by training models excluding embryo 2 and embryo 3, respectively. Mapping embryo 2 as a query revealed strong correspondence between identified niches and inferred program activities (Extended Data Fig. 2d). Finally, to test robustness against prior program selection, we trained models on limited program sets.

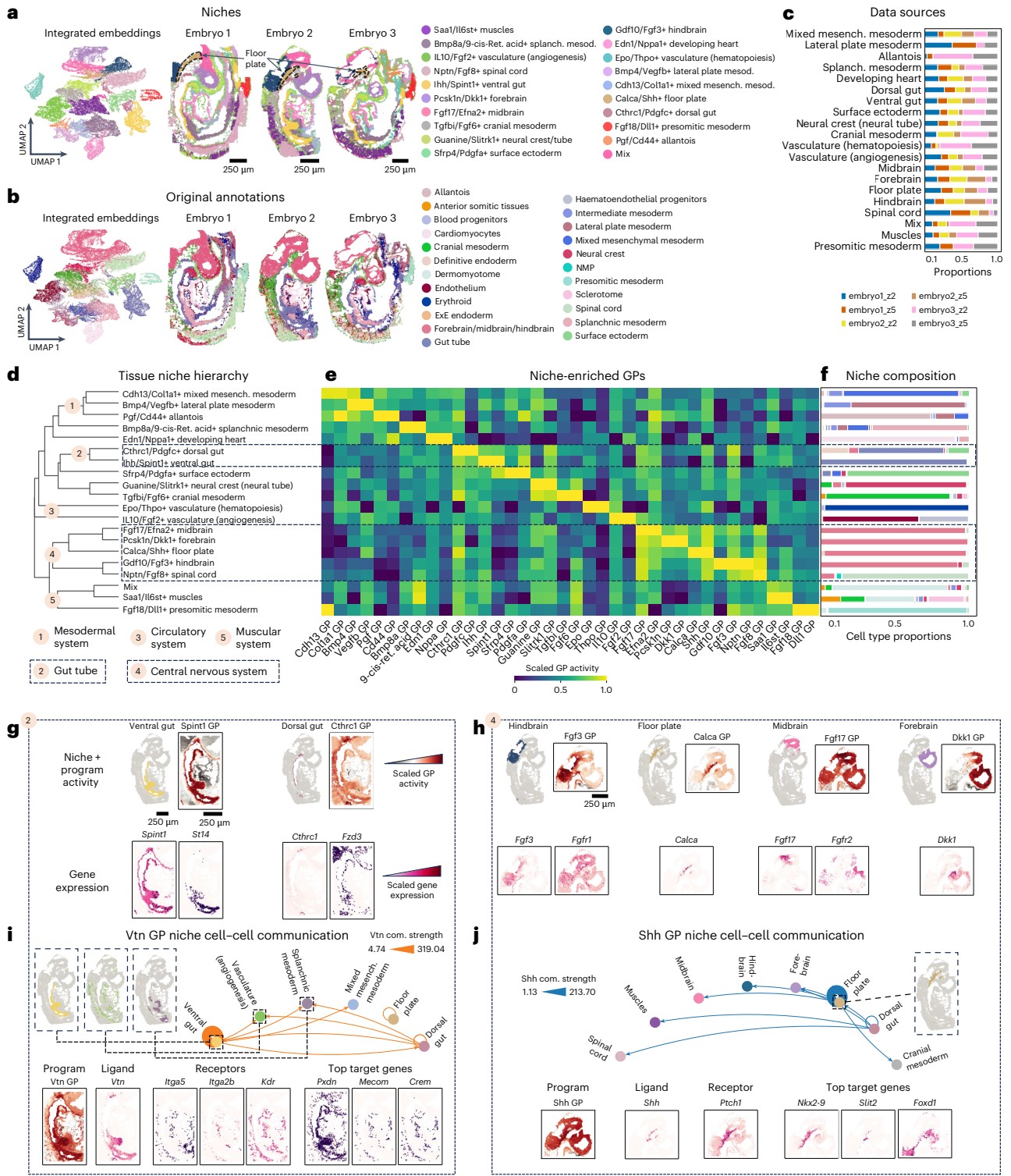

**Fig. 2 | NicheCompass reveals cellular interactions shaping tissue organization in mouse development. a**, Uniform manifold approximation and projection (UMAP) of integrated NicheCompass embeddings and the three embryo tissues[49], colored by niches annotated using characterizing programs (gene names in niche annotations refer to characterizing programs that are upregulated in the niche compared to all other niches). The floor plate niche is outlined and labeled. **b**, Same UMAP as **a** but colored by original cell type or region annotations. ExE endoderm, extraembryonic endoderm; NMP, neuromesodermal progenitor. **c**, Cell proportions from each section across niches. **d**, Dendrogram of average program activities showing a functional higher-order hierarchy. **e**, Heatmap of normalized activities for two characterizing programs per niche, showing gradients along the hierarchy. **f**, Cell type proportions for each niche (colors from **b**). **g,h**, Activities of characterizing programs differentiating ventral and dorsal gut niches (**g**) and CNS niches (**h**), with correlated expression of ligand-encoding and receptor-encoding genes. **i,j**, Cell–cell communication analysis for a ventral gut program (**i**) and a floor plate program (**j**), showing inferred communication strengths between niches and consistent member gene expression. Nodes represent niches and edges the strength (width) and direction (arrowheads) of the interaction. Com. strength, communication strength.

Niches remained robust, but distinct biology was unraveled across program sets (Supplementary Fig. 6).

Using the inferred program activities, we analyzed interactions by computing source-specific and target-specific communication potential scores for each cell, allowing us to quantify communication strengths between cell pairs and aggregate them at niche and cell type levels (Methods and Supplementary Note 4). We applied this strategy to the Vtn combined interaction program, enriched in the ventral gut niche (Fig. 2i and Supplementary Fig. 7a,b). This program included known interactions of Vtn with the Kdr receptor and integrin receptors encoded by *Itga5* and *Itga2b*, key regulators of cellular responses during gut development[63]. In addition to these, important target genes (*Pxdn*, *Mecom*, *Crem*) showed spatially correlated expression (Fig. 2i). Communication strength analysis revealed that this program mediated both intra-niche interactions in the ventral gut and inter-niche interactions with the vasculature (angiogenesis) and splanchnic mesoderm niches, aligning with vitronectin–integrin signaling being a key contributor to mouse angiogenesis[64]. We similarly interrogated the Shh combined interaction program, enriched in the floor plate niche (Fig. 2j and Supplementary Fig. 7c,d). Alongside the ligand-encoding and receptor-encoding genes *Shh* and *Ptch1*, NicheCompass identified downstream targets of Shh signaling, including *Nkx2-9* (implicated in dopaminergic neuron specification[65–67]), *Slit2* (supporting ventral nerve cord axon migration[68]) and *Foxd1* (known Shh target in retina patterning[69]). Although Shh program activity was primarily observed in the floor plate niche, it extended to other brain niches, consistent with broader Shh brain signaling[70].

These results demonstrate how, based on program activity, NicheCompass can infer a hierarchy of fine-grained niches and their underlying interaction mechanisms across tissues.

## NicheCompass accurately identifies niches in diverse data

We benchmarked NicheCompass against other methods[20,22,26,28,35] using simulated and real data from various technologies, species and tissues. On a SlideSeqV2 mouse hippocampus dataset[12], NicheCompass-identified niches corresponded closely with anatomical subcomponents in the Allen Brain Atlas[71] (Fig. 3a). Hierarchical clustering showed isocortex and hippocampus clusters aligned with known taxonomy, while deviations in the thalamus cluster were explained by similarities in niche composition (Fig. 3b and Supplementary Fig. 8a). Compared to BANKSY[28], GraphST[20] and CellCharter[22], NicheCompass uniquely identified spatially contiguous niches and outperformed all methods in spatial consistency and niche coherence metrics (Fig. 3c,d and Supplementary Notes 5 and 6). Owing to STACI's[26] inability to train on a 40 GB GPU, additional benchmarking was conducted on a 25% subsample, with NicheCompass maintaining superior performance (Supplementary Fig. 9 and Supplementary Note 5).

We validated NicheCompass on simulated data generated with SRTsim[72], which included ground-truth niche labels, including niche-specific signaling events (Extended Data Fig. 3a–c and Methods). Among all methods tested, only NicheCompass and BANKSY accurately recovered ground-truth niches. Additionally, NicheCompass outperformed alternative workflows in retrieving ground-truth programs (Extended Data Fig. 3d–f and Supplementary Note 7). We also conducted ablation studies to evaluate design choices and inform hyperparameter selection (Methods, Supplementary Figs. 10–13 and Supplementary Note 8). Further analysis on a binned version of the dataset demonstrated NicheCompass' robustness across resolutions (Supplementary Fig. 14 and Supplementary Note 9).

We then evaluated integration capability on a NanoString CosMx human non-small cell lung cancer (NSCLC) dataset[10]. As GraphST and STACI could not run on the full dataset, we used a 10% subsample with strong batch effects (Extended Data Fig. 4a). Only NicheCompass could integrate all replicates successfully (Fig. 3e, Extended Data Fig. 4b,c and Supplementary Note 10). It identified distinct niches,

including a lymphoid structures niche and a tumor-stroma-boundary niche, and it distinguished between endothelial-enriched and plasmablast-enriched stroma, each with clear compositional signatures. By contrast, CellCharter failed to separate niches, STACI missed the tumor-stroma-boundary niche, BANKSY struggled with integration and GraphST grouped unrelated niches. Quantitative evaluation confirmed NicheCompass' superior batch correction and competitive spatial consistency and niche coherence (Extended Data Fig. 4d).

Finally, we assessed scalability and applicability across datasets of varying sizes and gene panels. Among tested methods, only NicheCompass, BANKSY and CellCharter could process larger datasets (>70,000 cells). NicheCompass largely outperformed others, demonstrating robustness to subsampling and effectiveness in diverse multi-sample scenarios (Fig. 3f,g and Supplementary Figs. 15–23).

Across benchmarks, NicheCompass exhibited exceptional scalability and efficiency through its memory-efficient design (Supplementary Fig. 24 and Supplementary Note 11).

## NicheCompass discerns cancer niches through de novo programs

We applied NicheCompass to a Xenium human breast cancer dataset[73] with a limited gene panel of 313 probes (only 23% of genes were present in our prior knowledge programs). It integrated multiple tissue replicates (Fig. 4a–d) containing 11 cell types and 27 cell states (Fig. 4b and Supplementary Fig. 25a). Clustering the embeddings revealed 14 niches with specific anatomical localizations, highlighting tissue architecture (Fig. 4a,e). Owing to probe limitations, niches were annotated by their most abundant cell types (Supplementary Fig. 25b) and showed enrichment in immune, epithelial and epithelial-to-mesenchymal transition (EMT) states, with Epi-FB, CD4+T and EMT-immune niches comprising the largest proportions (26.9%, 24.9% and 18.6% of cells).

Despite limited probes, NicheCompass identified niche-specific programs critical for understanding tumor microenvironments. For instance, the Ptprc combined interaction program, enriched in the CD4+T niche (Fig. 4e), is associated with cancer prognosis[74]. Additionally, de novo programs revealed highly correlated genes (Fig. 4f,g and Supplementary Fig. 26), including two with increased activity in immune and EMT-associated niches (Supplementary Fig. 25c,d), highlighting their potential as pathology biomarkers and drug targets.

NicheCompass identified a de novo program (37 GP; Fig. 4f,h and Supplementary Fig. 26c) comprising basal markers *KRT16*, *KRT14*, *KRT5*, *KRT6B* and *KRT15*, all implicated in oncological studies. *KRT16*, linked to metastasis, promotes EMT and motility[75], while *KRT6B* and *KRT15* are associated with basal-like breast cancer and tumor metastasis, respectively[76]. Another program (86 GP; Fig. 4g,i and Supplementary Fig. 26c) included *MLPH*, *EPCAM*, *FOXA1*, *ELF3* and *KRT8*, genes central to breast cancer pathology. ELF3 activates KRT8, driving epithelial differentiation and tumorigenesis, and interacts with FOXA1 in endocrine-resistant ER+ breast cancer. These findings showcase NicheCompass' ability to uncover de novo programs and their connections to cellular processes and prior knowledge (Fig. 4h,i).

NicheCompass delineated niches anatomically, identifying de novo programs linked to histological structures (Fig. 4f,g). For instance, de novo 37 program highlighted a transcriptional signature of KRT14+ proliferative epithelial tumor cells cohabiting with myeloid cells[77], while de novo 86 program identified an epithelial-vascular niche driven by *EPCAM* and *KRT8*, associated with preneoplastic and luminal tumor progression. These biomarkers, linked to basal (*KRT14*) and luminal (*KRT8*) breast cancer cells[78], showed high activity in EMT-Mφ and EMT-Endo niches (Supplementary Fig. 25c,d).

In summary, NicheCompass identified cancer-related programs and niches, proving effective even with limited gene panels.

## NicheCompass constructs a spatial lung cancer atlas

To evaluate its ability to identify donor-specific tumor microenvironment features and interactions as well as its spatial reference mapping

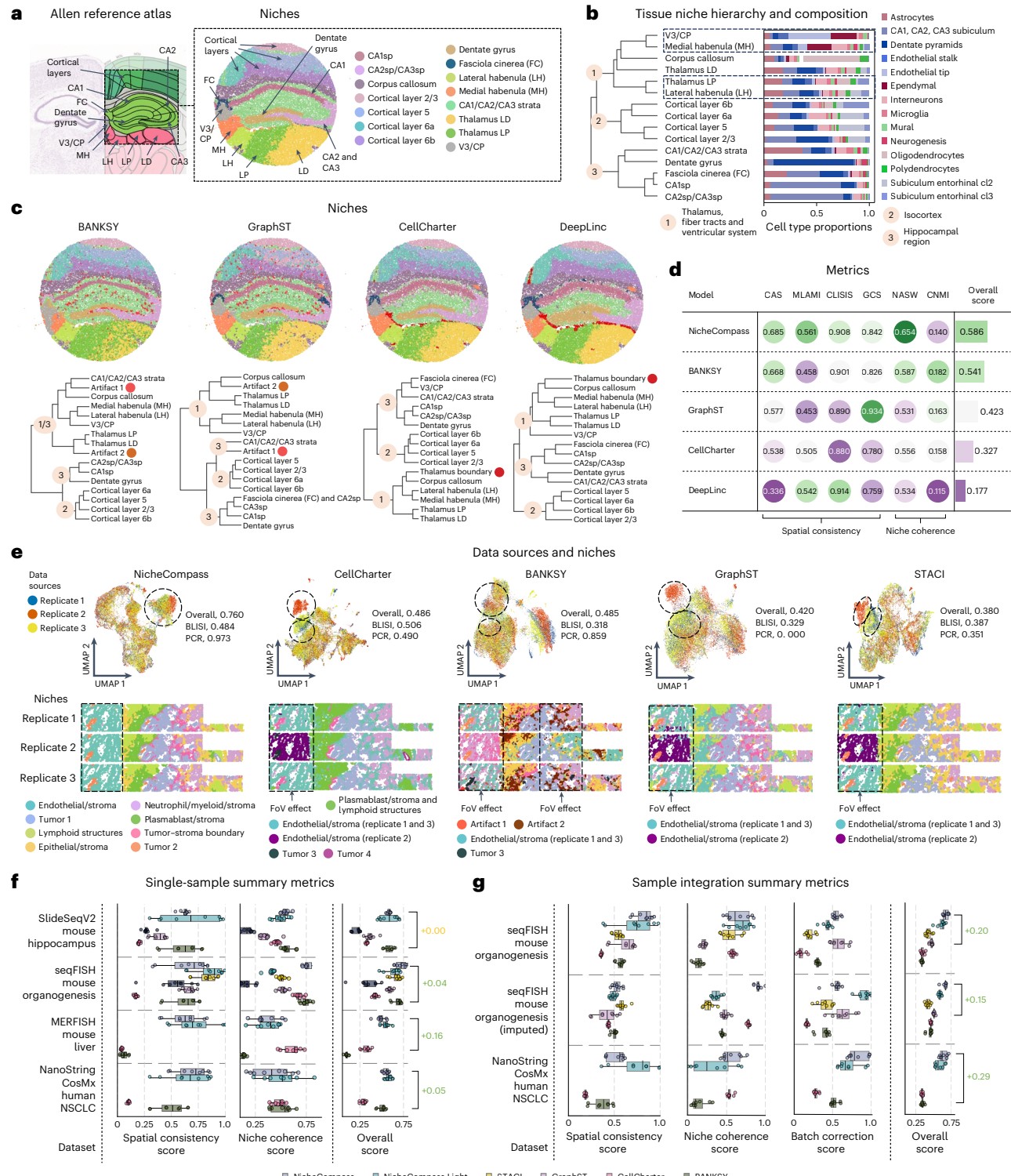

**Fig. 3 | Benchmarking NicheCompass across diverse scenarios. a**, Coronal mouse brain image from the Allen Brain Atlas (left) and a SlideSeqV2 hippocampus tissue (right)[12], showing corresponding niches identified by NicheCompass. CA1sp, CA1 pyramidal layer; CA2sp, CA2 pyramidal layer; CA3sp, CA3 pyramidal layer. **b**, Dendrogram of average program activities reveals a hierarchy of anatomically and molecularly similar niches, and their cell type compositions. **c**, Top: mouse hippocampus tissue colored by niches identified using four methods. Cluster colors match with **a**. Bottom: the corresponding dendrograms computed on each method's embeddings. **d**, Performance comparison across six metrics for spatial consistency and niche coherence, aggregated into an overall score (Methods). **e**, Integration performance of NicheCompass, CellCharter[22], BANKSY[28], GraphST[20] and STACI[26] on a NanoString CosMx NSCLC dataset subsample[10]. Top: UMAPs colored by data source highlight endothelial and stroma niches integrated only by

NicheCompass. Bottom: lung tissue replicates display differences in batch effect removal and niche resolution. Highlighted is the first field of view (FoV) across all three replicates where other methods show FoV effects hindering integration. Niche annotations below tissue sections refer to niches identified by the respective method. For methods other than NicheCompass, only differences compared to NicheCompass are displayed. **f**,**g**, Performance summary metrics of NicheCompass and similar methods on four single-sample (**f**) and three multi-sample (**g**) datasets. Metrics were computed for $n = 8$ training runs per dataset and method while varying sizes of the $k$-nearest neighbors graph (two runs per $k$ with $k = 4, 8, 12, 16$). Missing boxes indicate training failures resulting from memory constraints. Numbers on the right indicate mean score differences between NicheCompass and the second-best performing method on each dataset (green, NicheCompass performs better; yellow, NicheCompass is on par).

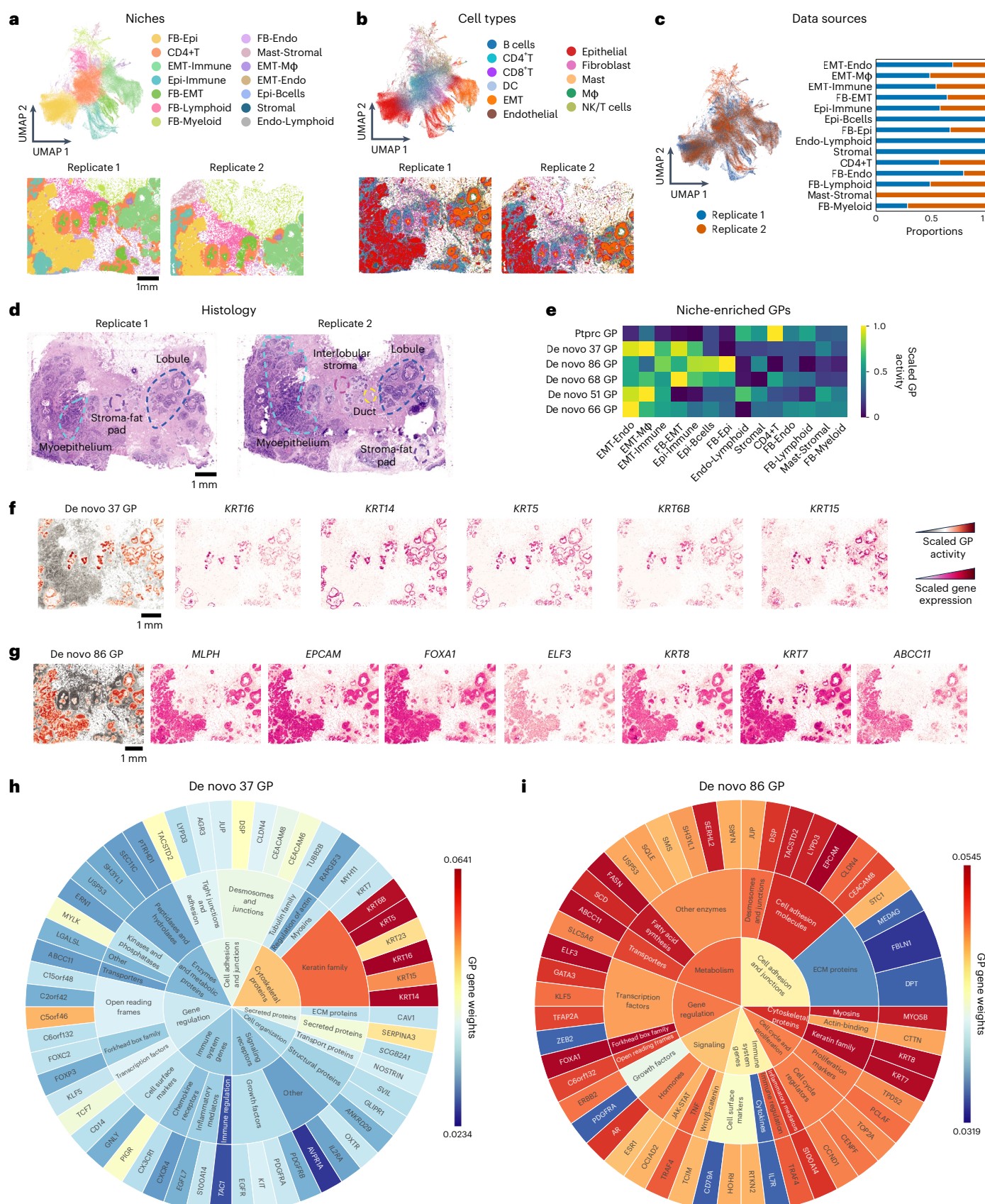

**a** Niches

FB-Epi, CD4+T, EMT-Immune, Epi-Immune, FB-EMT, FB-Lymphoid, FB-Myeloid, FB-Endo, Mast-Stromal, EMT-Mφ, EMT-Endo, Epi-Bcells, Stromal, Endo-Lymphoid

Replicate 1 / Replicate 2 / 1mm

**b** Cell types

B cells, CD4+T, CD8+T, DC, EMT, Endothelial, Epithelial, Fibroblast, Mast, Mφ, NK/T cells

Replicate 1 / Replicate 2

**c** Data sources

Replicate 1 / Replicate 2

EMT-Endo, EMT-Mφ, EMT-Immune, FB-EMT, Epi-Immune, Epi-Bcells, FB-Epi, Endo-Lymphoid, Stromal, CD4+T, FB-Endo, FB-Lymphoid, Mast-Stromal, FB-Myeloid — Proportions

**d** Histology

Replicate 1: Lobule, Stroma-fat pad, Myoepithelium. Replicate 2: Interlobular stroma, Lobule, Duct, Myoepithelium, Stroma-fat pad. 1 mm

**e** Niche-enriched GPs

Ptprc GP, De novo 37 GP, De novo 86 GP, De novo 68 GP, De novo 51 GP, De novo 66 GP — Scaled GP activity

**f** De novo 37 GP — KRT16, KRT14, KRT5, KRT6B, KRT15. Scaled GP activity / Scaled gene expression. 1 mm

**g** De novo 86 GP — MLPH, EPCAM, FOXA1, ELF3, KRT8, KRT7, ABCC11. 1 mm

**h** De novo 37 GP — GP gene weights 0.0234–0.0641

**i** De novo 86 GP — GP gene weights 0.0319–0.0545

capabilities, we applied NicheCompass to the full NSCLC dataset[10], which includes eight tissue sections from five donors.

We trained NicheCompass to build a reference atlas using four donors and two replicates. Clustering the embeddings revealed 12 niches with differential cell composition, spatial organization and gene expression (Fig. 5a,b and Extended Data Figs. 5c,e,f and 6a). Owing to their spatial segregation (Extended Data Fig. 5g and Supplementary Fig. 27), most cancer cells (92%) formed tumor-exclusive niches (>75%

**Fig. 4 | NicheCompass identifies meaningful niches and de novo programs in human breast cancer. a**, Top: UMAP of the NicheCompass embedding space after integrating two replicates of a 313-probe Xenium dataset[10]. Bottom: tissue replicates colored by identified niches. Niches include FB-Epi (fibroblast-epithelial), CD4+T (CD4[+]T cells), EMT-Immune, Epi-Immune (epithelial-immune), FB-EMT (fibroblast-EMT), FB-Lymphoid (fibroblast-lymphoid), FB-Myeloid (fibroblast-myeloid), FB-Endo (fibroblast-endothelial), Mast-Stromal (mast cells-stromal), EMT-Mφ (EMT-macrophage), EMT-Endo (EMT-endothelial), Epi-Bcells (epithelial-B cells), Stromal and Endo-Lymphoid (endothelial-lymphoid). **b**, Same UMAP as **a**, colored by cell types. DC, dendritic cell; Mφ, macrophage; NK, natural killer. **c**, UMAP colored by

data source, showing successful integration and proportion of cells from each data source across niches. **d**, Annotated H&E slides of the breast cancer tumor resection. **e**, Heatmap of normalized activities for characterizing programs associated with cancer progression and pathological histology. **f,g**, Program activity and expression of key genes for de novo 37 (**f**) and 86 (**g**) programs, showing correlations between activity and gene expression. **h,i**, Sunburst plots of gene weights for de novo 37 (**h**) and 86 (**i**) programs. De novo 37 program highlights keratin genes and an uncharacterized gene (*C5orf46*). De novo 86 program reveals a *KRT8*-driven program with links to fatty acid metabolism (*FASN*, *ABCC1*) and *ELF3* as a potential regulator. The scale represents inferred gene weights.

---

tumor cells) while only highly infiltrative stromal niches like niche 6 (tumor-infiltrating neutrophils) contained tumor cells (Extended Data Fig. 5c). Tumor niches were donor-specific but shared across technical replicates, confirming that the results were not driven by technical effects (Fig. 5c and Extended Data Fig. 5d). Stroma niches, while donor-dependent, showed shared structures when similar patterns existed (Fig. 5c and Extended Data Fig. 5d), aligning with findings that NSCLC patients can be stratified by tumor microenvironment infiltration patterns[79]. At the global level, hierarchical clustering separated tumor and stromal sub-niches robustly, despite inter-sample heterogeneity (Extended Data Fig. 5a).

In donor 9, tumor cells were divided into two niches: niche 1 (tumor-stroma border) and niche 3 (neutrophil-infiltrated tumor cells), labeled based on histological images and neighborhood composition (Fig. 5d,k). Niche 3 showed enrichment of the CXCL1 ligand–receptor program, consistent with CXCL1's role as a neutrophil chemoattractant[80] (Fig. 5d and Supplementary Fig. 28a). This highlights the ability of NicheCompass to distinguish niches with different interacting cells despite similar spatial organization. Notably, 11% of donor 12 tumor cells, which were surrounded by neutrophils (Supplementary Fig. 28b,c), also clustered into niche 3, demonstrating the identification of conserved niches across patients.

Stroma clusters were distinguished by dominant immune cell types and spatial arrangements, such as tumor-infiltrating or immune expansions (Fig. 5b and Extended Data Figs. 5c,e and 6). For example, two neutrophil-dominated niches with similar composition mapped closely but differed structurally: niche 7 (donor 5) formed a large expansion outside the tumor, while niche 6 (donors 9 and 12) consisted of smaller tumor-infiltrating expansions (Fig. 5e). This demonstrates the ability of NicheCompass to identify infiltrating immune cells across samples. Shared structures, such as lymphoid aggregates (niche 11) surrounded by plasmablast-rich stroma (niche 9) in donors 5 and 12, were correctly identified when composition and spatial arrangement were consistent (Fig. 5e and Extended Data Fig. 6b).

In summary, we constructed a spatial NSCLC reference atlas, demonstrating the ability of NicheCompass to integrate heterogeneous samples, identify shared and donor-specific niches and uncover underlying programs.

## NicheCompass discovers niches by spatial reference mapping

We evaluated spatial reference mapping to integrate matching niches while preserving donor-specific variation by mapping a held-out biological replicate (Supplementary Fig. 29a,b) and a new donor sample (Fig. 5f) onto the integrated reference.

Simulating limited dataset access, we first trained a k-nearest neighbors (k-NN) classifier on the reference to transfer niche labels to query cells (Fig. 5h and Supplementary Fig. 29c). Query cells from the biological replicate (donor 5) were correctly integrated into the reference with high assignment probability, preserving biological features while removing batch effects (batch ASW 0.97; Supplementary Figs. 29 and 30a). When mapping the new donor, label transfer distinguished tumor niches from macrophage-rich and lymphoid-rich niches (Fig. 5g,h), with some low-probability assignments suggesting novel query niches (Supplementary Fig. 30a). Jointly re-clustering embeddings revealed two shared lymphoid-rich niches (niches 10 and 14) and two novel niches with tumor cells (niche 15) and macrophages (niche 13; Fig. 5g,i).

The cellular composition and spatial distribution of shared niches (Fig. 5j) revealed between-donor similarities in tumor-infiltrating stromal niches dominated by stromal (niche 14) or lymphoid cells (niche 10; Supplementary Fig. 30b). By contrast, no query cells mapped to the non-infiltrating stromal niche 8 of donor 9, as all query cells were tumor-infiltrating (Fig. 5i,j).

Macrophage niche 13, consisting of tumor-infiltrating macrophages, mapped closely to but differed from the reference macrophage-rich niche 12, which was adjacent to tumors and primarily from donor 6 (squamous cell carcinoma; Fig. 5i,j), reflecting tissue organization differences[81]. Tumor niche 15, close in embedding space to macrophage niche 13 (Fig. 5i), was the only tumor niche with significant macrophage interaction based on neighborhood composition analysis (Fig. 5k).

Differential analysis revealed upregulation of the SPP1 ligand–receptor and combined interaction programs in niche 15 tumor cells and niche 13 macrophages (Fig. 5l). SPP1 characterizes a well-established subtype of profibrotic macrophages[82–85], drives macrophage polarity in the tumor microenvironment[86] and is a marker of pro-tumor-infiltrating macrophages associated with poor lung cancer

---

**Fig. 5 | NicheCompass spatial reference mapping contextualizes new donors and reveals emergent niches. a–c**, UMAP of NicheCompass embeddings for six NSCLC lung samples[10], colored by identified niches (**a**), pre-annotated cell types (**b**) and donor or donor replicate (**c**). **d**, Spatial visualization of tissue sections from donors 9 and 12, showing niches, cell types and CXCL1 ligand–receptor (LR) program activity, distinguishing tumor niches interacting with stromal tissue (niche 1) or neutrophils (niche 3). **e**, Spatial visualization of tissue sections colored by niche and cell type, highlighting shared and donor-specific stromal structures across donors. **f**, UMAP of NicheCompass spatial reference with query cells mapped by fine-tuning. **g,h**, UMAPs of mapped query cells colored by pre-annotated cell types (**g**) and niche labels as predicted by a k-NN classifier trained on the reference, including prediction probabilities (**h**). **i**, Joint UMAP of reference and query embeddings, colored by niches as identified by re-clustering. In addition, bar plots represent the donor distribution of the niches

the query sample maps to. **j**, Spatial visualization of query tissue (donor 13) and its most similar reference samples, colored by cell type (key at bottom) and niche (colored as in **i**), comparing newly identified niches to reference counterparts. **k**, Neighborhood composition in tumor niches (niche 1, 89,814 cells; niche 2, 60,131 cells; niche 3, 39,500 cells; niche 4, 41,864 cells; niche 5, 14,516 cells; niche 15, 25,271 cells). A boxplot per tumor niche and neighboring cell type represents the niche-specific distribution of cells of a given cell type among the 25 physically closest cells. Only cell types composing on average more than 5% and less than 60% of the neighborhood are shown. The query tumor niche is highlighted. **l**, Joint UMAP of reference and query embeddings, colored by SPP1 LR and combined interaction program activity, and expression of the ligand-encoding and receptor-encoding genes. **m**, Heatmap of SPP1 LR communication strengths between niches in the query (donor 13) and reference (donor 6) samples, the two donors with highest macrophage infiltration.

prognosis[87,88]. Closer gene expression analysis confirmed *SPP1* and related markers (*IFI27*, *CD9*)[83] over-expression in niche 13 relative to other macrophage niches, and a profibrotic phenotype with elevated extracellular matrix protein gene expression (*FN1*, *COL3A1*, *COL1A1*, *MMP2*, *MMP12*, *TIMP1*; Supplementary Fig. 31)[84]. Tumor niche 15 also

overexpressed *SPP1* and its receptor-encoding genes (*ITGAV*, *ITGB1*, *EGFR*). Cell–cell communication analysis revealed stronger SPP1-driven signaling in the query macrophage niche compared to the reference, with higher communication strengths both within the macrophage niche and to other niches (Fig. 5m).

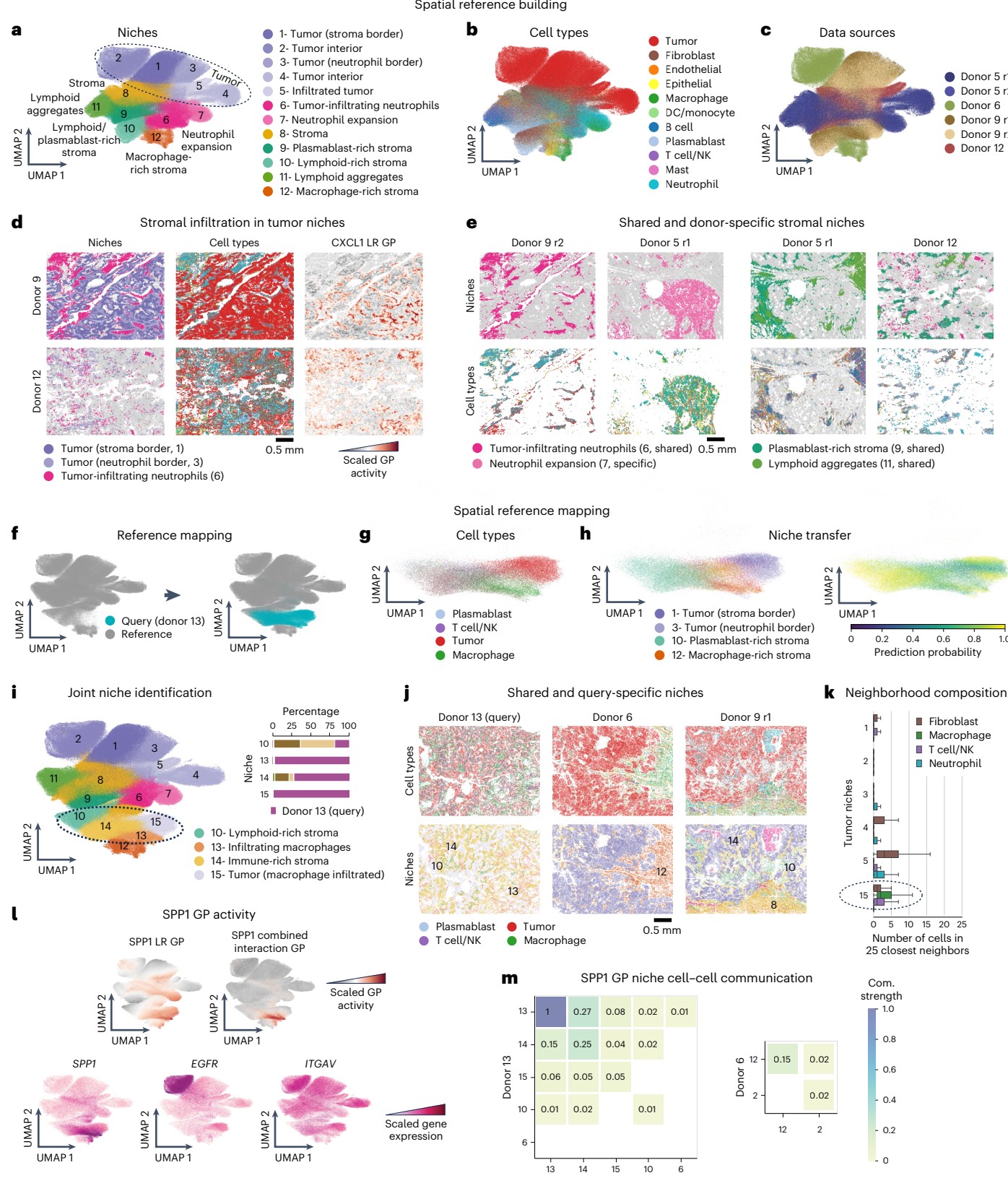

**Fig. 5** (Spatial reference building, mapping and analysis figure, panels a–m)

Our analysis demonstrates the ability of NicheCompass to detect novel niches and niche-specific interactions including in spatial reference mapping scenarios.

## NicheCompass enables multimodal niche characterization

Incorporating spatially resolved epigenetic factors like chromatin accessibility can aid in understanding tissue architecture[17]. Leveraging multimodal programs, we trained NicheCompass on a spatial multi-omics mouse brain dataset generated with the spatial assay for transposase-accessible chromatin and RNA using sequencing (spatial ATAC–RNA-seq) technology[17]. Despite sparse marker detection (Supplementary Fig. 32a), the identified niches corresponded well with the Allen Brain Atlas[71] (Supplementary Fig. 32b). Using our analysis workflow, we investigated the major island of Calleja and corpus callosum niches, revealing interesting transcriptional regulation programs with multimodal footprints (Supplementary Figs. 32c–f, 33 and 34 and Supplementary Note 12).

These findings highlight how chromatin accessibility can help to elucidate transcriptional regulatory mechanisms shaping niche identity.

## NicheCompass aligns millions of cells across technologies

To demonstrate scalability and cross-technology applicability, we constructed whole-organ spatial atlases. First, we applied NicheCompass to the STARmap PLUS mouse CNS dataset (~one million cells)[19], identifying 15 niches aligned across sequential sections and corresponding to anatomical regions in the Allen Brain Atlas[71] (Extended Data Fig. 7). We then integrated 8.4 million cells from 239 sections of a MERFISH whole mouse brain dataset[89], aligning matching brain regions into spatially consistent niches across donors (Extended Data Fig. 8). Finally, cross-technology integration of both datasets revealed anatomically consistent shared niches (Extended Data Fig. 9).

These results highlight the ability of NicheCompass to assemble spatial atlases across individuals and technologies[90].

## Discussion

We introduced NicheCompass, a graph deep-learning approach that identifies and quantitatively characterizes tissue niches using cellular communication principles. Benchmarking highlighted its superior niche identification and gene program inference (Fig. 3 and Extended Data Fig. 3). Its scalable design supports datasets with millions of cells and enables cross-technology integration for spatial atlas projects[91] and digital pathology analyses (Extended Data Figs. 7–9). NicheCompass also facilitates iterative integration through spatial reference mapping (Fig. 5f–i) and multimodal niche characterization (Supplementary Fig. 32). Applications to mouse organogenesis, the adult mouse brain and human cancers revealed tissue architecture and niche-specific programs, positioning NicheCompass as an innovative tool for spatial omics analysis.

Several avenues could enhance NicheCompass' workflow. (1) Data quality: datasets often have limited or uneven gene coverage. Experimental advancements providing higher resolution readouts[92] could improve performance. (2) Prior knowledge limitations: NicheCompass relies on incomplete and noisy databases. Program pruning, sparsity and de novo programs (Methods) mitigate this limitation, but database improvements and newly discovered pathways could enhance its capabilities. (3) Gene program limitations: although our selective gene regularization excludes causal effect genes encoding ligands and transcription factors and thus allows their prioritization by the model (Methods), there is no guarantee that prior program activity is linked to such genes, as it might instead be dominated by target gene expression. Additionally, although programs are often driven by spatial effects, some programs can be driven by cell type markers that are also differentially expressed in non-spatial analysis (Supplementary Fig. 35). Similarly, de novo programs may fail to identify genes encoding proteins that can structurally interact (for example, ligands and receptors). Incorporating structural protein data (for example, AlphaFold 2 (refs. 93,94)) could improve biological relevance. Finally, for a given program, our current approach uses the same weighting of genes across all cells; future extensions may benefit from dynamic models that adapt gene contributions to programs based on cell-specific contextual characteristics. (4) Spot-level data: NicheCompass' performance is lower on spot-level data (Supplementary Fig. 14). Spot deconvolution could enhance its utility for widely adopted technologies like Visium. (5) Spatial reference mapping: effective mapping requires comprehensive large-scale atlases[95] and consistent gene panels. Query niches absent in references can be identified but their characterization depends on shared programs (Extended Data Fig. 10). (6) Architectural enhancements: advanced graph-based encoders (for example, graph transformers[96]) and additional modalities (for example, histone modifications and protein expression) could further improve niche identification and characterization.

With the increasing availability of spatial omics data, we expect NicheCompass to become a key tool for characterizing tissue niches, enhancing our understanding of tissue architecture and responses to injury and disease.

## Online content

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

[1]Institute of AI for Health, Helmholtz Center Munich—German Research Center for Environmental Health, Neuherberg, Germany. [2]School of Computation, Information and Technology, Technical University of Munich, Munich, Germany. [3]Würzburg Institute of Systems Immunology (WüSI), University of Würzburg, Würzburg, Germany. [4]Wellcome Sanger Institute, Wellcome Genome Campus, Cambridge, UK. [5]Institute of Computational Biology, Helmholtz Center Munich—German Research Center for Environmental Health, Neuherberg, Germany. [6]Biomedical Center (BMC), Physiological Chemistry, Faculty of Medicine, Ludwig Maximilian University of Munich, Planegg-Martinsried, Germany. [7]Laboratory of Molecular Neurobiology, Department of Medical Biochemistry and Biophysics, Karolinska Institutet, Stockholm, Sweden. [8]Faculty of Medicine, University of Würzburg, Würzburg, Germany. [9]Department of Biomedical Engineering, Yale University, New Haven, CT, USA. [10]Yale Stem Cell Center and Yale Cancer Center, Yale University School of Medicine, New Haven, CT, USA. [11]Department of Pathology, Yale University School of Medicine, New Haven, CT, USA. [12]Human and Translational Immunology Program, Yale University School of Medicine, New Haven, CT, USA. [13]Ming Wai Lau Centre for Reparative Medicine, Stockholm Node, Karolinska Institutet, Stockholm, Sweden. [14]School of Life Sciences Weihenstephan, Technical University of Munich, Munich, Germany. ✉e-mail: carlos.talavera-lopez@uni-wuerzburg.de; ml19@sanger.ac.uk

## Methods

This study relies on the analysis of previously published data, adhering to ethical guidelines for human and mouse samples.

### NicheCompass model

**Dataset.** We define a spatial omics dataset as $\mathcal{D} = \{\mathbf{x}_i, \mathbf{s}_i, \mathbf{c}_i, \mathbf{y}_i\}_{i=1}^{N_{obs}}$, where $N_{obs}$ is the total number of observations (cells or spots), $\mathbf{x}_i \in \mathbb{R}^{N_{fts}}$ is the omics feature vector, $\mathbf{s}_i \in \mathbb{R}^2$ is the 2D spatial coordinate vector, $\mathbf{c}_i \in \mathbb{N}^{N_{cov}}$ is the label-encoded covariates vector (for example, sample or field of view) and $\mathbf{y}_i \in \mathbb{R}^{N_{lbl}}$ is the label vector (all vectors are row vectors). For unimodal data, $\mathbf{x}_i$ comprises raw gene expression counts such that $\mathbf{x}_i = \mathbf{x}_i^{(rna)} \in \mathbb{R}^{N_{rna}}$, where $N_{rna}$ is the number of genes. For multimodal data, $\mathbf{x}_i$ combines raw gene expression counts and chromatin accessibility peak counts, such that $\mathbf{x}_i = \mathbf{x}_i^{(rna)} || \mathbf{x}_i^{(atac)}$ (concatenation) with $\mathbf{x}_i^{(atac)} \in \mathbb{R}^{N_{atac}}$, where $N_{atac}$ is the number of peaks. We define corresponding matrices across observations with italic uppercase letters, for example, $X = [\mathbf{x}_1, ..., \mathbf{x}_{N_{obs}}]^T \in \mathbb{R}^{N_{obs} \times N_{fts}}$.

**Neighborhood graph.** We model the spatial structure of $\mathcal{D}$ using a neighborhood graph $\mathcal{G} = (\mathcal{V}, \mathcal{E}, X, Y)$, where each node $v_i \in \mathcal{V}$ represents an observation, each edge $(v_i, v_j) \in \mathcal{E}$ indicates spatial neighbors, $\mathbf{x}_i$ is the attribute vector and $\mathbf{y}_i$ is the label vector of node $v_i$. $\mathcal{G}$ is a disconnected graph composed of sample-specific, symmetric $k$-NN subgraphs $\mathcal{G}_1, ..., \mathcal{G}_{N_{spl}}$ determined using Euclidean distances, where $N_{spl}$ is the number of samples. Using this strategy, we adapt to variable observation densities in tissue[26], whereas alternative approaches, such as fixed-radius neighborhood graphs, can be used to consider local observation densities. We derive a spatial adjacency matrix $A \in \{0,1\}^{N_{obs} \times N_{obs}}$ from $\mathcal{G}$, where $A_{i,j} = 1$ if $(v_i, v_j) \in \mathcal{E}$ and $A_{i,j} = 0$ otherwise.

**Node labels.** For each observation $i$, we define a neighborhood omics feature vector $\mathbf{x}'_i$:

$$\mathbf{x}'_i = \sum_{j \in \mathcal{N}(i) \cup \{i\}} \left[ \frac{\mathbf{x}_j}{\sqrt{d_j d_i}} \right]$$

where $d_i$ denotes node degree, including a self-loop ($d_i = \sum_{j \in \mathcal{N}(i) \cup \{i\}} 1$). This aggregation combines node $i$'s omics feature vector with those of its neighbors $j \in N(i)$, weighted by a graph convolution norm operator[97]. Self-loops model autocrine signaling, while neighboring nodes capture juxtacrine and paracrine signaling. Node labels are defined as $\mathbf{y}_i = \mathbf{x}_i || \mathbf{x}'_i$.

**Covariates.** The covariates vector $\mathbf{c}_i$ models confounding effects. For multi-sample datasets, the sample ID ($k_i$) is used as the first covariate ($C_{i,1} = k_i$). Additional covariates, such as field of view and donor, are included if available to account for hierarchical effects. We further introduce a one-hot-encoded notation of covariate vectors with each covariate $l = 1, ..., N_{cov}$ represented by a separate vector $\mathbf{c}_i^{(l)} \in \{0,1\}^{N_{cat}(l)}$, where $N_{cat}(l)$ is the number of unique categories of covariate $l$. Given that $\mathcal{G}$ is composed of sample-specific subgraphs, some covariates (for example, sample, donor) are tied to connected components. We denote such covariates as pure ($L_p$), while covariates that vary within components (for example, field of view) are denoted as mixed ($L_m$).

**Gene programs.** Prior programs are represented by two binary program gene matrices $P^{(pr,rna)}, P'^{(pr,rna)} \in \{0,1\}^{N_{pr} \times N_{rna}}$, where $N_{pr}$ is the number of prior programs. $P^{(pr,rna)}$ indicates genes in the self component, while $P'^{(pr,rna)}$ indicates genes in the neighborhood component. For multimodal data, two additional binary program peak matrices, $P^{(pr,atac)}$ and $P'^{(pr,atac)} \in \{0,1\}^{N_{pr} \times N_{atac}}$, capture peaks linked to genes in the self components and neighborhood components, respectively. $P^{(pr,rna)}$ and $P'^{(pr,rna)}$ must be provided to NicheCompass by in-built database APIs or custom user inputs. By default, $P^{(pr,atac)}$ and $P'^{(pr,atac)}$ are derived

from program gene matrices by associating peaks overlapping gene bodies or promoter regions (up to 2,000 bp upstream of transcription start sites); however, users can customize these to represent specific regulatory networks. De novo programs are analogously defined by binary matrices $P^{(nv,rna)}, P'^{(nv,rna)} \in \{0,1\}^{N_{nv} \times N_{rna}}$ and, for multimodal data, $P^{(nv,atac)}$ and $P'^{(nv,atac)} \in \{0,1\}^{N_{nv} \times N_{atac}}$, where $N_{nv}$ is the number of de novo programs (default, $N_{nv} = 100$). In $P^{(nv,rna)}$ and $P'^{(nv,rna)}$, elements are set to 1 for genes not included in the respective self or neighborhood components of prior programs. In peak matrices, elements are set to 1 for peaks linked to genes. The total number of programs is $N_{gp} = N_{pr} + N_{nv}$.

**Default prior programs.** NicheCompass provides default prior programs through APIs with interaction databases. For cell–cell-communication programs, ligand–receptor interactions are retrieved from OmniPath[37] and metabolite-sensor interactions from MEBOCOST[38]. For transcriptional regulation programs, transcription factors and their downstream genes are retrieved from CollecTRI[42] through decoupler[40]. For combined interaction programs, NicheNet's regulatory potential matrix (V2)[39], consisting of ligands, receptors and downstream target genes, is used. As recommended by MultiNicheNet[41], programs are filtered to include at most 250 target genes, ranked by regulatory score. In our experiments, we filtered subsets within prior programs and merged programs if they shared at least 90% source and target genes. This resulted in 2,925 (2,904) mouse (human) prior programs, including 548 (490) ligand–receptor programs, 114 (116) metabolite-sensor programs, 1,286 (1,225) combined interaction programs and 977 (1,073) transcriptional regulation programs (the latter were only included in multimodal scenarios).

**Model overview.** NicheCompass extends the variational graph autoencoder framework[48] to enable interpretable, scalable and integrative modeling of spatial multi-omics data. The model includes a graph encoder and a multi-module decoder, trained in a self-supervised, multi-task learning setup with node-level and edge-level tasks. The decoder comprises a graph decoder to reconstruct $A$ from $Z$ and two omics decoders per modality: a self-omics decoder to reconstruct modality-specific features $X^{(mod)}$ and a neighborhood omics decoder to reconstruct neighborhood features $X'^{(mod)}$. This ensures embeddings $Z$ integrate spatial information from $\mathcal{G}$ and molecular information from $X$ and $X'$, thus providing spatially and molecularly consistent embeddings $\mathbf{z}_i \in \mathbb{R}^{N_{gp}}$ for each observation $i$. Program matrices are used to mask the reconstruction of $X$ and $X'$, ensuring each feature $u$ in $\mathbf{Z}_{:,u}$ represents a spatial program. Embeddings for prior programs are denoted as $Z^{(pr)} \in \mathbb{R}^{N_{obs} \times N_{pr}}$ and those for de novo programs are denoted as $Z^{(nv)} \in \mathbb{R}^{N_{obs} \times N_{nv}}$, with $\mathbf{z}_i = \mathbf{z}_i^{(pr)} || \mathbf{z}_i^{(nv)}$. Following the variational autoencoder standard, we use a standard normal prior for the latent variables $Z_u^{(i)} \sim \mathcal{N}(0,1)$ and apply the reparameterization trick to enable end-to-end training by backpropagation.

**Encoder.** The first layer of the graph encoder is fully connected with hidden size $N_{hid} = N_{gp}$, serving two purposes: learning internal cell or spot representations from the full omics feature vector $\mathbf{x}_i$ before neighborhood aggregation and reducing the dimensionality of $\mathbf{x}_i$ when $N_{fts} > N_{gp}$. This layer is followed by two parallel message-passing layers that compute the mean ($\boldsymbol{\mu}_i$) and log standard deviation ($\log(\boldsymbol{\sigma}_i)$) vectors of the variational posterior, where $\boldsymbol{\mu}_i$ is extracted as cell embedding vector $\mathbf{z}_i$. The default model uses graph attention layers with dynamic attention[98] ($N_{head} = 4$); in NicheCompass Light, graph convolutional layers replace graph attention layers (Supplementary Methods). Additionally, the model learns an embedding matrix $W^{(emb\_e(l))} \in \mathbb{R}^{N_{emb} \times N_{cat}(l)}$ for each covariate $l$, where $N_{emb}$ is the embedding size, to retrieve an embedding vector $\mathbf{e}_i^{(l)}$ from the one-hot-encoded vector representation $\mathbf{c}_i^{(l)}$. The final covariate embedding is $\mathbf{e}_i = \mathbf{e}_i^{(1)} || \cdots || \mathbf{e}_i^{(N_{cov})} \in \mathbb{R}^{N_{emb}}$.

**Decoder.** The graph decoder reconstructs $A$ using cosine similarity between node embeddings, restricted to nodes with identical pure categorical covariates (for example, same sample):

$$\widetilde{A}_{i,j} = \text{cosine similarity} \left( \mathbf{z}_i, \mathbf{z}_j \right) = \frac{\mathbf{z}_i \cdot \mathbf{z}_j}{|\mathbf{z}_i| \, |\mathbf{z}_j|}$$

Omics decoders reconstruct node labels $Y$ by estimating mean parameters $\Phi_{i,f}, \Phi'_{i,f}$ of negative binomial distributions that generate omics features ($X_f^{(i)} \sim \mathcal{NB}\left(\Phi_{i,f}, \theta_f\right)$ and $X_f'^{(i)} \sim \mathcal{NB}\left(\Phi'_{i,f}, \theta'_f\right)$, where $f$ is an omics feature, $X^{(i)}$ and $X'^{(i)}$ are random variables and $\theta_f, \theta'_f$ represent inverse dispersion parameters). They are composed of modality-specific single-layer linear decoders such that each embedding feature $u$ in $Z^{(\text{pr})}_{:,u}$ is incentivized to learn the activity of a specific prior program. This is achieved by prior program matrices ($P^{(\text{pr,rna})}$, $P'^{(\text{pr,rna})}, P^{(\text{pr,atac})}, P'^{(\text{pr,atac})}$) constraining decoder contributions to specific genes or peaks. For instance, if $P_{u,q}^{(\text{pr,rna})} = 1$, embedding feature $Z_{:,u}$ contributes to reconstructing gene $q$ in the self component. Similar logic applies to neighborhood components and multimodal features. $Z_{i,u}$ can therefore be interpreted as observation $i$'s representation of program $u$, where the self component of $u$ is composed of all genes $q$ and peaks $s$ for which $P_{u,q}^{(\text{pr,rna})} = 1$ and $P_{u,s}^{(\text{pr,atac})} = 1$, and its neighborhood component of all genes $r$ and peaks $t$ for which $P_{u,r}'^{(\text{pr,rna})} = 1$ and $P_{u,t}'^{(\text{pr,atac})} = 1$. De novo programs are similarly masked using $P^{(\text{nv,rna})}, P^{(\text{nv,atac})}, P'^{(\text{nv,rna})}$ and $P'^{(\text{nv,atac})}$, allowing them to reconstruct omics features not included in prior knowledge. Confounding effects are removed by injecting covariate embeddings $\mathbf{e}_i$ into omics decoders. For observation $i$, the reconstructed mean parameter is:

$$\phi_i^{*(\text{mod})} = \text{Softmax}\left( P^{(\text{pr,mod})^T} \circ W^{(\text{pr}\_\phi^{*(\text{mod})})} \mathbf{z}_i^{\text{pr}} \right.$$
$$\left. + P^{(\text{nv,mod})^T} \circ W^{(\text{nv}\_\phi^{*(\text{mod})})} \mathbf{z}_i^{\text{nv}} + W^{(\text{emb}\_\phi^{*(\text{mod})})} \mathbf{e}_i \right) \exp\left( \iota_i^{*(\text{mod})} \right)$$

where * indicates either the self component or neighborhood component, mod represents the modality (rna or atac), $\iota_i^{*(\text{mod})}$ is the empirical log library size and $W^{(\text{pr}\_\phi^{*(\text{mod})})} \in \mathbb{R}^{N_{\text{mod}} \times N_{\text{pr}}}$, $W^{(\text{nv}\_\phi^{*(\text{mod})})} \in \mathbb{R}^{N_{\text{mod}} \times N_{\text{nv}}}$ and $W^{(\text{emb}\_\phi^{*(\text{mod})})} \in \mathbb{R}^{N_{\text{mod}} \times N_{\text{emb}}}$ are learnable weights. The Softmax activation operates across features, constraining omics decoders to output mean proportions. The multiplication with the empirical library size ensures the same size factors as in the input domain.

**Neighbor sampling data loaders.** NicheCompass uses mini-batch training with inductive neighbor sampling data loaders[99] for scalability and efficiency. For each node $v_i \in \mathcal{V}$, only $n = 4$ sampled neighbors from $\mathcal{G}$ are used for message passing. NicheCompass' multi-task architecture uses two data loaders: a node-level loader to reconstruct $X$ and $X'$ and an edge-level loader to reconstruct $A$. One iteration of the model includes one forward pass per loader and a joint backward pass for simultaneous gradient computation. For the node-level loader, a batch consists of $N_{\mathcal{V}_{\text{bat}}}$ randomly selected nodes $\mathcal{V}_{\text{bat}} \in \mathcal{V}$, shuffled at each iteration. For the edge-level loader, a batch includes $N_{\mathcal{E}_{\text{bat}}}$ positive node pairs $(i,j) \in \mathcal{E}$, shuffled per iteration, and an equal number of randomly sampled negative pairs $(i,j)$ for which $A_{i,j} = 0$. We denote the corresponding batch of positive and sampled negative node pairs as $\mathcal{E}_{\text{bat}}$. To ensure valid negative examples, we retain only node pairs that share identical pure covariates ($\mathbf{c}_i^{(l)} = \mathbf{c}_j^{(l)} \forall l \in L_{\text{p}}$). The final edge batch $\mathcal{E}_{\text{rec}} = \{\mathcal{E}_{\text{rec}}^+, \mathcal{E}_{\text{rec}}^-\}$ consists of positive pairs $\mathcal{E}_{\text{rec}}^+ = \{(i,j) \in \mathcal{E}_{\text{bat}} | A_{i,j} = 1\}$ and valid negative pairs $\mathcal{E}_{\text{rec}}^- = \{(i,j) \in \mathcal{E}_{\text{bat}} | A_{i,j} = 0 \text{ and } \mathbf{c}_i^{(l)} = \mathbf{c}_j^{(l)} \forall l \in L_{\text{p}}\}$.

**Program pruning.** To prioritize relevant programs, NicheCompass uses a dropout-based pruning mechanism. This addresses issues with overlapping genes across programs that dilute correlations between embeddings $Z$ and program member genes. After a warm-up period, pruning is based on each program's contribution to reconstructing $X^{(\text{rna})}$ and $X'^{(\text{rna})}$. Contributions ($\delta_u$) are calculated by aggregating absolute values of gene expression decoder weights at the program level

(across self and neighborhood components) and scaling them by an estimate of the mean absolute embeddings across observations. This estimate is obtained as the exponential moving average of batch-wise forward passes. The maximum contribution ($\delta_{\max}$) serves as a reference, and programs with contributions below a threshold ($\tau * \delta_{\max}$, where $\tau$ is a hyperparameter) are dropped. To balance pruning, two aggregation methods are used: sum-based (to avoid penalizing programs with many unimportant but few very important genes) and non-zero mean-based (to prevent prioritizing programs with many genes). Pruning is applied separately to prior and de novo programs, with independent $\delta_{\max}$ calculations.

**Program regularization.** To prioritize critical genes within programs while considering different functional importances (for example, a ligand is critical for the pathway), NicheCompass uses selective regularization. Genes in prior programs are categorized (ligand, receptor, transcription factor, sensor, target gene), and an L1 regularization loss is applied to decoder weights of specified categories. In our analyses, regularization was applied to target genes. De novo programs, which may include hundreds to thousands of genes, are similarly regularized with an L1 loss to encourage specificity. If decoder weights for gene expression are regularized to zero, corresponding weights for chromatin accessibility are set to zero, effectively deactivating those peaks within the program.

**Loss function.** With unimodal data, the loss function consists of four components: (1) a binary cross-entropy loss for reconstructing edges in $A$; (2) a negative binomial loss for reconstructing the self component $X^{(\text{rna})}$; that is, the nodes' gene expression counts; (3) a negative binomial loss for reconstructing the neighborhood component $X'^{(\text{rna})}$; that is, the aggregated gene expression counts of node neighborhoods; and (4) the Kullback–Leibler divergence between variational posteriors and standard normal priors for latent variables. In multimodal scenarios, additional negative binomial losses are included for reconstructing self ($X^{(\text{atac})}$) and neighborhood peak counts ($X'^{(\text{atac})}$). The mini-batch-wise formulation of the edge reconstruction loss is:

$$\mathcal{L}^{(\text{edge})}\left(\widetilde{A}; A, \mathcal{E}_{\text{rec}}\right) = -\frac{1}{|\mathcal{E}_{\text{rec}}|} \sum_{(i,j) \in \mathcal{E}_{\text{rec}}} \left[ \omega_{\text{pos}} A_{i,j} \log\left(\sigma\left(\widetilde{A}_{i,j}\right)\right) \right.$$
$$\left. + (1 - A_{i,j}) \log\left(1 - \sigma\left(\widetilde{A}_{i,j}\right)\right) \right].$$

where $\widetilde{A}$ represents edge reconstruction logits computed by the cosine similarity graph decoder. To balance the contribution of positive and negative edge pairs, a weight $\omega_{\text{pos}} = \frac{|\mathcal{E}_{\text{rec}}^-|}{|\mathcal{E}_{\text{rec}}^+|}$ is applied as $|\mathcal{E}_{\text{rec}}^+| \geq |\mathcal{E}_{\text{rec}}^-|$, owing to filtering negative pairs where pure covariates differ.

The mini-batch-wise formulation of the modality-specific omics reconstruction losses is:

$$\mathcal{L}^{(\text{mod})}\left(\Phi^{(\text{mod})}, \Phi'^{(\text{mod})}, \boldsymbol{\theta}^{(\text{mod})}, \boldsymbol{\theta}'^{(\text{mod})}; X^{(\text{mod})}, X'^{(\text{mod})}, \mathcal{V}_{\text{bat}}\right)$$
$$= \frac{1}{N_{\mathcal{V}_{\text{bat}}}} \sum_{i \in \mathcal{V}_{\text{bat}}} \mathcal{L}_i^{(\text{mod})}\left(\boldsymbol{\Phi}_i^{(\text{mod})}, \boldsymbol{\Phi}_i'^{(\text{mod})}, \boldsymbol{\theta}^{(\text{mod})}, \boldsymbol{\theta}'^{(\text{mod})}; \mathbf{x}_i^{(\text{mod})}, \mathbf{x}_i'^{(\text{mod})}\right)$$

where the observation-level loss includes the self component and neighborhood component negative binomial losses (Supplementary Methods):

$$\mathcal{L}_i^{(\text{mod})}\left(\boldsymbol{\Phi}_i^{(\text{mod})}, \boldsymbol{\Phi}_i'^{(\text{mod})}, \boldsymbol{\theta}^{(\text{mod})}, \boldsymbol{\theta}'^{(\text{mod})}; \mathbf{x}_i^{(\text{mod})}, \mathbf{x}_i'^{(\text{mod})}\right)$$
$$= \text{NBL}\left(\boldsymbol{\Phi}_i^{(\text{mod})}, \boldsymbol{\theta}^{(\text{mod})}; \mathbf{x}_i^{(\text{mod})}\right) + \text{NBL}\left(\boldsymbol{\Phi}_i'^{(\text{mod})}, \boldsymbol{\theta}'^{(\text{mod})}; \mathbf{x}_i'^{(\text{mod})}\right)$$

where mod represents the modality, $\boldsymbol{\theta}^{*(\text{mod})}$ are feature-specific learned inverse dispersion parameters and $\boldsymbol{\Phi}_i^{*(\text{mod})}$ are the estimated means, retrieved as output of the omics decoders.

The L1 regularization losses are defined as:

$$\mathcal{L}^{(\text{L1,pr})}\left(W^{(\text{pr}\_\phi^{*(\text{rna})})}\right) = \sum_{u=1}^{N_{\text{pr}}} \sum_{q=1}^{N_{\text{rna}}} \left| W_{q,u}^{(\text{pr}\_\phi^{*(\text{rna})})} \right| \circ I_{q,u}^{(\text{pr}\_\phi^{*(\text{rna})})}$$

and

$$\mathcal{L}^{(\text{L1,nv})}\left(W^{(\text{nv}\_\phi^{*(\text{rna})})}\right) = \sum_{u=1}^{N_{\text{nv}}} \sum_{q=1}^{N_{\text{rna}}} \left| W_{q,u}^{(\text{nv}\_\phi^{*(\text{rna})})} \right|$$

where $I^{(\text{pr}\_\phi^{*(\text{rna})})} \in \{0,1\}^{N_{\text{rna}} \times N_{\text{pr}}}$ is an indicator matrix for selective regularization of prior programs, with an entry of 1 indicating that the corresponding gene is part of a regularized category.

The mini-batch-wise formulation of the KL divergence consists of node-level and edge-level components:

$$\mathcal{L}^{(\text{KL})}\left(M, \Sigma; X, \mathcal{V}_{\text{bat}}, \mathcal{E}_{\text{bat}}\right) =$$

$$\frac{1}{N_{\mathcal{V}_{\text{bat}}}} \sum_{i \in \mathcal{V}_{\text{bat}}} \mathcal{L}_i^{(\text{KL})}\left(\boldsymbol{\mu}_i, \boldsymbol{\sigma}_i; \mathbf{x_i}\right)$$

$$+ \frac{1}{4 * N_{\mathcal{E}_{\text{bat}}}} \sum_{(i,j) \in \mathcal{E}_{\text{bat}}} \mathcal{L}_i^{(\text{KL})}\left(\boldsymbol{\mu}_i, \boldsymbol{\sigma}_i; \mathbf{x_i}\right) + \mathcal{L}_j^{(\text{KL})}\left(\boldsymbol{\mu}_j, \boldsymbol{\sigma}_j; \mathbf{x_j}\right)$$

with the observation-level loss:

$$\mathcal{L}_i^{(\text{KL})}\left(\boldsymbol{\mu}_i, \boldsymbol{\sigma}_i; \mathbf{x_i}\right) = D_{\text{KL}}\left(q_{\boldsymbol{\mu}_i, \boldsymbol{\sigma}_i}\left(Z^{(i)} | X^{(i)}\right) \| p\left(Z^{(i)}\right)\right)$$

$$= -\frac{1}{2} \sum_{u=1}^{N_{\text{gp}}} \left[1 + \log\left(\boldsymbol{\sigma}_{i_u}^2\right) - \boldsymbol{\mu}_{i_u}^2 - \boldsymbol{\sigma}_{i_u}^2\right]$$

where $\boldsymbol{\mu}_i$ and $\boldsymbol{\sigma}_i$ are the estimated mean and standard deviation of the variational posterior normal distribution.

The final mini-batch loss combines all components:

$$\mathcal{L}\left(M, \Sigma, \Phi, \Phi', \boldsymbol{\theta}, \boldsymbol{\theta}', \tilde{A}, W^{(\text{rna})}; A, X, X', \mathcal{V}_{\text{bat}}, \mathcal{E}_{\text{bat}}\right)$$

$$= \mathcal{L}^{(\text{KL})}\left(M, \Sigma; X, \mathcal{V}_{\text{bat}}, \mathcal{E}_{\text{bat}}\right)$$

$$+ \lambda^{(\text{edge})} \mathcal{L}^{(\text{edge})}\left(\tilde{A}; A, \mathcal{E}_{\text{rec}}\right)$$

$$+ \lambda^{(\text{rna})} \mathcal{L}^{(\text{rna})}\left(\Phi^{(\text{rna})}, \Phi'^{(\text{rna})}, \boldsymbol{\theta}^{(\text{rna})}, \boldsymbol{\theta}'^{(\text{rna})}; X^{(\text{rna})}, X'^{(\text{rna})}, \mathcal{V}_{\text{bat}}\right)$$

$$+ \lambda^{(\text{atac})} \mathcal{L}^{(\text{atac})}\left(\Phi^{(\text{atac})}, \Phi'^{(\text{atac})}, \boldsymbol{\theta}^{(\text{atac})}, \boldsymbol{\theta}'^{(\text{atac})}; X^{(\text{atac})}, X'^{(\text{atac})}, \mathcal{V}_{\text{bat}}\right)$$

$$+ \lambda^{(\text{L1,pr})} \mathcal{L}^{(\text{L1,pr})}\left(W^{(\text{pr}\_\phi^{(\text{rna})})}\right)$$

$$+ \lambda^{(\text{L1,pr})} \mathcal{L}^{(\text{L1,pr})}\left(W^{(\text{pr}\_\phi'^{(\text{rna})})}\right)$$

$$+ \lambda^{(\text{L1,nv})} \mathcal{L}^{(\text{L1,nv})}\left(W^{(\text{nv}\_\phi^{(\text{rna})})}\right)$$

$$+ \lambda^{(\text{L1,nv})} \mathcal{L}^{(\text{L1,nv})}\left(W^{(\text{nv}\_\phi'^{(\text{rna})})}\right)$$

where $\lambda$ values denote weighting factors.

**Spatial reference mapping.** To map unseen query datasets onto spatial reference atlases, we use weight-restricted fine-tuning inspired by architectural surgery[95]. A NicheCompass model is first trained to construct a reference. During query training, all weights are frozen except for covariate embedding matrices ($W^{(\text{emb}\_e^{(l)})}$), allowing us to capture query-specific variation without catastrophic forgetting. Programs can be pruned differently during query training owing to updating exponential moving averages of embeddings.

**Program feature importances.** Gene and peak importances for each program are determined using the learned weights of omics decoders. Absolute values of the gene expression or chromatin accessibility decoder weights are normalized across genes or peaks in the self and neighborhood components, ensuring that the importances sum to 1 per program.

**Program activities.** NicheCompass embeddings quantify pathway activity in cells or spots but are agnostic to sign. To ensure positive embedding values represent upregulation, the embeddings are adjusted based on omics decoder weight signs. For prior programs, embeddings are reversed if the aggregated weight of source genes (or target genes if source genes are absent) is negative. For de novo programs, the sign is reversed if the aggregated weight of all genes is negative. These sign-corrected embeddings are referred to as program activities.

**Differential testing of program activities.** We test differential program activity between groups of interest using the logarithm of the Bayes factor ($\log K$), a Bayesian generalization of the $P$ value[100]. The hypothesis $H_0 : \mathbb{E}_a\left[Z_u^{(a)}\right] > \mathbb{E}_b\left[Z_u^{(b)}\right]$ is tested against $H_1 : \mathbb{E}_a\left[Z_u^{(a)}\right] \leq \mathbb{E}_b\left[Z_u^{(b)}\right]$, where $u$ is the program index, and $Z^{(a)}$ and $Z^{(b)}$ denote random variables for the program activities of group $a$ and comparison group $b$. The test statistic, $\log K = \log \frac{p(H_0)}{p(H_1)} = \log \frac{p(H_0)}{1 - p(H_0)}$, quantifies the evidence for $H_0$ (Supplementary Methods). Programs with $|\log K| \geq 2.3$ are considered differentially expressed, corresponding to strong evidence[101], equivalent to a relative ratio of probabilities of $\exp(2.3) \approx 10$.

**Selection of characterizing niche programs.** To identify characterizing programs, we first perform a one-vs-rest differential log Bayes factor test to determine enriched programs. From these, we select two programs per niche based on the correlation between program activities and the expression of the program's important target genes and ligand-encoding and receptor-encoding or enzyme-encoding and sensor-encoding genes.

**Program communication potential scores.** To compute source and target communication potential scores, we first scale gene expression between 0 and 1 to avoid bias towards highly expressed genes. For each program, the scaled expression of each member gene is multiplied by its corresponding omics decoder weight, yielding program-specific scores for each gene in the self and neighborhood components. These scores are averaged within each component and then multiplied by the program activity. The target score is derived from the self component average, while the source score is based on the neighborhood component average. Negative scores are set to 0.

**Program communication strengths.** To compute program communication strengths, we create program-specific $k$-NN graphs to reflect program-specific length scales (defaulting to $9$). For each pair of neighboring nodes, we calculate directional communication strengths by multiplying their source and target communication potential scores. These strengths can be aggregated at the cell or niche level and are normalized between 0 and 1.

## Statistics and reproducibility

**Datasets.** All datasets used in this study except for simulated data were previously published (Data Availability section). No statistical method was used to predetermine sample size, and no data were excluded from the analyses unless explicitly stated. Cell type labels and metadata were sourced from the original publications unless specified otherwise.

*Simulated data.* We customized SRTsim[72] to enable the mixing of reference-based and freely simulated genes and the injection of ground-truth spatial program activity into niches using an additive gene expression model. Our version is available at https://github.com/Lotfollahi-lab/nichecompass-reproducibility. Using STARmap mouse brain reference data[72], we simulated 10,000 cells distributed across eight niches with diverse cell type compositions and 1,105 genes (Supplementary Table 1 and Supplementary Methods). To create the spot-level version, we segmented the tissue into 55 μm diameter circular bins, resulting in 1,587 spots with an average of 6.44 cells per

spot. Gene expression counts were aggregated within bins to produce spot-level data.

*seqFISH mouse organogenesis.* This dataset includes 57,536 cells across six sagittal tissue sections from three 8–12 somite stage mouse embryos: 19,451 (embryo 1), 14,891 (embryo 2) and 23,194 (embryo 3). The dataset contains 351 genes, and imputation was performed by the original authors to generate a full transcriptome (29,452 features). Cells designated as low quality by the original authors were excluded, resulting in a final set of 52,568 cells. Given that imputation was performed on log counts, we computed a reverse log normalization and rounded the results to obtain estimated counts. We filtered genes based on their maximum imputed counts per cell: genes with counts of >141 (the maximum in the original data) were removed, resulting in 29,239 features; of these, we selected the 5,000 most spatially variable genes using Moran's $I$ score, computed by squidpy.gr.spatial_autocorr()[102]. For multi-sample models, we defined the sample as the only covariate, and tissue sections were treated as separate samples.

*SlideSeqV2 mouse hippocampus dataset.* This dataset consists of a puck with 41,786 observations at near-cellular resolution and 4,000 genes. Given that the dataset contained log counts, we computed a reverse log normalization and rounded the results to obtain raw counts.

*MERFISH mouse liver dataset.* This dataset includes 395,215 cells and 347 genes. Following the vignette from squidpy (https://squidpy.readthe-docs.io/en/stable/notebooks/tutorials/tutorial_vizgen_mouse_liver.html), we filtered cells with <50 counts, leaving 367,235 cells. Cell types were annotated using a typical scanpy[103] workflow, encompassing PCA (20 components), $k$-NN graph computation (ten neighbors), Leiden clustering and marker gene-based annotation using the markers from https://static-content.springer.com/esm/art%3A10.1038%2Fs41421-021-00266-1/MediaObjects/41421_2021_266_MOESM1_ESM.xlsx.

*NanoString CosMx human NSCLC dataset.* This dataset includes 800,559 cells across eight tissue sections from five donors (donor 6, squamous cell carcinoma; others: adenocarcinoma). Cell counts per section are 93,206 cells (donor 1, replicate 1), 93,206 cells (donor 1, replicate 2), 91,691 cells (donor 1, replicate 3), 91,691 cells (donor 2), 77,391 cells (donor 3, replicate 1), 115,676 (donor 3, replicate 2), 66,489 cells (donor 4) and 76,536 cells (donor 5). Expression levels of 960 genes were measured across 20–45 fields of view per section. After filtering cells with <50 counts, cells without spatial coordinates and cells without cell type annotation, 702,199 cells remained. For multi-sample models, sample, field of view and donor were defined as covariates.

*Xenium human breast cancer dataset.* This dataset includes 286,523 cells across two replicates (replicate 1, 167,780; replicate 2, 118,752) with 313 genes. Cells with less than ten counts or non-zero counts for fewer than three genes were filtered, leaving 282,363 cells. Cell types and states were annotated using a typical scanpy[103] workflow, encompassing PCA (50 components), $k$-NN graph computation (50 neighbors), Leiden clustering and marker gene-based annotation.

*STARmap PLUS mouse CNS dataset.* This dataset includes 1,091,527 cells and 1,022 genes. Genes expressed in at least 10% of cells across all samples were retained. Coronal tissue sections were aligned to the Allen Brain Atlas[71] using STAlign[104]. For model training, sample was defined as a covariate. For ablation studies, only the first sagittal tissue section was used (91,246 cells).

*MERFISH whole mouse brain dataset.* This dataset includes 8.4 million cells across 239 sections from four animals (animal 1, 4,167,869 cells; animal 2, 1,915,592 cells; animal 3, 2,081,549 cells; animal 4, 215,278 cells) with 1,122 genes. For model training, sample and donor were defined as covariates. To integrate this dataset with the STARmap PLUS mouse CNS dataset, filtering was applied to only keep 432 overlapping genes.

*Spatial ATAC–RNA-seq mouse brain dataset.* This dataset consists of 9,215 spot-level observations, with 22,914 genes and 121,068 peaks. Genes and peaks present in <46 cells were filtered. The top 3,000 spatially variable genes and 15,000 peaks were selected using Moran's $I$ spatial autocorrelation. Non-annotated genes were excluded using GENCODE 25, resulting in 2,785 genes. Peaks not overlapping with any gene body or promoter region were dropped, leaving 3,337 peaks.

*Stereo-seq mouse embryo dataset.* This dataset includes 5,913 spot-level observations with ground-truth niche labels and 25,568 genes. The top 3,000 spatially variable genes were selected based on Moran's $I$ score. Niche coherence scores at the spot level were computed using a standard preprocessing workflow including read depth normalization, log transformation of gene expression counts, Leiden clustering and cluster labels as proxies for cell types.

**Experiments.** All experiments were performed on a NVIDIA A100-PCIE-40 GB GPU. No blinding was applicable in this study because no sample group allocation was performed. Clusters were computed with scanpy.tl.leiden() unless otherwise specified.

*SlideSeqV2 mouse hippocampus.* Each method was trained once using a symmetric $k$-NN graph ($k$ = 4). Clustering resolutions were adapted to recover fine-grained anatomical niches.

*SlideSeqV2 mouse hippocampus 25% subsample.* A 25% subsample was created by sampling cells from the tissue's center along the $y$ axis while retaining the full $x$ axis range. The analysis followed the same workflow as the full dataset experiment.

*Simulated data.* For each method, we performed $n$ = 8 training runs, varying the number of neighbors from 4 to 16 at increments of four (two runs each). Clustering resolutions were adapted until the number of niches matched the ground truth.

*NanoString CosMx human NSCLC 10% subsample.* To create a 10% subsample, cells were sampled field-by-field until the threshold was reached. The analysis followed the workflow of the SlideSeqV2 mouse hippocampus experiment. Separate $k$-NN graphs were computed for each sample and combined into a disconnected graph. The standard NicheCompass model included sample and field of view as covariates, and clusters were annotated with niche labels based on cell type proportions.

*Single-sample and integration benchmarking.* For each method, we conducted $n$ = 8 training runs on full and subsampled datasets, varying neighbors from 4 to 16 in increments of four (two runs each). Subsampling included 1%, 5%, 10%, 25% and 50% of the dataset while preserving spatial consistency.

*Ablation on simulated data.* Niche identification was evaluated using Leiden clustering, adjusting resolutions to match predicted and ground-truth niche counts. Ground-truth prediction accuracy was assessed with performance metrics (NMI, ARI, HOM and COMS) from SDMBench[105]. For program inference, we identified enriched programs per niche using one-vs-rest differential testing (log Bayes factor, 4.6) and calculated F1 scores between enriched and ground-truth programs. Gene-level F1 scores were computed separately for source and target genes of prior and de novo programs by comparing the three most important inferred genes with simulated upregulated genes.

A random baseline was established by sampling random programs and genes, matching enriched counterparts in number. Mean F1 scores were reported across all niches (and all seeds, niches and configurations for the random baseline).

*Ablation on real data.* Niche identification was evaluated using *k*-means clustering, with NMI and ARI metrics computed by scib.nmi_ari_cluster_labels_kmeans()[106]. Ground-truth niche and region labels were taken from the original authors[19].

**Data visualization.** Micrographs and other visualizations displaying program activities or cell–cell communication strengths represent results from single trained models on the respective dataset, except for the seqFISH mouse organogenesis dataset in which we tested reproducibility and robustness of results across *n* = 3 seeds and *n* = 4 neighborhood graphs (Extended Data Fig. 2). Boxplot elements are always defined as center line, median; box limits, upper and lower quartiles; whiskers, 1.5× interquartile range. We used scanpy.tl.umap() to embed cells in 2D for visualization. *k*-NN graphs were computed on embeddings using scanpy.pp.neighbors(). For the 8.4 million-cell whole mouse brain spatial atlas, before neighborhood graph computation, PCA was applied using scanpy.tl.pca(). De novo programs were visualized using sunburst plots, categorizing genes into 'pathway' (inner circle) and 'gene family' (outer circle) using BioMart. Genes were colored based on their weights learned by NicheCompass. To simplify plot creation, we developed a ChatGPT-optimized prompt and supporting notebook, available at https://github.com/Lotfollahi-lab/nichecompass-reproducibility.

**Hierarchical niche identification.** Tissue niche hierarchies were identified through a two-step process. First, Leiden clustering was applied to the embeddings using scanpy.tl.leiden() to identify niches, with additional rounds of clustering for sub-niche identification. Second, hierarchical clustering was performed on the embeddings, incorporating niche labels, using scanpy.tl.dendrogram().

**Reporting summary**
Further information on research design is available in the Nature Portfolio Reporting Summary linked to this article.

## Data availability
All datasets used in this study were previously published. Processed versions are available as AnnData[107] objects for download as outlined at https://github.com/Lotfollahi-lab/nichecompass-reproducibility. The seqFISH mouse organogenesis dataset[49] was sourced from https://marionilab.cruk.cam.ac.uk/SpatialMouseAtlas. The SlideSeqV2 dataset[12] was obtained from squidpy.datasets.slideseqv2()[102]. The MERFISH mouse liver dataset was retrieved from https://info.vizgen.com/mouse-liver-access (animal 1, replicate 1). The NanoString CosMx NSCLC dataset[10] was collected from https://nanostring.com/products/cosmx-spatial-molecular-imager/ffpe-dataset/nsclc-ffpe-dataset. The Xenium human breast cancer dataset[73] was downloaded from https://www.10xgenomics.com/products/xenium-in-situ/preview-dataset-human-breast. The STARmap PLUS mouse CNS dataset[19] was obtained from https://zenodo.org/records/8327576. The MERFISH whole mouse brain dataset[89] was retrieved from https://cellxgene.cziscience.com/collections/0cca8620-8dee-45d0-aef5-23f032a5cf09. The spatial ATAC–RNA-seq mouse brain dataset[17] (postnatal day 22) was collected from https://www.ncbi.nlm.nih.gov/geo/query/acc.cgi?acc=GSE205055 (gene expression counts and spatial coordinates) and https://brain-spatial-omics.cells.ucsc.edu/ (peak counts and cell type labels). Lastly, the stereo-seq mouse embryo dataset[108] was downloaded from http://sdmbench.drai.cn (Data ID 13). To collect default ligand–receptor and transcriptional regulation programs, we used the omnipath (v.1.0.8) and decoupler (v.1.7.0) Python packages, respectively. Default metabolite-sensor programs were retrieved from https://github.com/zhengrongbin/MEBOCOST (on 18 May 2023). Default combined interaction programs were constructed using NicheNet's regulatory potential matrix, retrieved from https://zenodo.org/record/7074291.

## Code availability
NicheCompass is available as a Python package, deposited at https://doi.org/10.5281/zenodo.14621258 (ref. 109) and maintained at https://github.com/Lotfollahi-lab/nichecompass. Code to reproduce our analyses, data simulation, ablation and benchmarking experiments is retrievable from https://doi.org/10.5281/zenodo.14632687 (ref. 110) and https://github.com/Lotfollahi-lab/nichecompass-reproducibility. Documentation is provided at https://nichecompass.readthedocs.io.

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

## Acknowledgements
This research was funded in part by the Wellcome Trust (grant number 220540/Z/20/A). S.B. and I.B.-P. are supported by the Helmholtz Association under the joint research school "Munich School for Data Science—MUDS". M.L. acknowledges financial support from the Joachim Herz Stiftung. G.C.-.B. acknowledges financial support from the Knut and Alice Wallenberg Foundation (grants 2019-0107 and 2019-0089), the Swedish Cancer Society (Cancerfonden, 190394 Pj) and the Swedish Brain Foundation (FO2023-0032). A.M. and C.T.-L. received funding from the Faculty of Medicine of the Julius-Maximilians-Universität Würzburg and the Joint Federal and State Support Program for Young Academics (WISNA). The funders

had no role in study design, data collection and analysis, decision to publish or preparation of the paper. M.L. appreciates feedback and fruitful discussions with K. Meyer and G. Ciriello regarding cancer applications. S.B. is thankful to S. Rybakov and all members of the Lotfollahi Group for valuable feedback; in particular, S. Megas, A. Vahidi, K. Ly and M. Moullet. S.B. is grateful to P. Villa Fulton for feedback on figure design and to P. Villa Fulton and Pebble, Pixel, Mickey and Octavious O. Villa for their inspirational support. We thank D. Zhang for providing cell type annotations for the spatial ATAC–RNA-seq mouse brain dataset.

## Author contributions

M.L. conceived the project. S.B. designed the algorithm with feedback from M.L. S.B. implemented the algorithm. M.L. designed the experiments with contributions from S.B. and C.T.-L. S.B. performed the benchmarking, data simulation and ablation experiments with feedback from M.L. and E. Armingol. A.B. curated the STARmap PLUS mouse data and analyzed it with contributions from S.B. and M.L. C.T.-L. curated the Xenium human breast cancer data and C.T.-L., S.B. and A.M. analyzed it. I.B.-P. performed the analysis of the NanoString CosMx human NSCLC data with contributions from S.B., M.L. and C.T.-L. S.B. curated the remaining datasets and performed all other analyses with contributions from M.L., A.M.F., A.Y., F.M., O.A.B., E. Agirre., G.C.-B. and R.F. F.J.T. supported M.L. during his work on the project and provided the environment to perform the work. S.B., I.B.-P., M.L. and C.T.-L. wrote the paper with contributions from A.B. M.L. and C.T.-L. supervised the research. All authors reviewed the paper.

## Competing interests

S.B. is a part-time employee at Avanade Deutschland. M.L. owns interests in Relation Therapeutics and is a scientific cofounder and part-time employee at AIVIVO. F.J.T. consults for Immunai Inc., Singularity Bio B.V., CytoReason Ltd. and Omniscope and has an ownership interest in Dermagnostix GmbH and Cellarity. As of 1 February, 2025, C.T-L. is an employee at Cellzome GmbH/GSK. His contributions were done while being at the University of Würzburg. The other authors declare no competing interests.

## Additional information

**Extended data** is available for this paper at https://doi.org/10.1038/s41588-025-02120-6.

**Correspondence and requests for materials** should be addressed to Carlos Talavera-López or Mohammad Lotfollahi.

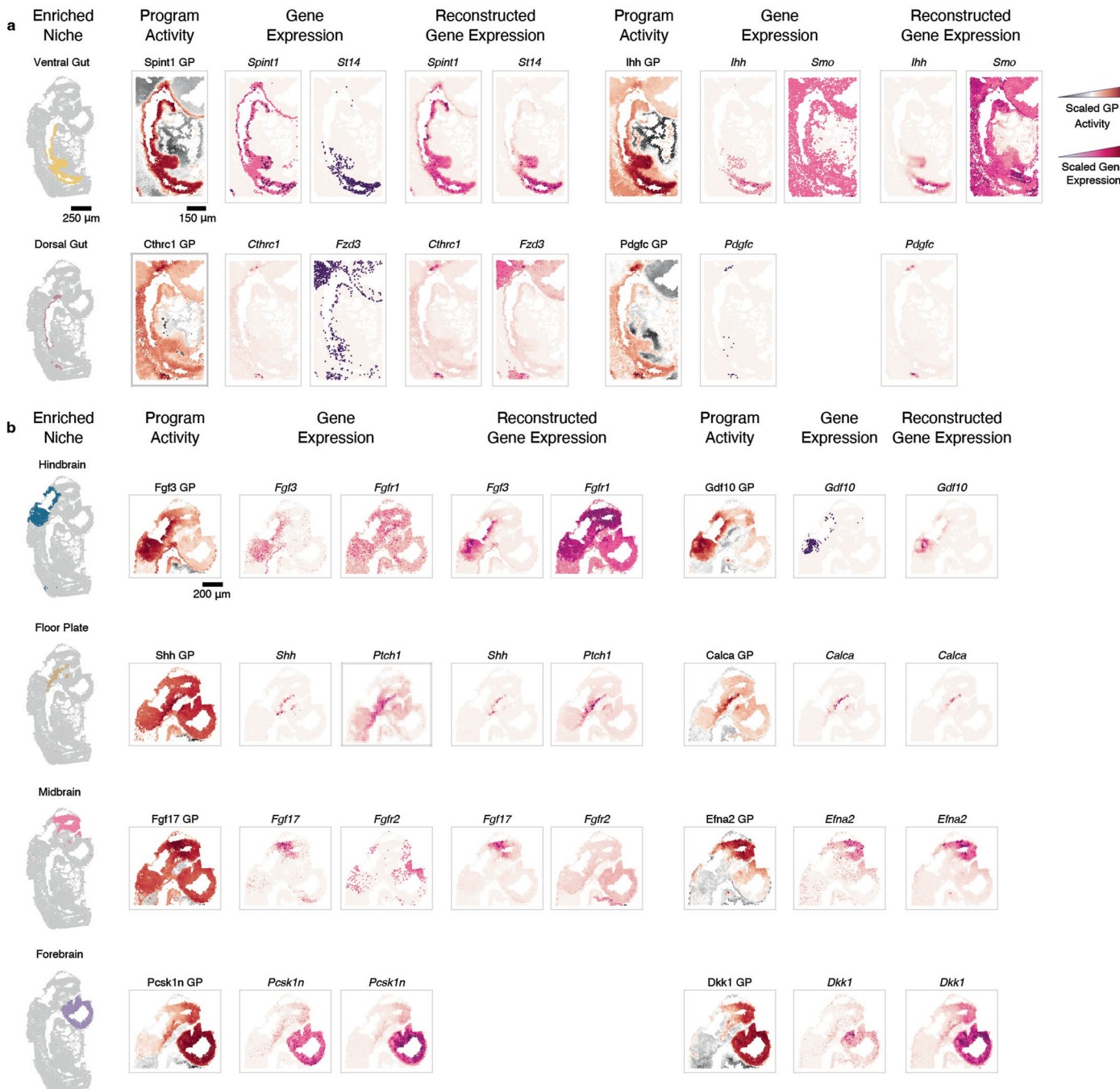

**Extended Data Fig. 1 | Enriched programs in gut and brain niches. a**, Programs enriched in gut niches show strong spatial correlation with the expression of their ligand- and receptor-encoding genes. Model-reconstructed gene expression closely matches the original while providing a smoothing effect.

**b**, Similarly, programs enriched in brain niches exhibit strong spatial correlation with the expression of ligand- and receptor-encoding genes. The reconstructed expression aligns closely with the original but is smoother. GP: gene program.

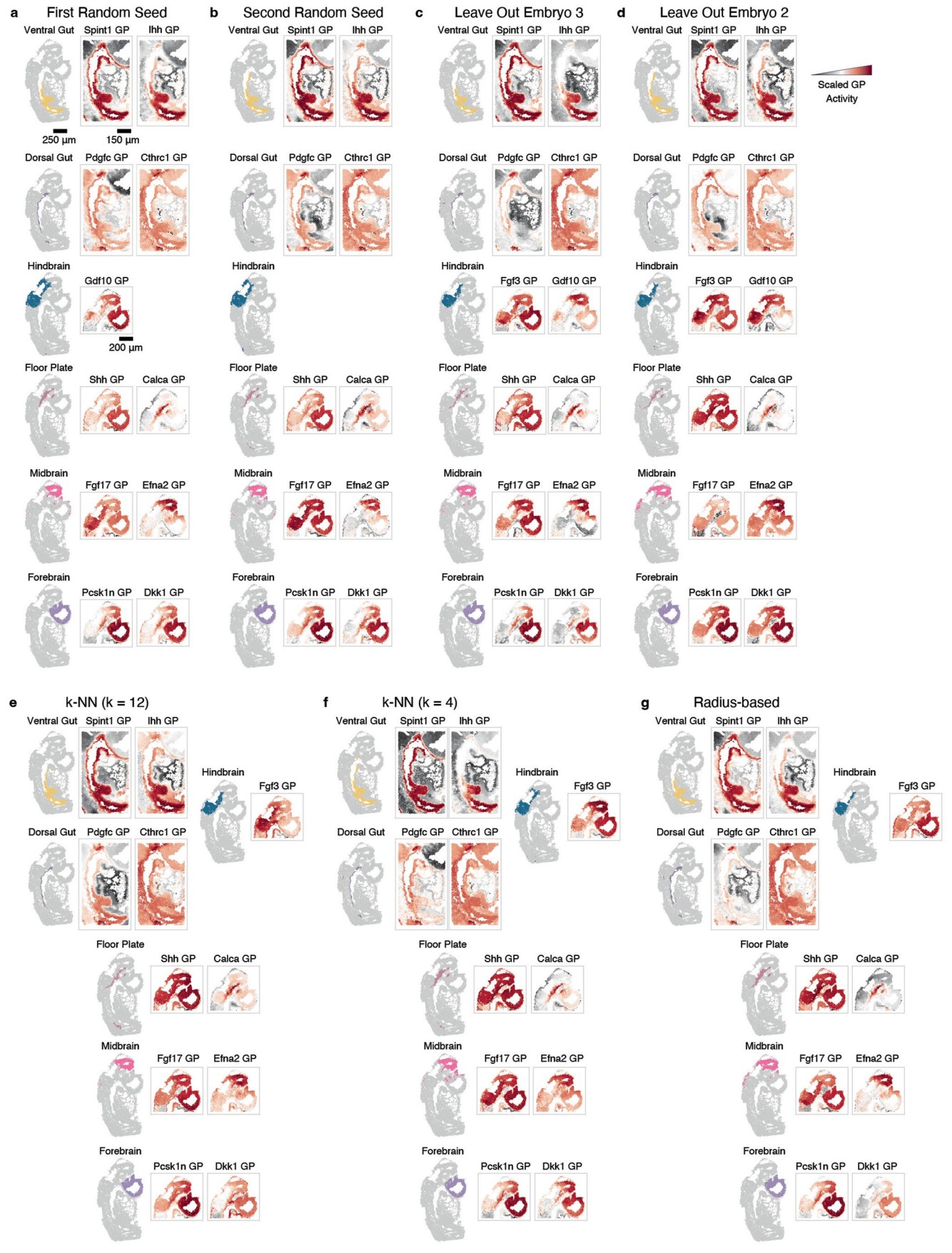

**Extended Data Fig. 2 | See next page for caption.**

**Extended Data Fig. 2 | Niche and program inference reproducibility, generalizability and robustness. a**, **b**, Embryo 2 niches and program activities inferred by NicheCompass with different random seeds. Displayed are characterizing programs from the main analysis in Fig. 2. Missing programs were filtered by program pruning in the respective model. Overall, there is good robustness of inferred niches and program activities across random seeds; however, there are also minor differences, most pronounced in the Hindbrain niche. **c**, Embryo 2 niches identified by NicheCompass when leaving out embryo 3 during reference model training. Next to it, the inferred program activity for the characterizing programs from the main analysis in Fig. 2. **d**, Same as **c** but when leaving out embryo 2 during reference model training. Overall,

there is high robustness of inferred niches and program activities providing evidence for generalizability. **e**, Embryo 2 niches and program activities inferred by NicheCompass with a longer-range k-NN graph (k = 12). Displayed are characterizing programs from the main analysis in Fig. 2. **f**, Same as **e** but with a shorter-range k-NN graph (k = 4). **g**, Embryo 2 niches and program activities inferred by NicheCompass with a radius-based neighborhood graph (average number of neighbors ~9). Missing programs were filtered by program pruning in the respective model. Overall, there is good robustness of inferred niches and program activities across neighborhood graphs; however, there are also minor differences, most pronounced in the Hindbrain niche. GP: gene program. k-NN: k-nearest neighbors.

**a** Ground Truth Niches

**b** Cell Types

**c** Injected GP Example

| GP | Prl2c2 GP |
|---|---|
| Niche | Niche 2 |
| Source Genes | *Prl2c2* |
| Target Genes | *Bcl6, Mmp1b, Ccn2* |
| Source Cell Type | Cell Type 2 |
| Target Cell Type | Cell Type 3 |
| Increment Factor | 8 |

*Prl2c2*  *Bcl6*  *Mmp1b*  *Ccn2*

Scaled Gene Expression

- Niche 1
- Niche 2
- Niche 3
- Niche 4
- Niche 5
- Niche 6
- Niche 7
- Niche 8

- Cell Type 1
- Cell Type 2
- Cell Type 3
- Cell Type 4
- Cell Type 5
- Cell Type 6
- Cell Type 7
- Cell Type 8

**d** (Predicted) Niches

NicheCompass Light · BANKSY · NicheCompass · GraphST · CellCharter · DeepLinc · STACI

**e** Metrics

| Model | CAS | MLAMI | CLISIS | GCS | NASW | CNMI | Overall Score |
|---|---|---|---|---|---|---|---|
| NicheCompass Light | 0.729 | 0.579 | 0.932 | 0.876 | 0.770 | 0.435 | 0.866 |
| BANKSY | 0.609 | 0.580 | 0.938 | 0.890 | 0.657 | 0.431 | 0.755 |
| NicheCompass | 0.173 | 0.588 | 0.906 | 0.858 | 0.736 | 0.415 | 0.708 |
| GraphST | 0.241 | 0.547 | 0.916 | 0.914 | 0.564 | 0.406 | 0.584 |
| STACI | 0.299 | 0.351 | 0.912 | 0.852 | 0.628 | 0.284 | 0.456 |
| CellCharter | 0.102 | 0.529 | 0.683 | 0.773 | 0.616 | 0.479 | 0.444 |
| DeepLinc | 0.148 | 0.446 | 0.891 | 0.796 | 0.519 | 0.262 | 0.314 |

| Model | NARI | NNMI | HOM | COM | Ground Truth Prediction Score |
|---|---|---|---|---|---|
| NicheCompass Light | 0.930 | 0.929 | 0.930 | 0.928 | 0.929 |
| BANKSY | 0.924 | 0.925 | 0.925 | 0.926 | 0.925 |
| NicheCompass | 0.891 | 0.903 | 0.902 | 0.903 | 0.900 |
| GraphST | 0.874 | 0.878 | 0.878 | 0.879 | 0.877 |
| CellCharter | 0.792 | 0.889 | 0.868 | 0.912 | 0.865 |
| DeepLinc | 0.725 | 0.727 | 0.723 | 0.730 | 0.726 |
| STACI | 0.368 | 0.533 | 0.511 | 0.556 | 0.492 |

**f**

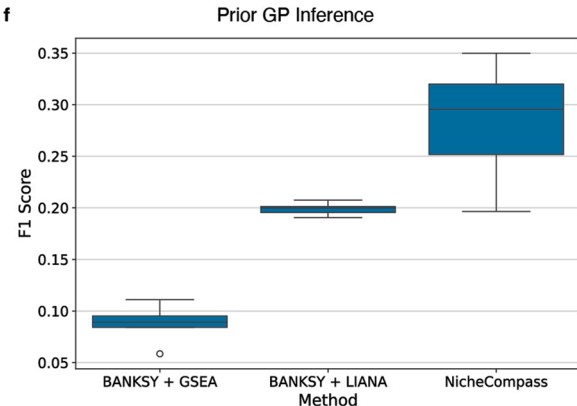

Prior GP Inference

F1 Score — Method: BANKSY + GSEA, BANKSY + LIANA, NicheCompass

**Extended Data Fig. 3 | See next page for caption.**

**Extended Data Fig. 3 | Data simulation. a**, The reference-based simulated tissue and a UMAP representation of the gene expression space reduced by principal component analysis, colored by ground truth niches. **b**, Same as **a** but colored by ground truth cell types. **c**, Example of an injected ground truth program which was upregulated in Niche 2 via an additive gene expression model. The target genes were upregulated in all Cell Type 3 cells if they had Cell Type 2 cells in their k neighborhood (with k = 6). Equally, the source genes were upregulated in Cell Type 2 cells. The increment factor determined the strength of upregulation. **d**, The reference-based simulated tissue colored by the predicted niches of each method. **e**, Metrics from the NicheCompass benchmarking suite (left) and metrics that measure the performance of the predicted niches compared to the ground truth (right). The overall score and ground truth prediction score are computed by min-max normalization and subsequent aggregation of the individual metrics. The ranking of methods is largely consistent between the two metrics suites. **f**, F1 scores between inferred and ground truth upregulated programs across n = 8 training runs for each workflow to infer niche-specific programs, with varying random seeds and a k-nearest neighbors graph with k = 6 (the ground truth cell interaction range). NicheCompass considerably outperforms alternative methods, providing evidence that it is useful to integrate pathways during training. GP: gene program.

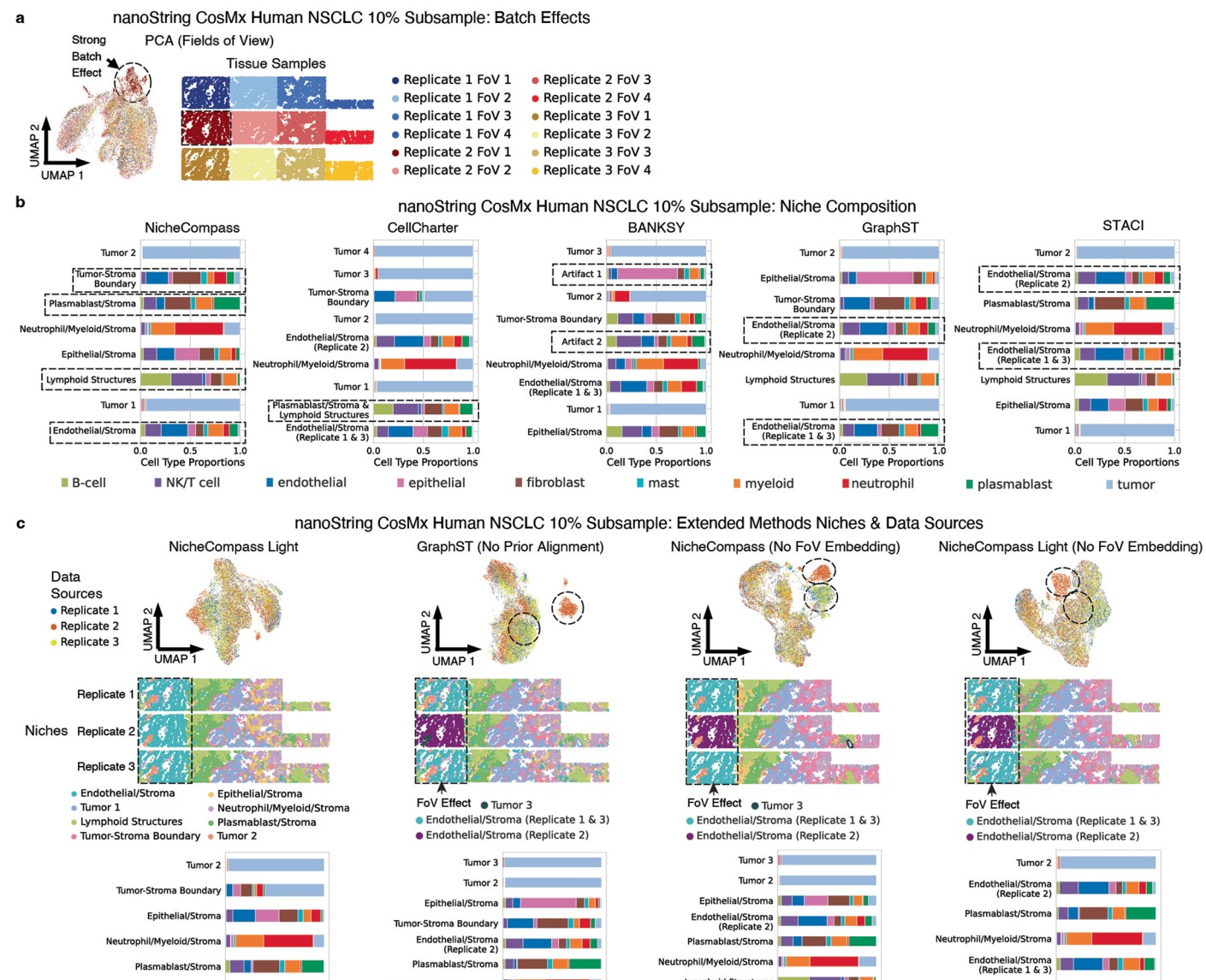

**Extended Data Fig. 4 | See next page for caption.**

**Extended Data Fig. 4 | Benchmarking on the nanoString CosMx human NSCLC 10% subsample. a**, UMAP representation after applying principal component analysis (PCA) to the raw gene expression of the three lung replicates[10], showing the presence of strong batch effects in the first field of view of the second replicate. **b**, Cell type composition of niches identified by each method. NicheCompass identified Lymphoid Structures and Tumor-Stroma Boundary niches and could differentiate between Stroma enriched by endothelial cells and Stroma enriched by plasmablast cells. CellCharter could not separate Plasmablast/Stroma from the Lymphoid Structures. BANKSY could not identify the Lymphoid Structures and Plasmablast/Stroma but instead identified artifact clusters. GraphST separated two Endothelial-enriched Stroma niches due to batch effects; however, these niches had very similar cell type composition, suggesting they should be unified. In addition, plasmablast cells were misallocated to one of those niches. STACI showed a similar failure

to unify the two Endothelial-enriched Stroma niches. **c**, Comparison of the integration performance of further method variants. Illustrated are the UMAP representations of the learned embedding spaces and the tissue, colored by annotated niches. Niches in the first field of view are highlighted, showing differences in batch effect removal capabilities. UMAP representations colored by data source further emphasize differences in batch effect removal for the first field of view. FoV: field of view. GraphST (No Prior Alignment) was trained without prior alignment through PASTE. **d**, Metrics for the training runs from **c** and Fig. 3d. The overall score is computed by aggregating min-max-normalized individual metrics into the two categories spatial consistency and niche coherence, followed by equal weighting of these categories. NicheCompass Light is a variation of our model that uses graph convolutional layers instead of dynamic graph attention layers. NSCLC: non-small cell lung cancer.

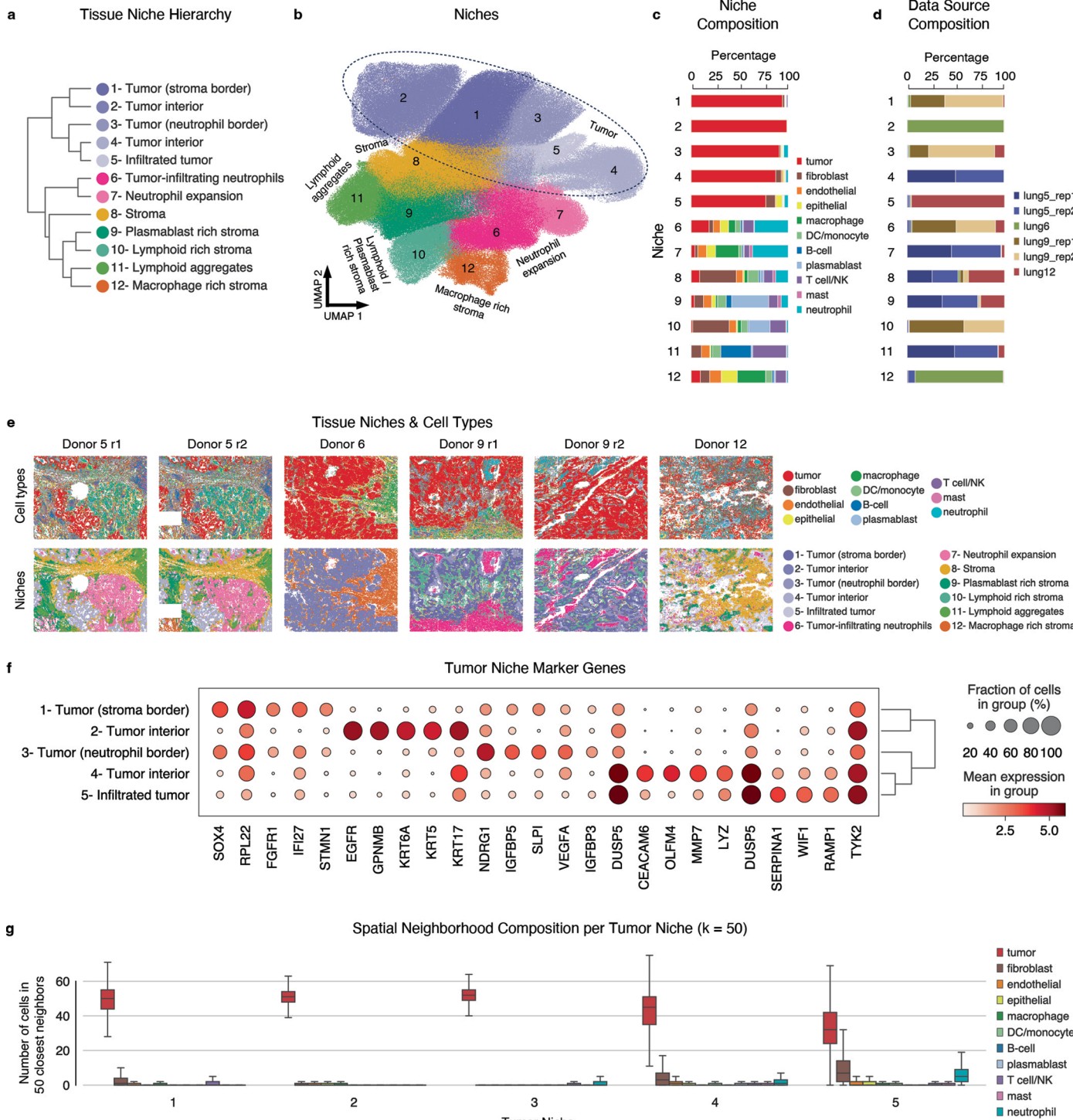

**Extended Data Fig. 5 | Analysis of inter-tumoral heterogeneity.**
**a**, A dendrogram computed based on average program activities, showing a hierarchy of niches. **b**, UMAP representation of the reference atlas, colored by niches identified with NicheCompass. **c**, **d**, Bar plots representing the cellular composition (**c**) and donor composition (**d**) of the identified niches. **e**, Spatial visualization of the six tissue sections included in the ref. 10, colored by cell type and identified niche. **f**, Dot plot showing the five most differential genes expressed in each tumor niche compared to the rest. The dot size represents the fractions of cells in a niche with expression higher than 0, while the dot color represents the mean expression level within expressing cells. **g**, Cell type composition in the spatial neighborhood of all cells in tumor niches 1 to 5 (niche 1: n = 81,577 cells, niche 2: n = 59,263 cells, niche 3: n = 38,937 cells, niche 4: n = 34,920 cells, niche 5: n = 10,820 cells), using a symmetric k-nearest neighbors graph with 25 neighbors. In this dataset, tumor niches consist of spatially segregated tumor cells, reflected by the identification of pure tumor niches where cells only have tumor cells in their spatial neighborhood.

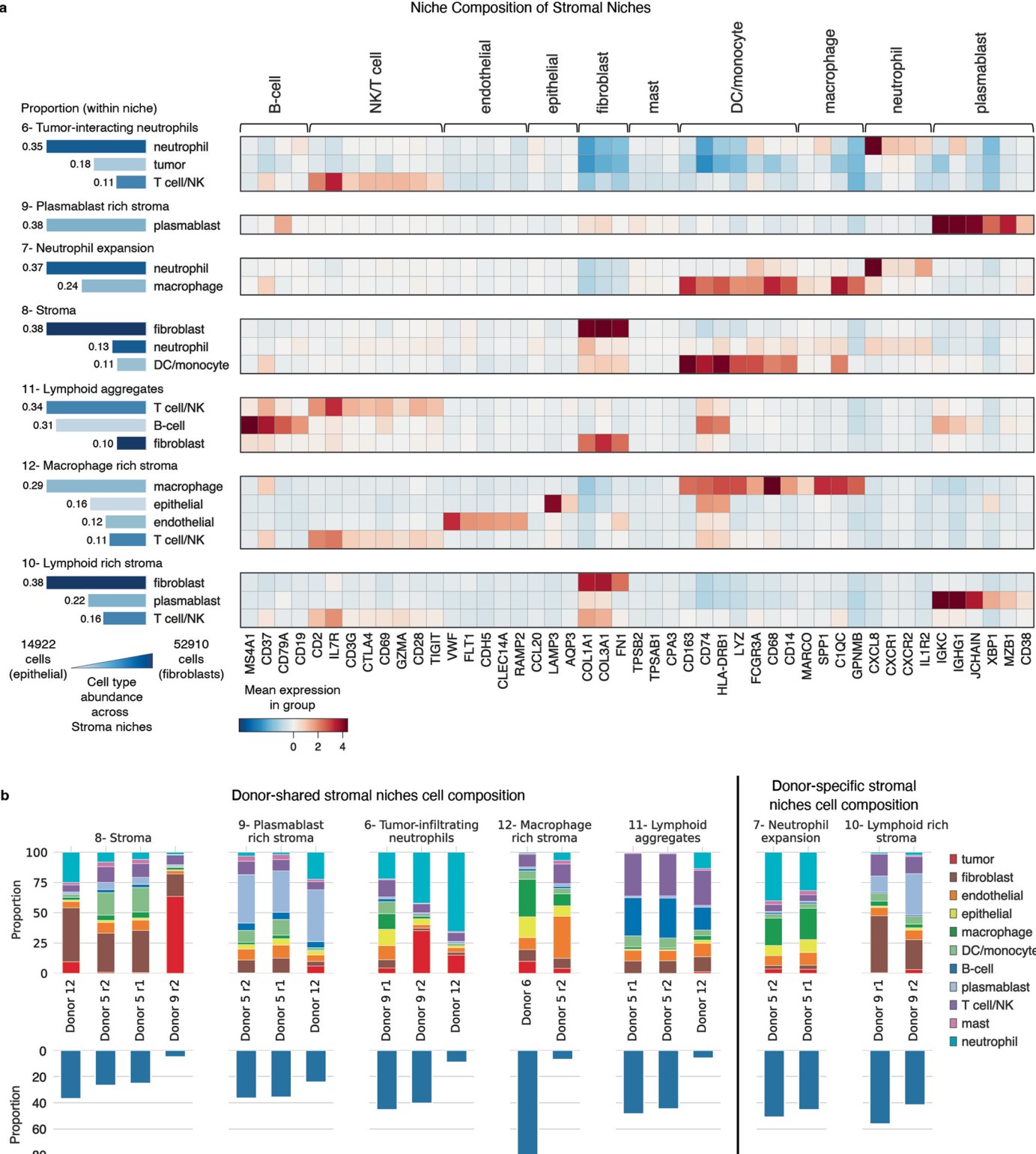

**Extended Data Fig. 6 | Characterization of stromal niches. a,** Each row represents a niche. The bar plots on the left represent cell proportions for the most abundant cell types in that niche (that is more than 10% of the cells in the niche). The length of the bars is proportional to the cell abundance within the niche and the color is proportional to the cell abundance across all 7 stroma niches (ranging from epithelial cells with 14,922 cells to fibroblasts with 52,910 cells). The heatmaps show mean expression of selected gene markers across cell types in each niche separately, with color representing mean gene expression.

Shown are selected marker genes per cell type that are differential in that cell type compared to the rest, considering all the niches together. Indicated at the top are the cell types represented by each set of markers. **b,** Niche cell type composition for all the samples where the niche is present (that is more than 5% of the cells in the niche are from that sample). Top bar plots show the cell type composition and bottom bar plots show the proportion of the cells from each niche in each of the samples.

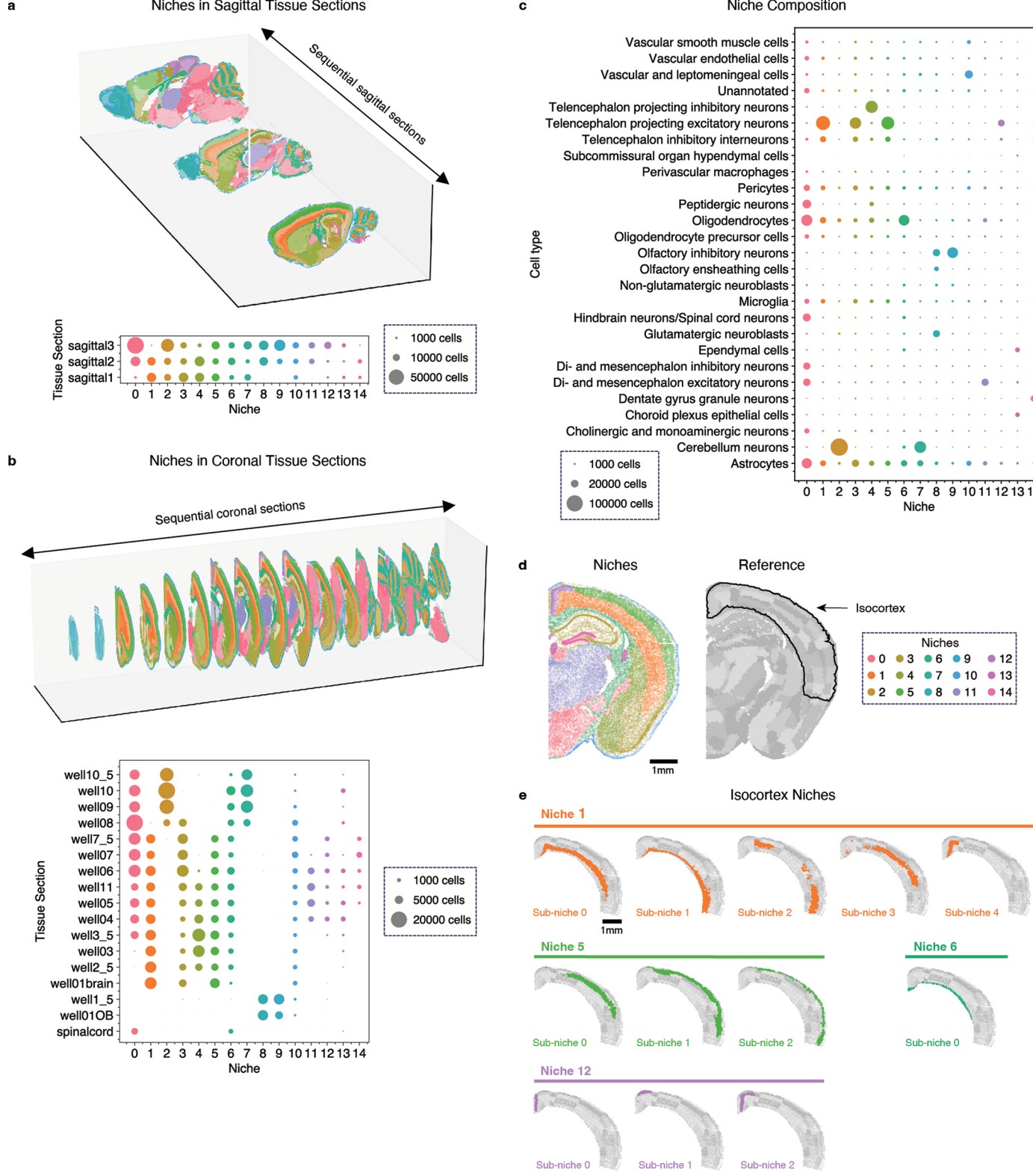

**Extended Data Fig. 7 | Niches identified in the mouse brain are consistent across sections and correspond to regions from a reference atlas. a,** Sagittal tissue sections[19] ordered by 3D position and colored by identified niches, showing consistency across sequential tissue sections. Below it the number of cells occurring in each tissue section for each niche. **b,** Same as **a** but for the coronal tissue sections (spinal cord is not shown). Cell numbers are scaled separately for coronal and sagittal tissue sections. **c,** Number of cells of different cell types in each niche. 10,683 of 1,091,280 cells are not assigned to a niche

and are not shown. **d,** Coronal section showing NicheCompass niches obtained through clustering of the embedding space (left) and regions from the Allen Brain Atlas (right). The isocortex is highlighted. **e,** Magnified view showing cells assigned to the isocortex, based on the Allen Brain Atlas annotations. Sub-niches with more than 250 cells annotated in this tissue section are shown. Sub-niches are obtained through clustering of cells in a niche and correspond with regions in the reference annotation.

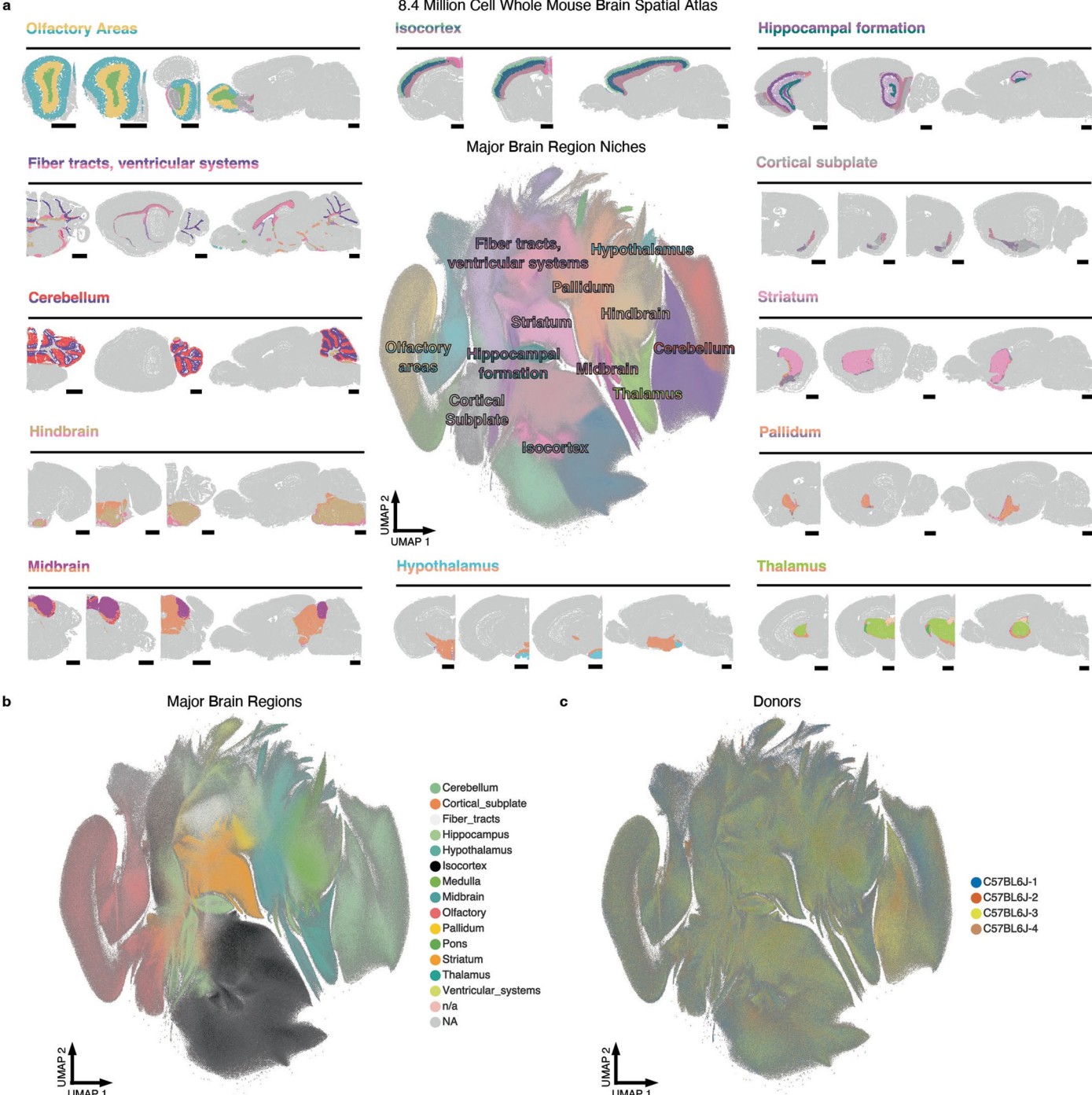

**Extended Data Fig. 8 | NicheCompass integrates 8.4 million cells across 239 tissue sections. a**, UMAP representation of the NicheCompass (Light) embedding space, colored by identified niches. Around it, randomly selected tissue slices[89] for each major brain region, colored by identified niches. Only cells belonging to the specific region are shown. Scale bars, 1 mm. **b**, **c**, UMAP representations colored by major brain regions (**b**) and donor mouse (**c**), showing successful integration of cells in matching brain regions across donors.

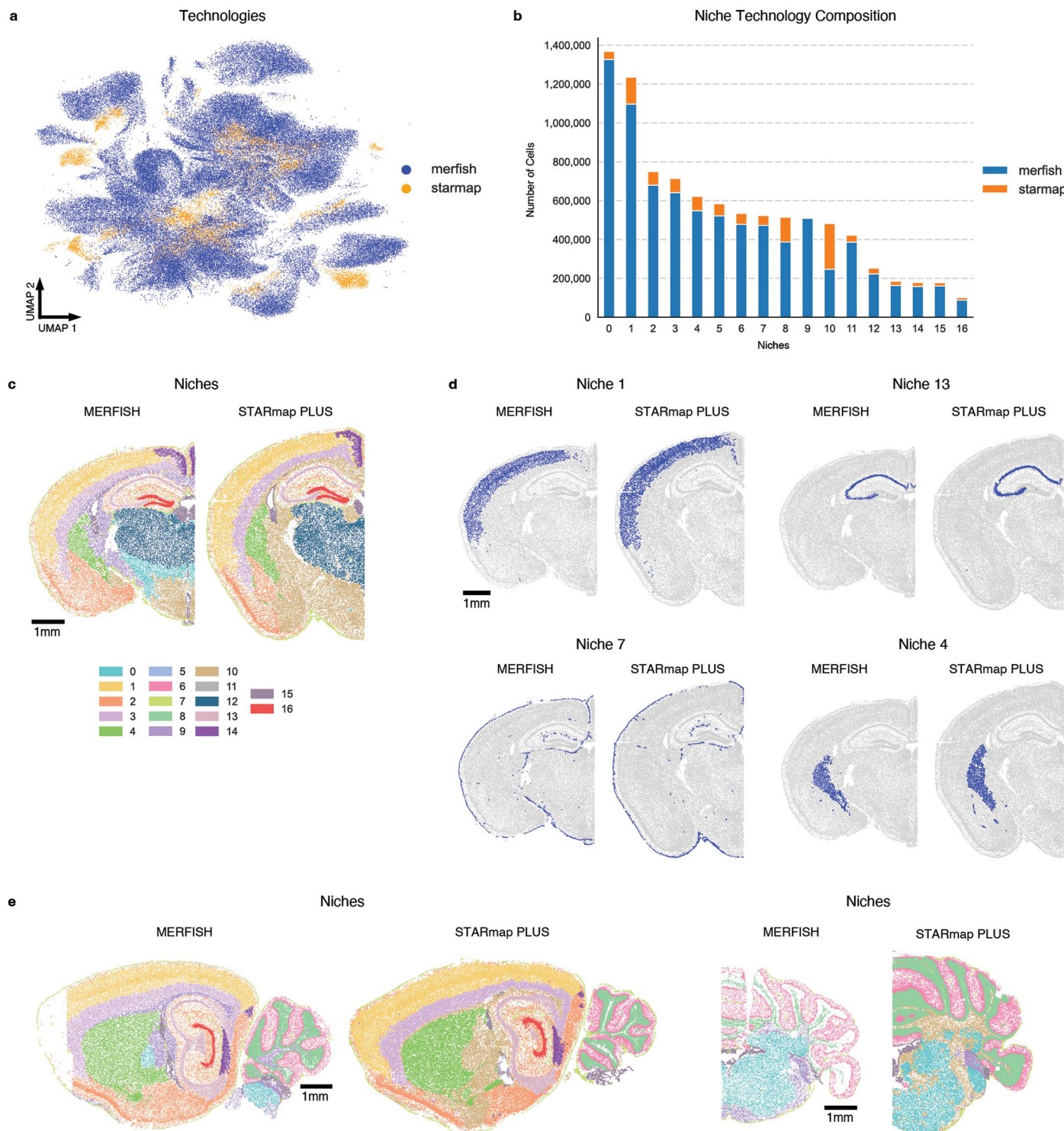

**Extended Data Fig. 9 | NicheCompass integrates samples across different spatial transcriptomics technologies. a**, UMAP representation of the NicheCompass embedding space after integrating the MERFISH mouse brain[89] and STARmap PLUS mouse CNS[19] datasets, colored by dataset/sequencing technology. **b**, Composition of niches in terms of cells from each of the two technologies, showing that all niches except niche 9 were present in both datasets. Only niches with more than 100,000 cells are displayed. **c**, Two example tissue slices of the same brain region, one from the MERFISH mouse brain dataset and the other from the STARmap PLUS mouse CNS dataset, highlighting consistent anatomical niches. **d**, Zoom in on four specific niches that emphasize the consistency in niche identification across technologies. **e**, Two additional pairs of tissue slices showing consistent NicheCompass niches across technologies.

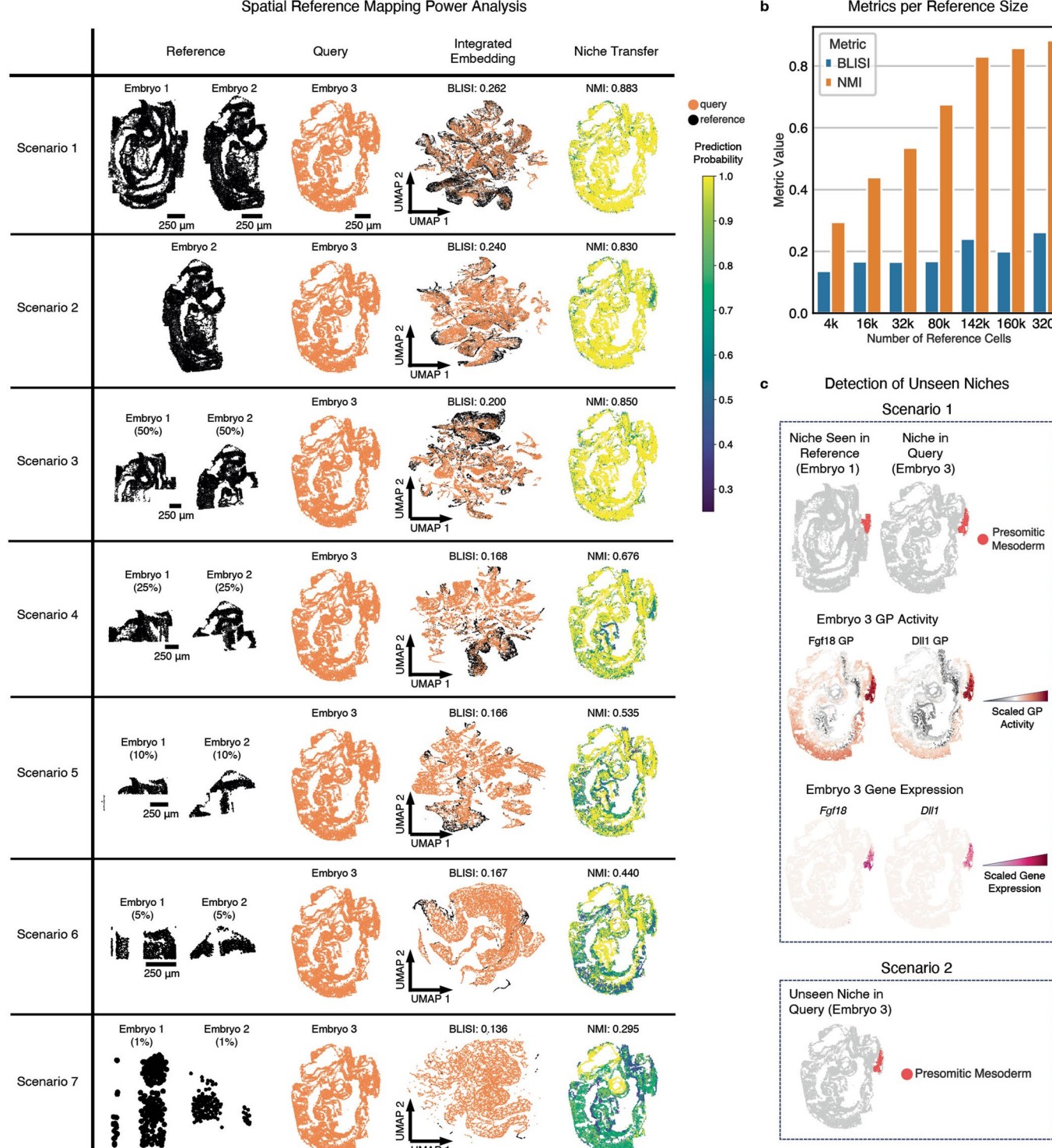

**Extended Data Fig. 10 | seqFISH mouse organogenesis spatial reference mapping. a**, Power analysis using different dataset proportions of the mouse embryos 1 and 2 as reference while holding out embryo 3 as query. Embryo 3 is mapped onto the reference using weight-restricted fine-tuning. UMAPs represent the integrated embedding space. BLISI quantifies the integration performance. Label transfer from reference to query is performed via a k-nearest neighbors (k-NN) classifier trained on the reference. The prediction probability of this k-NN classifier quantifies uncertainty in niche label transfer. NMI quantifies niche prediction performance based on niche labels from the full analysis in Fig. 3. **b**, Metrics from the scenarios in **a** per number of cells in the reference. NMI significantly reduces at a size of ~80,000 reference cells. **c**, Comparison of niche detection of the Presomitic Mesoderm niche in scenarios 1 and 2. In scenario 1, this niche is seen in the reference, and we recover the same characterizing programs as in the analysis on the full dataset, supported by expression of the respective ligand-encoding genes. In scenario 2, this niche is not seen in the reference, yet it is detected as a novel niche; however, the same programs could not be recovered as these were not relevant during reference training. GP: gene program.

# Reporting Summary

## Statistics

For all statistical analyses, confirm that the following items are present in the figure legend, table legend, main text, or Methods section.

| n/a | Confirmed | |
|---|---|---|
| ☐ | ☒ | The exact sample size (*n*) for each experimental group/condition, given as a discrete number and unit of measurement |
| ☐ | ☒ | A statement on whether measurements were taken from distinct samples or whether the same sample was measured repeatedly |
| ☐ | ☒ | The statistical test(s) used AND whether they are one- or two-sided *Only common tests should be described solely by name; describe more complex techniques in the Methods section.* |
| ☐ | ☒ | A description of all covariates tested |
| ☐ | ☒ | A description of any assumptions or corrections, such as tests of normality and adjustment for multiple comparisons |
| ☐ | ☒ | A full description of the statistical parameters including central tendency (e.g. means) or other basic estimates (e.g. regression coefficient) AND variation (e.g. standard deviation) or associated estimates of uncertainty (e.g. confidence intervals) |
| ☐ | ☒ | For null hypothesis testing, the test statistic (e.g. *F*, *t*, *r*) with confidence intervals, effect sizes, degrees of freedom and *P* value noted *Give P values as exact values whenever suitable.* |
| ☐ | ☒ | For Bayesian analysis, information on the choice of priors and Markov chain Monte Carlo settings |
| ☒ | ☐ | For hierarchical and complex designs, identification of the appropriate level for tests and full reporting of outcomes |
| ☒ | ☐ | Estimates of effect sizes (e.g. Cohen's *d*, Pearson's *r*), indicating how they were calculated |

*Our web collection on statistics for biologists contains articles on many of the points above.*

## Software and code

Policy information about availability of computer code

| | |
|---|---|
| Data collection | To collect default ligand-receptor and transcriptional regulation programs, we used omnipath (version 1.0.8) and decoupler (version 1.8.0) with Python version 3.9.19, respectively. Default metabolite-sensor programs were retrieved from https://github.com/zhengrongbin/MEBOCOST (on 18.05.2023). Default combined interaction programs were constructed using NicheNet's regulatory potential matrix, retrieved from https://zenodo.org/record/7074291. No spatial omics data was collected as part of this study (all data was previously published). R source data files were collected and converted to AnnData objects using R (version 4.4.0) with the following libraries: Bioconductor (version 3.20), rhdf5 (version 2.50.1), scRNAseq (version 2.20.0), Seurat (version 5.1.0), SeuratData (version 0.2.2.9001), SeuratDisk (version 0.0.0.9021), and zellkonverter (version 1.16.0). Package versions of our R and Python environments are also available at https://github.com/Lotfollahi-lab/nichecompass-reproducibility. |
| Data analysis | All experiments were conducted with Python version 3.9.19 and R version 4.4.0 unless otherwise specified. All experiments were performed with NicheCompass version 0.1.2 (except ablation study experiments which were performed with NicheCompass version 0.2.0). Other major Python software that we used for data analysis included: altair (5.5.0), anndata (version 0.10.8), GraphST (version 1.1.1), matplotlib (version 3.8.4), liana (version 1.3.0), networkx (version 3.2.1), numpy (1.26.4), pandas (2.2.3), plotly (version 5.24.1), plottable (version 0.1.5), pyensembl (2.3.13), pynrrd (version 1.0.0), pywaffle (version 0.0.8), scanpy (version 1.9.8), scarches (version 0.5.9), scib-metrics (version 0.5.1), scikit-learn (version 1.6.1), scipy (1.12.0), seaborn (version 0.13.2), skimage (version 0.24.0), squidpy (version 1.6.1), tiledb (version 0.20.0), tiledbsoma (version 0.1.22), torch (version 2.0.0), torch_geometric (version 2.5.3). We copied the BANKSY source code from https://github.com/prabhakarlab/Banksy (12.07.2024). We copied the STACI source code from https://github.com/uhlerlab/STACI/blob/master (23.11.2023). The data simulation was performed with the following R libraries: srtSIM (version 0.99.6), readr (version 2.1.5), data.table (version 1.16.4), dplyr (version 1.1.4), SingleCellExperiment (version 1.28.1), and zellkonverter (version 1.16.0). Package versions of our R and Python environments are also available at https://github.com/Lotfollahi-lab/nichecompass-reproducibility. DeepLinc was run with Python version 3.7.0 and its dependencies are available in a separate yaml file at https://github.com/Lotfollahi-lab/nichecompass-reproducibility. |

CellCharter (version 0.3.2) and scvi -tools (version 0.20.3) were run with Python version 3.10.0 and dependencies that are available in a separate yaml file at https://github.com/Lotfollahi-lab/nichecompass-reproducibility. STalign (version 1.0.1) was run with Python version 3.9.19 and its dependencies are available in a separate yaml file at https://github.com/Lotfollahi-lab/nichecompass-reproducibility.

For manuscripts utilizing custom algorithms or software that are central to the research but not yet described in published literature, software must be made available to editors and reviewers. We strongly encourage code deposition in a community repository (e.g. GitHub). See the Nature Portfolio guidelines for submitting code & software for further information.

## Data

Policy information about availability of data

All manuscripts must include a data availability statement. This statement should provide the following information, where applicable:
- Accession codes, unique identifiers, or web links for publicly available datasets
- A description of any restrictions on data availability
- For clinical datasets or third party data, please ensure that the statement adheres to our policy

All datasets used in this study were previously published. Processed versions are available as AnnData objects for download as outlined at https://github.com/Lotfollahi-lab/nichecompass-reproducibility. The seqFISH mouse organogenesis dataset was sourced from https://marionilab.cruk.cam.ac.uk/SpatialMouseAtlas/. The SlideSeqV2 dataset was obtained via squidpy.datasets.slideseqv2(). The MERFISH mouse liver dataset was retrieved from https://info.vizgen.com/mouse-liver-access (animal 1, replicate 1). The nanoString CosMx NSCLC dataset was collected from https://nanostring.com/products/cosmx-spatial-molecular-imager/ffpe-dataset/nsclc-ffpe-dataset/. The Xenium human breast cancer dataset was downloaded from https://www.10xgenomics.com/products/xenium-in-situ/preview-dataset-human-breast. The STARmap PLUS mouse CNS dataset was obtained from https://zenodo.org/records/8327576. The MERFISH whole mouse brain dataset was retrieved from https://cellxgene.cziscience.com/collections/0cca8620-8dee-45d0-aef5-23f032a5cf09. The Spatial ATAC-RNA seq mouse brain dataset (postnatal day 22) was collected from https://www.ncbi.nlm.nih.gov/geo/query/acc.cgi?acc=GSE205055 (gene expression counts and spatial coordinates) and https://brain-spatial-omics.cells.ucsc.edu/ (peak counts and cell type labels). Lastly, the Stereo-seq mouse embryo dataset was downloaded from http://sdmbench.drai.cn/ (Data ID 13).

## Research involving human participants, their data, or biological material

Policy information about studies with human participants or human data. See also policy information about sex, gender (identity/presentation), and sexual orientation and race, ethnicity and racism.

| | |
|---|---|
| Reporting on sex and gender | No human data were collected as part of the study. |
| Reporting on race, ethnicity, or other socially relevant groupings | No human data were collected as part of the study. |
| Population characteristics | No human data were collected as part of the study. |
| Recruitment | No human data were collected as part of the study. |
| Ethics oversight | No human data were collected as part of the study. |

Note that full information on the approval of the study protocol must also be provided in the manuscript.

# Field-specific reporting

Please select the one below that is the best fit for your research. If you are not sure, read the appropriate sections before making your selection.

☒ Life sciences ☐ Behavioural & social sciences ☐ Ecological, evolutionary & environmental sciences

For a reference copy of the document with all sections, see nature.com/documents/nr-reporting-summary-flat.pdf

# Life sciences study design

All studies must disclose on these points even when the disclosure is negative.

| | |
|---|---|
| Sample size | No statistical method was used to predetermine sample size, and no data were excluded from the analyses unless explicitly stated. The seqFISH mouse organogenesis dataset included 57,536 cells across six sagittal tissue sections from three 8-12 somite stage mouse embryos: 19,451 (embryo 1), 14,891 (embryo 2), and 23,194 (embryo 3). The SlideSeqV2 mouse hippocampus dataset included 41,786 observations at near-cellular resolution. The MERFISH mouse liver dataset included 367,335 cells. The nanoString CosMx human NSCLC dataset included 800,327 cells across 8 tissue sections from 5 donors. Cell counts per section are: 93,206 cells (donor 1, replicate 1), 93,206 cells (donor 1, replicate 2), 91,691 cells (donor 1, replicate 3), 91,691 cells (donor 2), 77,391 cells (donor 3, replicate 1), 115,676 (donor 3, replicate 2), 66,489 cells (donor 4) and 76,536 cells (donor 5). The Xenium human breast cancer dataset included 282,363 cells across two replicates (replicate 1: 164,000, replicate 2: 118,363). The STARmap PLUS mouse CNS dataset included 1,091,527 million cells. For ablation studies, only the first sagittal tissue section was used (91,246 cells). The MERFISH whole mouse brain dataset included ~8.4 million cells across 239 sections from four animals. The Spatial ATAC-RNA seq mouse brain dataset included 9,215 spot-level observations. The Stereo-seq mouse embryo dataset included 5,913 spot-level observations. |

| | |
|---|---|
| Data exclusions | In the seqFISH mouse organogenesis dataset, we filtered cells annotated as low quality by the original authors. In the nanoString CosMx human NSCLC dataset, we filtered cells with < 50 counts, which were assessed to be low quality cells after QC. |
| Replication | For method benchmarking, we performed n = 8 training runs for each method. For ablation studies, we performed n = 8 training runs for each model configuration. For data analysis, we trained a single NicheCompass model per dataset unless otherwise specified. In the seqFISH mouse organogenesis dataset, we evaluated robustness and reproducibility of analysis results by repeating training runs across n = 3 seeds and n = 4 neighborhood graphs, confirming that results are robust and reproducible. |
| Randomization | To generate simulated data, we randomly sampled (1) niches to upregulate programs, (2) increment parameters with which programs were upregulated, (3) source and target cell types of programs, and (4) prior programs and (5) program member genes to be upregulated. Different random seeds were used for different runs during benchmarking. |
| Blinding | Blinding was not applicable to this study because no sample group allocation was performed. |

# Reporting for specific materials, systems and methods

We require information from authors about some types of materials, experimental systems and methods used in many studies. Here, indicate whether each material, system or method listed is relevant to your study. If you are not sure if a list item applies to your research, read the appropriate section before selecting a response.

### Materials & experimental systems

| n/a | Involved in the study |
|---|---|
| ☒ | Antibodies |
| ☒ | Eukaryotic cell lines |
| ☒ | Palaeontology and archaeology |
| ☒ | Animals and other organisms |
| ☒ | Clinical data |
| ☒ | Dual use research of concern |
| ☒ | Plants |

### Methods

| n/a | Involved in the study |
|---|---|
| ☒ | ChIP-seq |
| ☒ | Flow cytometry |
| ☒ | MRI-based neuroimaging |

## Plants

| | |
|---|---|
| Seed stocks | No plant data were collected or analyzed as part of this study. |
| Novel plant genotypes | No plant data were collected or analyzed as part of this study. |
| Authentication | No plant data were collected or analyzed as part of this study. |

