## [Peer Review File · Nature Genetics]

Quantitative characterization of cell niches in spatially-resolved omics data

Corresponding Author: Dr Mohammad Lotfollahi

Version 0:

Decision Letter:

22nd May 2024

Dear Mo,

Your Technical Report, "Quantitative characterization of cell niches in spatial atlases" has now been seen by 3 referees. You will see from their comments copied below that while they find your work of considerable potential interest, they have raised quite substantial concerns that must be addressed. In light of these comments, we cannot accept the manuscript for publication, but would be very interested in considering a revised version that addresses these serious concerns.

In brief, the three reviews overall span the full range of support of publication, but they also note a range of important issues that require resolution.

Referee #1 is skeptical about the overall advance, and also has questions over the conceptual clarity of what NicheCompass is actually analysing (cell types, or niches?), as well as suggesting the benchmarking requires improvement, including simulations. They do, however, provide substantial, specific and constructive guidance to address their concerns. Reviewer #2 is positive, suggesting NicheCompass offers something new and useful for the field. They do, however, have specific requests that overlap with other reports (e.g. on hyperparameter selection, benchmarking, de novo gene program analysis, etc.).

Referee #3 falls in between: they appreciate the approach but sound unconvinced that NicheCompass offers a broadly-useful approach over existing methods, and they also highlight some biological results that do not align with past knowledge. Again, guidance for improvement is supplied.

In our reading, these are useful reports that provide specific instruction to improve this work in revision. In our reading these requests are, by and large, reasonable and doable and we hope you and your co-authors would be motivated to address them in full. As well as the various overlapping concerns highlighted in the reviews, we particularly agree with Referee #1's comment on simulations, and how doing so would help clarify various other issues raised in these reports.

We hope you will find the referees' comments useful as you decide how to proceed. If you wish to submit a substantially revised manuscript, please bear in mind that we will be reluctant to approach the referees again in the absence of major revisions.

To guide the scope of the revisions, the editors discuss the referee reports in detail within the team, including with the chief editor, with a view to identifying key priorities that should be addressed in revision and sometimes overruling referee requests that are deemed beyond the scope of the current study. We hope that you will find the prioritised set of referee points to be useful when revising your study. Please do not hesitate to get in touch if you would like to discuss these issues further.

If you choose to revise your manuscript taking into account all reviewer and editor comments, please highlight all changes in the manuscript text file. At this stage we will need you to upload a copy of the manuscript in MS Word .docx or similar editable format.

*2) If you have not done so already please begin to revise your manuscript so that it conforms to our Technical Report format instructions, available here. Refer also to any guidelines provided in this letter.

*3) Include a revised version of any required Reporting Summary: <https://www.nature.com/documents/nr-reporting-summary.pdf>

Please be aware of our guidelines on digital image standards.

Link Redacted

If you wish to submit a suitably revised manuscript we would hope to receive it within 6 months. If you cannot send it within this time, please let us know. We will be happy to consider your revision so long as nothing similar has been accepted for publication at Nature Genetics or published elsewhere. Should your manuscript be substantially delayed without notifying us in advance and your article is eventually published, the received date would be that of the revised, not the original, version.

Nature Genetics is committed to improving transparency in authorship. As part of our efforts in this direction, we are now requesting that all authors identified as 'corresponding author' on published papers create and link their Open Researcher and Contributor Identifier (ORCID) with their account on the Manuscript Tracking System (MTS), prior to acceptance. ORCID helps the scientific community achieve unambiguous attribution of all scholarly contributions. You can create and link your ORCID from the home page of the MTS by clicking on 'Modify my Springer Nature account'. For more information please visit please visit www.springernature.com/orcid.

Thank you for the opportunity to review your work.

Sincerely,

Michael Fletcher, PhD
Senior Editor, Nature Genetics
ORCID: 0000-0003-1589-7087

Referee expertise:

Referee #1: spatial omics methods development; ML.

Referee #2: spatial omics methods development.

Referee #3: spatial omics, neuroscience.

Reviewers' Comments:

Reviewer #1:

Remarks to the Author:

The manuscript introduces NicheCompass, a graph deep learning method designed to quantitatively characterize tissue niches by integrating spatial omics data with gene programs. The model incorporates both molecular and spatial information to identify cell niches and their underlying signaling events. The authors applied NicheCompass to several biological contexts, including mouse organogenesis and human cancer, showcasing its ability to uncover gene programs that may drive niche identity. However, while the method seems overall quite reasonable and makes sense, the manuscript has a few major issues, including a lack of comparison with other approaches, a lack of quantitative simulation evaluation, and the confusing use of some of key concepts and assumptions throughout the work. My comments are listed below.

== Major concerns on the methodology / approach ==

1. The NicheCompass model is tailored for biological processes by including gene programs, and the neighbor/self dissection makes sense for spatial omics. The graph decoder approach seems like a meaningful way to ensure the cell embeddings include spatial information. However, even though the authors used a few datasets to assess NicheCompass, the overall evaluation is limited to the specific contexts and the analysis often feels lengthy and ad hoc. One major deficiency of this work is the lack of a well-designed simulation study to quantify the impact of different designs in the models and provide guidance on parameter selection. This is certainly an issue for the current state of spatial transcriptome and single-cell computational methods papers. However, a leading journal like Nature Genetics should set a high benchmark standard for the field.

2. The NicheCompass model uses a set of hyperparameters to balance the factors (Methods), along with many others. How to set these parameters seems vague and confusing. For the entire loss function in Equation (18) of the Methods section, how to do the hyper-parameter selection for the weighting factors of each loss component is unclear. This is related to the previous comment on the lack of well-designed simulations in the work to carefully assess the model parameters and their influence on the analysis results.

3. Conceptually, the authors did not clearly articulate the novelty of the niche that NicheCompass is focused on. Throughout the paper, it seems like the authors often conflate cell type and niche in their analysis. Do the metrics for scoring different methods actually make sense in the context of niche identification? Specifically:

- CAS was designed to work on cell types. Do we expect that it should do well on niches that were identified by each method?
- For CLISI, are the cell type annotations for the calculation coming from the original paper?
- GCS is measuring how close the latent embeddings are of physically-proximal cells.

Since the niche embedding for each cell is incorporating its neighbors within the embedding, these metrics are all likely to score NicheCompass higher than competitors that don't rely as highly on neighbor information. This is unfair and somewhat misleading. At the bare minimum, the scores should be normalized, since the improvement of NicheCompass over prior methods is listed as a percent improvement over the next-best tool over these metrics, which is not very meaningful, since they are on different scales. Indeed, Fig. 4a shows an instance where the embedding from NicheCompass would likely not score well on the supervised metrics for spatial conservation, since the niches captured in the breast cancer dataset are composed of multiple different cell types.

4. The authors claim that the NicheCompass is novel in its ability to "quantitatively characterize tissue niches". While the method covers many use cases and multiple applications are demonstrated in the Results section, it is unclear what is the quantitative characterization of niches. Is it the niches (spatial domains) recognized by clustering or cell encoding with spatial information integrated? This is quite confusing and is not well defined.

5. NicheCompass focuses on showing the importance of prior information in identifying cellular niches. But how important is that and whether prior information leads to bias toward known processes heavily? I think the design of identifying potential de novo gene programs is a bit awkward in NicheCompass as it requires users to define the number. But more importantly, how are de novo gene programs analyzed for biological significance? Is it easy to differentiate between de novo gene programs that are biologically meaningful and those that are due to simply noise within the data? For example, for the breast cancer dataset, it seems like there are only 5 out of at least 86 de novo gene programs that seem to encode biological knowledge.

6. In Line 152 of the Methods section, the graph decoder module is used to reconstruct the entire adjacency matrix A through computing the cosine similarities of pairwise latent feature vectors. However, the adjacency matrix A is extracted from a disconnected graph composed of all sample-specific symmetric k -nearest neighbor subgraphs. I have doubts about encouraging recovering the adjacent elements between samples to be 0, since there may be spatial relationships between cells from different samples (e.g., adjacent slices in a spatial atlas of the whole mouse brain).

7. The work lacks comparisons with some of the more recent niche embedding learning methods such as COVET (Haviv et al. 2024 Nat Biotech) and BANKSY (Singhal et al. 2024 Nat Genet). For example, in Line 342, the author mentioned that only NicheCompass could identify four separable cortical layer niches. It would be important to see whether other methods can identify the cortical layer by trying different cluster numbers.

== Other major concerns on the data analysis ==

8. In Fig. 3a, it seems that the assignment of the identified CA1 niche is unreasonable since it is connected with the identified V3 niche. However, in fact, from the Allen reference atlas (in Fig. 3a), these two niches are separated from each other, and other competitor methods successfully separate these two niches.

9. In Figure 3c, it seems like the intent is to show that other methods do not create clusters in the isocortex that are close together in their embedding space. This comparison seems to be done in the UMAP space, which is not quite meaningful. It is possible that given different UMAP parameters, the other methods would place the isocortex clusters all closer together. This analysis should be done in the original latent spaces, using the distance between cluster centers.

10. In Figure 6, there is a comparison of the original clusters from the spatial ATAC-RNA dataset paper to the niches from NicheCompass. This comparison relies on the original clustering from the paper, which identified a specific number of clusters based on the parameters given to the clustering method. It would be much more meaningful to compare the actual underlying representations that were given to the clustering algorithm to compare the ability to identify these substructures without the influence of the clustering algorithm. It is possible that under different parameters, the original data was sufficient to separate the brain regions. Looking at the UMAP from the original paper, it does seem like the Corpus Callosum and Anterior Commissure can be easily separated and correctly identified.

11. It is reasonable that highly heterogeneous tumor cells form their own niches (e.g., Fig 5a), but when the goal is to understand the microenvironment and immune cell infiltration, a user may be interested in tolerating more diverse tumor cells in a niche. How will NicheCompass deal with this?

12. Line 527 (Fig. 5): A reason donor 6 (see Table S6 in the original publication) has very different niches might be because it is squamous cell carcinoma (all others are adenocarcinoma), so their surroundings (i.e. the sample collected) may be different in the first place. This is unclear.

13. Supp. Note 4: It appears to me that while NicheCompass scales to a large number of cells, more genes significantly impact the running time (note the log-scale y-axis in Supp. Fig. 17). Why is this the case?

14. The manuscript overall is quite dense and can be significantly shortened to highlight the most important novelty. The figures are not made in high quality and are often difficult to read.

- In Line 122 of the Methods section, I found that the latent vector z_i , which should be the output of the graph encoder module, is not introduced in the graph encoder module part. I guess the latent vector z_i is just the variational posterior mean vector μ_i .

- The y-axis labels for Fig 3f are somewhat confusing.

- Fig 4j isn't clearly discussed in the main text. In general Fig 4f-j is a large number of panels and it would be helpful to label which sentence each of them corresponds to.

Reviewer #2:

Remarks to the Author:

NicheCompass is a novel and promising tool for spatial niche mapping based on gene programs (GP), particularly GPs for cell-cell interactions. The biological assumption indicated in the manuscript is that nearby cells that express enriched gene programs for cell-cell interactions likely form a niche. This is a valid assumption.

The paper presents a well-developed variational graph neural network model with several innovative components, including the multi-module decoder for preserving spatial neighborhood information, neighbor sampling for memory-efficient training, and weight-restricted fine-tuning for querying unseen data. NicheCompass models confounding factors and batch effects by including covariates vectors. This allows for the analysis of multiple samples and this is a very important feature that is needed in the field. The method has been thoroughly evaluated across multiple biological systems, technologies, and sample sizes, demonstrating its potential for diverse and broad applications. The manuscript is well-written and the technical information and formula are clearly described.

The detailed software documentation and tutorials, the well-organized code (including code to reproduce figures) are complemented. NicheCompass leverages functionalities in common software platforms such as AnnData, scanpy and squidpy and would be implementable by many users. This reviewer has installed the software and run through the key functionalities smoothly.

Major comments:

- As the paper emphasizes cell-cell interactions as a key input and focus of NicheCompass, the authors should discuss the limitations of using panel-based technologies which can only measure a subset of cell-cell interaction genes, missing many in the ligand-receptor gene databases.

- How would NicheCompass perform differently when applying for two types of data with different resolutions and number of genes. The first type is with hundreds of thousands of cells like Xenium and CosMX but with fewer genes. The second data type has few spots/grids, but more genes (e.g. Visium, DBiT).

- The authors should provide more detailed information on the model training process and its sensitivity to data size and composition. The authors should clarify/discuss: How much data is required to optimize model parameters? For example, were all three embryo tissues used for training the model in Figure 1 and would the model trained on two samples be able to map niches as expected when tested on a third, leave-out sample?

- For the NSCLC analysis, would the authors suggest that the training using 4 donors with two replicates could train a model generalisable enough to find niches in unseen data with high inter-sample heterogeneity? If so, additional testing on independent dataset would be needed. Could the authors discuss about the power calculation and sample size?

- The GP pruning step is an interesting feature that reduces the dependency on specific GP selection. However, further analysis of the effects of using different prior GP sets (e.g., ligand-receptor vs. metabolite-sensor vs. TF-target GPs) on niche identification would be useful to provide insights into the relative importance of different interaction types. The authors should also discuss whether using more GPs or combined interaction GPs improves niche identification.

- While the chosen performance metrics are comprehensive and appropriate, giving equal weight to spatial conservation scores and niche coherence scores might not be optimal in all scenarios. The authors should consider discussing the potential for weighting these metrics differently depending on the specific biological question and the relative importance of spatial structure versus cell-cell interaction signatures.
- The paper should state whether batch correction was applied to the data in Figure 5c, which shows a large separation of cells by samples.
- To increase confidence in the cell type and niche mapping results, the authors could consider incorporating annotation and assessment by pathologists and visualising more gene markers. For example, the Spp1 Macrophage cluster would need a plot to show the distinct expression of Spp1 and macrophage markers for this cluster.
- Some denovo GPs do not show a clear pattern (e.g., Sfrp1 GP as in Figure S30). Could the authors suggest down-stream analyses or post-hoc tests to gain the confidence of the de novo GPs predicted by NicheCompass?

Minor comments:

- The authors should provide a clear definition of a tissue niche at the beginning of the paper.
- Lines 74, 790: The claim that NicheCompass is the first work to characterize niches via cell-cell interactions needs to be revised, as other methods to characterize cell-cell interaction in the tissue microenvironment exist.
- The marker plot for the niches identified in small cell lung cancer samples (Figure 22f) would need more revision because some groups are not convincingly distinct (e.g. the lymphoid rich niche 10)
- It would be useful for users that the authors discuss on whether NicheCompass Light is recommended. The results shown in Figures S15-S17 show similar performance compared to the NicheCompass with attention weights.

Reviewer #3:

Remarks to the Author:

A. Summary of the key results:

In this manuscript, the authors describe NicheCompass, a computational framework and software package for identifying and characterizing cellular niches in spatially resolved omics measurements. NicheCompass' performance is described through application to several relevant datasets and comparison to related analysis packages. An admirable effort has been taken to apply NicheCompass to publicly-available datasets and interpret the results in biological context.

B. Originality and significance:

This work presents a novel framework for integrating spatial domain detection with gene program prior information. Importantly, their implementation scales to handle large datasets in challenging configurations that will have broad applicability to large spatial transcriptomic datasets.

C. Data and Methodology:

The incorporation of publicly-available data and its subsequent processing and presentation is generally excellent. Because of this, the analysis of the mouse spatial ATAC data greatly weakens the manuscript:

As the authors point out, the major island of Calleja is readily distinguished from the surrounding tissue by its expression of the D3 dopamine receptor. Expression of this one (one!) gene is a good marker for the Islands of Calleja, so discrimination of this region from the nearby structures does not require more data beyond gene expression (as the authors suggest), but rather less and/or better data. Unfortunately, the exact placement of the dataset used shows only the major Island of Calleja and not the others in the ventral striatum that may have helped the authors recognize the structures in the context of niche detection. Supplemental Figure 28 shows likely root cause of this confusion: extremely sparse detection of Drd3 (contrast to Allen ISH atlas or recent BICCN MERFISH datasets). This may help explain why the rest of the section shows rather haphazard correspondence to anatomical structures, leading me to conclude that the interesting results from the white matter structures and Islands of Calleja were the exception, rather than the rule.

The distinction between the anterior commissure and corpus callosum is similarly questionable- all of the biological interpretation of enriched GPs confirming NicheCompass performance (e.g. starting at line 709) surely also apply to the same cell types doing the same thing in the anterior commissure. Given the poor detection of Drd3 in a similarly sized structure and the lower detection of myelin basic protein (Mbp) transcripts in the anterior commissure vs corpus callosum in Supplemental Figure 31, I suspect that differences between these two structures is primarily due to poor detection efficiency. Furthermore, Figures 6f and S31 all show GPs present in both white matter tracts, a fact that is glossed over in the text, which says that these GPs are "meaningful in the context of the Corpus Callosum niche". Figure S31 shows something quite different from that: these GPs are common to both white matter tracts.

These problems and the poor correspondence between other niches and anatomy in Fig 6 leads me to conclude that NicheCompass applied to this dataset produced a few reasonable results that were picked for further analysis. I suggest that this dataset and analysis only be included as cautionary tale in the supplemental section, illustrating the limits of NicheCompass in the face of sparse detection.

D. Appropriate use of statistics and treatment of uncertainties

The authors do include some assessment of variability in their benchmarking against other computational approaches, but in other parts of the manuscript, this is absent. Are the NicheCompass results that are presented in biological context robust against the variation in neighborhood size discussed in the benchmarking? See comment 2 in F below.

E. Conclusions:

There are some biological results presented, but it's unclear to me exactly which biological conclusions are exclusively

available through NicheCompass versus simpler analyses. In this way, some aspects of NicheCompass' design and implementation have questionable value. I have included a few specifics below.

F. Suggested Improvements.

Beyond the issue of the mouse ATAC data above, I have 2 main concerns about this manuscript and a few technical questions for the authors that may lead to clearer text:

1. The NicheCompass approach is presented as if there are no alternative approaches to arrive at the biological conclusions. At the same time, it's unclear which aspects of NicheCompass contribute to the final quality and interpretability of the results. In their own examples, the authors end up referring to the spatial distribution of known marker genes as an achievement of NicheCompass. In these cases where there are small sets of marker genes that readily define a spatial region or subset of cells, what exactly is NicheCompass bringing to the table beyond a gene-expression-based connection to a GP database? This kind of issue is mentioned in passing in the Supplementary Notes ("...necessitating future method development to investigate the disentanglement of intrinsic and spatially-induced variation."), but this topic is a critical aspect of the work that should be addressed quantitatively or discussed (see below).

Examples of specific comparisons that could be quantitatively assessed:

Do GPs have to be integrated during training? In the end, space of prior GPs is whittled down to "...two characteristic GPs per niche based on correlation between GP activities and the expression of the GP's important target genes and ligand and receptor or enzyme and sensor genes". Could this same exercise determine the "characteristic GPs" of GP-agnostic spatial domains?

In the reduced space of final GPs, how many genes are represented? What is the overlap between these genes and highly-variable genes from non-spatial clustering of the same data?

2. The paper claims to produce a "quantitative definition of a niche", but such a thing is difficult to find in the manuscript. For example, the Methods section pronounces that the underlying graph G "... can be defined with a fixed neighborhood-radius threshold if different numbers of neighboring observations ... are desired. Equally, a weighted version of G ... can be used to reflect differences in Euclidean distances between neighboring observations." This kind of flexibility is intriguing- what is the impact of these kinds of changes? Are the results quantitatively identical? I strongly doubt it. Similarly, spatial domain detection methods produce sometimes surprising variability based on random seeds, implementation details of low-level operations, etc. (see e.g. the BANKSY paper <https://doi.org/10.1038/s41588-024-01664-3>) The authors do present evidence of variability in metrics across training runs with different numbers of neighbors, but the impacts of this variation on spatial domains or downstream analysis is unclear. Does this variation result in small changes at the borders of niches? Or entirely different classification of GPs? Are there other parameters or data characteristics that have predictable impact on the results? Inclusion of this kind of information greatly increases the real-world usability of computational tools and allows researchers to establish realistic expectations of NicheCompass for application to their work.

Technical questions:

- I'm curious about some of the outcomes of the GP pruning and regularization. Can these procedures remove genes from the GPs that are present in the data? How many genes are present in the set of GPs after the 'warm up'? and after the completed training?

- NicheCompass framework is the same for spatial data at cellular resolution and also at lower resolution. How does this variation in scale impact the results? Testing NicheCompass on spatially-binned versions of high-resolution data would be one way to help understand how spatial resolution impacts the niche designations.

- NicheCompass produces biologically relevant parcellations of the mouse brain in the case of STARMAP+ and MERFISH data, but they seem to be at varying resolution. Can NicheCompass integrate data from these 2 experimental modalities? What drives the difference in resolution between these 2 datasets? Spatial integration and congruent niche definition across modalities would be a noteworthy accomplishment and extremely useful to the field.

G. References:

The literature is adequately cited, although the collection of spatial domain detection methods has no doubt grown since this manuscript was submitted.

H. Clarity and Context:

The overall organization of the manuscript is clear, the writing is good and the figures clearly communicate the main points of the paper. The breadth of the data exploration effort and clarity of presentation set this manuscript apart from the large pool of similar projects.

Version 1:

Decision Letter:

5th Oct 2024

Dear Mo,

Your Technical Report, "Quantitative characterization of cell niches in spatial atlases" has now been seen by the original 3 referees. You will see from their comments below that while they continue to find your work of interest, some important points are still raised in their reviews. We remain interested in the possibility of publishing your study in Nature Genetics, but would like to consider your response to these concerns in the form of a revised manuscript before we make a final decision on publication.

Briefly, the three reviewers all acknowledge the improvement in this revision but there are still a number of issues that require addressing.

Reviewer #1 provides a thoughtful review, of which the most important point seems to be the parameter choices underlying the NicheCompass model, and how these will affect its usage and performance; they also seem to be framing these comments within the broader concept of what, exactly, a "niche" is, how this relates to cell type, and how NicheCompass's approach may help to clarify these concepts.

Referees #2 and #3 both appreciate the revision and sound supportive of an eventual publication. They have a number of new comments, but these are by and large minor.

In our reading we think the new comments from Referees #2 and #3 are easily addressable. Conversely, Reviewer #1 is asking for, perhaps substantially, more work. Some of this would, to our eyes, be non-negotiable (e.g. selection of model weights/hyperparameters using real ST data) whereas others (on e.g. the conceptual relationship between cell type and niche) are very thought-provoking but, without detailed specific guidance, may be harder to address! Nonetheless we think these are constructive and useful comments and we agree that responding to them will improve your work.

To guide the scope of the revisions, the editors discuss the referee reports in detail within the team, including with the chief editor, with a view to identifying key priorities that should be addressed in revision and sometimes overruling referee requests that are deemed beyond the scope of the current study. We hope that you will find the prioritized set of referee points to be useful when revising your study. Please do not hesitate to get in touch if you would like to discuss these issues further.

We therefore invite you to revise your manuscript taking into account all reviewer and editor comments. Please highlight all changes in the manuscript text file. At this stage we will need you to upload a copy of the manuscript in MS Word .docx or similar editable format.

*2) If you have not done so already please begin to revise your manuscript so that it conforms to our Technical Report format instructions, available

[here](http://www.nature.com/ng/authors/article_types/index.html).

*3) Include a revised version of any required Reporting Summary: <https://www.nature.com/documents/nr-reporting-summary.pdf>

Link Redacted

Sincerely,

Michael Fletcher, PhD
Senior Editor, Nature Genetics
ORCID: 0000-0003-1589-7087

Reviewers' Comments:

Reviewer #1 (Remarks to the Author):

I would like to thank the authors for their effort to address my comments and revise the manuscript. The manuscript has indeed improved in several aspects. However, there are still some major issues that require further attention. These include confusing aspects in the simulation study, potential confounding effects of the identified niches in the model, the need for greater clarity in the method description, and additional analysis to support for the identified niches in the brain.

My additional comments are listed below:

1. R1C1: While most scores appear to be unimodal or stable, it is difficult to discern a clear trend in the F1 for De-Novo GPs. How should this be interpreted?
2. R1C2: The authors highlight the importance of the balance between the gene expression reconstruction loss and the edge reconstruction loss. They set the default hyperparameters in the NicheCompass package based on ablation studies conducted on simulated data and advise users against altering these settings. However, there is a notable gap between simulated data and real ST data, especially regarding specific spatial patterns and expression sensitivity which are significantly different. Therefore, this overall approach doesn't make sense. The authors need to explore metrics for determining the weight based solely on real ST data. A similar justification is also needed for other hyperparameters.
3. R1C3: It appears there are performance trade-offs between niche identification (NID) and gene program recovery (GPR) when selecting different GNN layers and varying the number of neighbors. Given the authors' definition of niches (communities of spatially localized cells with coordinated functions), synergy, rather than trade-off, may be expected. This confusion should be addressed carefully.
4. I do not fully understand the statement, "The red dotted line indicates the baseline of mean F1 scores between the artificially incremented GPs and member genes and randomly selected candidate GPs and genes (with matching numbers)."
5. R1C3 & R1C4: On a related note, the claim that NicheCompass is designed to identify niches and not cell types seems to rest heavily on the weighting of gene expression and edge reconstruction. The model did not take steps to remove the effect of cell type. This suggests that the two concepts – niches and cell types – are closely related. Additional analysis is needed to develop a general principle on their relationship, which would strengthen the work. For example, when two cells of the same type should be in different niches, and when two cells of different types should be considered part of the same niche.
6. R1C6: The visualization is helpful. It would also be important to provide more details in the Methods section regarding how the genes are characterized. Additionally, [minor] is "C5orf36" a typo for "C5orf46" (as shown in the figure)?
7. R1C10: The authors need to clarify in the Methods whether the underlying hierarchical clustering for the dendrogram was performed in the embedding space.
8. R1C9: The authors increased the clustering resolution parameter, leading to further separation of a Fasciola cinerea-like niche, distinguishing it from the V3-like and CA1-like niches. While the authors have provided cell type proportions for each identified niche, it would be more reliable and convincing to show the association between identified niches and brain regions by displaying the expression levels of known marker genes for brain regions, as readily available in databases such as the Allen Brain Atlas.

Reviewer #2 (Remarks to the Author):

The authors have made significant efforts to address each comment from reviewers, resulting in substantial improvements to the manuscript, especially regarding the accuracy and consistency of the figures and the text describing them. This is challenging, given that there is large biological and technical variation of the analysis results across samples, disease types, technologies, and parameter settings in different methods. The additional analyses and figure updates and new supplementary figures provide a comprehensive assessment of the method across various considerations, which will be beneficial for broader audience and applications.

Several remaining, minor comments are:

- Cell type signatures for many niches shown in the Supplementary Fig. 24a are not as expected, e.g., all or most cell types in niches 6, 7, and 12 don't have their corresponding markers highly expressed, but display markers of other cell types. This raises the concern about how reliable and consistent this cell-type specific gene marker analysis is.

- Regarding batch effect, this reviewer agrees with the authors regarding the expected separation of tumour clusters across patients due to heterogeneity, however Figure 5c appears to show clear batch effect in multiple niches that are not just tumour niches. For examples, niches 2 (tumor) and 12 (macrophage enriched) seem to be exclusively in patient 6, while niches 11 (lymphoid aggregate) and 7 (neutrophil expansion) are for patient 5. Similarly for Figure 5e "Shared and donor-specific stromal niches", the niches 9, 11, 6, 7 appear to be specific to donors, but not shared. The Figure 5 will benefit from highlighting more about the shared niches. Finding consistent niches across patients is a question that is attracting a lot of attention in the field.

- NicheCompass light seems to perform not as good as other methods (Figure S16). The authors may add discussion about the usage case for this option to guide users on when they should apply NicheCompass light.

Reviewer #3 (Remarks to the Author):

The authors have greatly improved this manuscript in response to peer review.

The detailed assessment of the model under different parameters and conditions and the testing on simulated data generally support the robustness of the method and validity of the NicheCompass design. The successful integration of two independent mouse brain spatial datasets suggests that NicheCompass will be useful in atlas creation (as the authors point out) but also as a more general-purpose tool for data integration.

I have one major issue on biological interpretation and presentation of the results:

Figure 3a-d and related text:

a. the "V3" (third ventricle) region of the brain is open space, not tissue. The choroid plexus is a distinct structure inside the third ventricle and likely the source of mRNA sampled in that region. Unless there are specific marker genes or other information to disambiguate, I suggest labeling this region V3/CP to clarify.

b. When comparing these results to the anatomy of the mouse brain, there is very little justification for taking NicheCompass over the other results here.

They are all imperfect:

- none of these methods produce a convincing L6b (which is much thinner than the laminar regions shown)

- BANKSY doesn't differentiate thalamic nuclei

- GRAPH ST fails to report laminar structure in cortex

-Cell Charter and DeepLinc smear the pyramidal layer in CA1-3 well beyond their cytoarchitectural boundaries.

Furthermore, the 'artifacts' present in BANKSY, GraphST and CellCharter almost certainly correspond to inhibitory neuron populations that are detected as distinct spatial domains (compare to Gad2 expression in Allen ISH atlas), and the 'artifact' in DeepLinc may correspond to glial cells along the boundary of the thalamus (compare to Aqp4 expression in Allen ISH atlas). In what way is NicheCompass better for not detecting these spatially-localized variations in gene expression?

Similarly, the hierarchy matching of niches to anatomy is different among the methods, but I don't think it's accurate to claim that NicheCompass' is better matched than others.

Despite these shortcomings, the analysis presented in the figure can be a valuable contribution if the writing is improved. The authors should explain how NicheCompass' design leads to the results in 3a-b, describing points of agreement and disagreement with the anatomy as well as with other methods. NicheCompass is one of many tools in this space and the comparisons here are useful in showing how NicheCompass performs in a spatially and transcriptomically complex dataset. This kind of transparent comparison is more valuable than stretching to make NicheCompass seem "better".

Minor Notes:

1. L69-70. Existing methods also incorporate local spatial organization of gene expression

2. The supplemental note 5 on model components and hyperparameter tuning will be helpful for users and some version of it should be included in the package documentation

3. In supplemental figure 36 the mouse brain should be displayed with the dorsal side up. Showing more side-by-side comparisons of niches across modalities would also strengthen the conclusions about successful integration.

Version 2:

Decision Letter:

Our ref: NG-TR65030R1

28th Nov 2024

Dear Mo,

Thank you for submitting your revised manuscript "Quantitative characterization of cell niches in spatial atlases" (NG-TR65030R1). It has now been seen by the original referees and their comments are below. The reviewers find that the paper has improved in revision, and therefore we'll be happy in principle to publish it in Nature Genetics, pending minor revisions to satisfy the referees' final requests and to comply with our editorial and formatting guidelines.

Sincerely,

Michael Fletcher, PhD
Senior Editor, Nature Genetics
ORCID: 0000-0003-1589-7087

Reviewer #1 (Remarks to the Author):

The authors have satisfactorily addressed most of the concerns raised in the previous round of review regarding evaluation metrics, hyperparameter selection strategies, the definition of niches, and other components of the paper. The inclusion of ablation studies on real spatial data strengthens the manuscript.

While the responses are thorough, the manuscript may benefit from additional efforts to streamline the text and improve accessibility as it seems not only too long but some of the more technical descriptions could move to the supplement. But I will leave this to the editors and authors to decide.

Overall, I believe this work represents a valuable contribution to the literature on spatial transcriptomics analysis methods.

Reviewer #3 (Remarks to the Author):

The authors have responded to reviewers' concerns and returned an improved manuscript that balances technical detail, applications of the method, biological exploration and helpful information for eventual users of NicheCompass. I have no further comments on the manuscript and support its publication.

NG-TR65030 Quantitative characterization of cell niches in spatial atlases

Response to Reviewers

We would like to thank the reviewers and the editor for their comments and suggestions, which have substantially improved our manuscript. In short, we mainly addressed the following points, as then described in the answer letter below:

- **Extended evaluation, additional comparisons with other methods, and applications on new datasets:**
 - As suggested by reviewer 1, we created a simulated dataset with ground truth niche labels and ground truth gene program activity for additional rigorous benchmarking. Based on the availability of ground truth niche labels, we included supervised metrics to measure ground truth niche prediction accuracy across methods. We also used this simulated dataset to compare GP inference performance of NicheCompass with two alternative workflows chaining together tools for non-interpretable niche identification and post-hoc characterization (**Methods:** *"Data simulation"*, *"GP inference comparison"*; **Supplementary Fig. 14**).
 - We performed extensive ablation studies on this simulated dataset, evaluating all major design choices and hyperparameters with respect to niche identification and gene program recovery performance. Informed by these ablations, we added a section with user guidance (**Methods:** *"Ablation on simulated data"*; **Supplementary Fig. 15 & Supplementary Note 5**).
 - We added BANKSY¹ as an additional method in all our benchmarking experiments (**Methods:** *"BANKSY"*; **Fig. 3 & Supplementary Fig. 13,14,16,17-20**).
 - We revised the niche identification approach during benchmarking to improve comparability between methods (**Fig. 3c,e & Supplementary Fig. 13,17**).
 - We added additional experiments on spot-level resolution data and artificially binned single-cell resolution data to explore the effect of resolution on niche identification and gene program recovery performance (**Supplementary Fig. 16**).
 - We quantitatively evaluated how many of the genes identified in spatial GPs by NicheCompass are also recoverable via non-spatial differential expression analysis (**Supplementary Fig. 37**).
- **Assessment of model robustness and generalizability**
 - We evaluated the robustness of identified niches and recovered GPs under diverse scenarios including different random seeds, neighborhood graphs, and different sets of prior GPs (**Supplementary Fig. 8-10**).

- We added additional analyses to test the generalizability and identification of unseen niches in leave-one-out scenarios and performed a spatial reference mapping power analysis to evaluate the impact of the size of the reference (**Supplementary Fig. 38**).
- **Additions and modifications to analyses in the biological context**
 - We added a new analysis in which we integrate the STARmap and MERFISH mouse brain datasets to show NicheCompass' applicability across technologies (**Supplementary Fig. 36**).
 - We revised parts of and added additional supporting evidence for the NSCLC analysis (**Fig. 5 & Supplementary Fig. 24,25,29**).
- **Clarification of text and method focus and improvements in presentation**
 - We have considerably shortened the manuscript and, as suggested by reviewer 3, moved the analysis on the spatial ATAC-RNA seq mouse brain dataset to the Supplementary Material (**Fig. 6 -> Supplementary Fig. 30 & Supplementary Note 7**).
 - We provided conceptual clarifications (e.g. definition of a "niche" and relation/difference to cell types).
 - We clarified benchmarking metrics and the metric normalization procedure and improved the general presentation of benchmarking results (**Methods: "Evaluation Metrics"; Fig. 3f,g**).
 - We improved the quality, readability and formatting of all figures.

Overall, besides modifications to and shortening of existing analyses and figures (including removal of **Fig. 6**), we added 13 new supplementary figures and two new supplementary notes:

- **Supplementary Fig. 8:** Niche and GP inference reproducibility and generalizability
- **Supplementary Fig. 9:** Niche and GP inference neighborhood graph robustness
- **Supplementary Fig. 10:** Niche and GP inference with different prior GP sets
- **Supplementary Fig. 14:** Data simulation
- **Supplementary Fig. 15:** NicheCompass ablation
- **Supplementary Fig. 16:** Additional ablation and benchmarking on spot-level resolution data
- **Supplementary Fig. 24:** Characterization of stromal niches
- **Supplementary Fig. 25:** Tumor niches neighborhood composition
- **Supplementary Fig. 29:** Characterization of infiltrating macrophages in Donor 13 as SPP1+ macrophages
- **Supplementary Fig. 30:** Spatial ATAC-RNA-seq mouse brain tissue niche organization
- **Supplementary Fig. 36:** NicheCompass integrates tissue samples across different spatial transcriptomics technologies
- **Supplementary Fig. 37:** Overlap between characterizing GP genes and differentially expressed genes (DEGs) from non-spatial analysis in the seqFISH mouse organogenesis analysis
- **Supplementary Fig. 38:** seqFISH mouse organogenesis spatial reference mapping
- **Supplementary Note 5:** Ablation experiments and guidance on hyperparameter selection
- **Supplementary Note 7:** Spatial ATAC-RNA-seq mouse brain niche annotation & characterization

In the following, reviewers' comments appear in blue, our point-by-point answers appear in green. Text and figures we have added or altered in the manuscript are highlighted in green in the manuscript.

Response to Reviewer 1

Remarks to the Author:

The manuscript introduces NicheCompass, a graph deep learning method designed to quantitatively characterize tissue niches by integrating spatial omics data with gene programs. The model incorporates both molecular and spatial information to identify cell niches and their underlying signaling events. The authors applied NicheCompass to several biological contexts, including mouse organogenesis and human cancer, showcasing its ability to uncover gene programs that may drive niche identity. However, while the method seems overall quite reasonable and makes sense, the manuscript has a few major issues, including a lack of comparison with other approaches, a lack of quantitative simulation evaluation, and the confusing use of some of key concepts and assumptions throughout the work. My comments are listed below.

Thank you so much for your nice summary and the constructive suggestions and comments that have drastically improved our work.

== Major concerns on the methodology / approach ==

1. The NicheCompass model is tailored for biological processes by including gene programs, and the neighbor/self dissection makes sense for spatial omics. The graph decoder approach seems like a meaningful way to ensure the cell embeddings include spatial information. However, even though the authors used a few datasets to assess NicheCompass, the overall evaluation is limited to the specific contexts and the analysis often feels lengthy and ad hoc. One major deficiency of this work is the lack of a well-designed simulation study to quantify the impact of different designs in the models and provide guidance on parameter selection. This is certainly an issue for the current state of spatial transcriptome and single-cell computational methods papers. However, a leading journal like Nature Genetics should set a high benchmark standard for the field.

R1C1: We are happy that you agree that the modeling approach makes sense, and highly appreciate the suggestion to design a simulation study to evaluate different model designs and provide guidance on hyperparameter selection. To address this, we have customized the SRTsim² package to simulate spatial transcriptomics data with ground truth niche labels based on a reference dataset and niche-specific ground truth spatial gene program activity. We then used this simulated data to implement careful ablation experiments which we evaluated based on niche identification and GP recovery performance. Details of the simulation can be found below, in the **Methods** section under the title **Data Simulation**, and in **Supplementary Fig. 14**. Details of the ablation can be found below, in the **Methods** section under the title **Ablation on simulated data**, and in **Supplementary Note 5 & Supplementary Fig. 15**.

Data simulation: we have adapted the SRTsim package² to be able to inject ground truth spatial gene program activity into different tissue niches. Specifically, we (1) have sampled GPs from the NicheCompass prior GPs to be upregulated, (2) for each GP have sampled a tissue niche and two cell types, (3) have sampled genes from these GPs, (4) have used an additive gene expression model in the simulation to upregulate the sampled genes in cells of the sampled tissue niche for cells of the sampled cell types that were spatial neighbors.

Extract Supplementary Fig. 14 | Data simulation. **a**, The reference-based simulated tissue and a UMAP representation of the PCA-reduced gene expression space colored by ground truth niches. **b**, Same as **a** but colored by ground truth cell types. **c**, An example for the injection of a ground truth gene program which was upregulated in Niche 2 of the tissue via an additive gene expression model. The target genes were upregulated in all Cell Type 3 cells if they had Cell Type 2 cells in their K neighborhood (with K=6). Equally, the source genes were upregulated in Cell Type 2 cells. The increment factor determined the strength of up-regulation (see full version in **Supplementary Fig. 14**).

While it took us some time to modify the SRTsim package to flexibly generate simulations, we believe that this helped us greatly in clarifying basic components and generating ground truth labels for niches. We will release the code for the modified data simulation package, which we believe can further help the community to build upon and benchmark existing work.

Ablation study: Based on the simulated data, we have performed extensive ablation studies, i.e. we have systematically altered major design choices and hyperparameters in our model and compared the impact on performance. To this end, we (1) have used prediction accuracy metrics from the SDMBench package³ to evaluate our identified niches against the simulation ground truth niches (to evaluate niche identification performance) and (2) have reported the (niche-aggregated) F1 scores between the niche-specific set of enriched gene programs and important GP genes discovered by NicheCompass and the niche-specific set of ground truth gene programs and sampled genes injected into the simulation data (to evaluate GP recovery).

Extract Supplementary Fig. 15 | NicheCompass ablation. Niche identification (NID) and GP recovery (GPR) metrics on the simulated data under different model design choices and hyperparameters. The first row in each panel contains metrics that measure the correspondence between predicted and ground truth niche labels, the second row contains F1 scores between enriched and artificially incremented GPs and member genes. Boxplot elements are defined as: center line, median; box limits, upper and lower quartiles; whiskers, 1.5x interquartile range. The red dotted line indicates the baseline of mean F1 scores between the artificially incremented GPs and member genes and randomly selected candidate GPs and genes (the number was matched). Ablations were performed on **a**, the weighting of the reconstruction loss hyperparameters indicating that both edge and gene expression reconstruction are essential components (see full version in **Supplementary Fig. 15**).

We included the following design choices and hyperparameters in our ablation study:

1. Edge reconstruction and gene expression reconstruction loss weights
2. Gene expression decoder weight regularization loss weights
3. Encoder architecture
4. Neighborhood graph size
5. Number of *de novo* GPs
6. GP selection via active GP threshold (GP pruning)
7. Use of prior GPs versus fully connected decoder (i.e. uninterpretable embeddings)

We found that:

1. Finding a balance between gene expression and edge reconstruction is a key element for good niche identification (NID) and GP recovery (GPR) performance.
2. Regularization of *de novo* GP weights is essential for GPR and especially the correct retrieval of important *de novo* genes.
3. The NicheCompass Light variation with a GCN layer could achieve better NID on this dataset while the GATv2 layer led to better GPR.
4. A smaller number of neighbors was more efficient in NID while a larger number of neighbors facilitated GPR.
5. The inclusion of *de novo* GPs was crucial for GPR with a default number of 100 *de novo* GPs leading to the best results.
6. GP pruning slightly improved NID and GPR while reducing the embedding size.
7. Comparable NID performance was achievable without prior GPs but this came at the cost of GP gene recovery preventing an inherent niche characterization.

We also want to apologize that the analysis often felt lengthy and ad-hoc. To address this, we have shortened various segments in the **Results** section. Among many small adjustments we have:

- Removed vast parts of the analysis related to **Fig. 6**, also based on feedback from reviewer 3 (see **R3C1**; we now only focus on the newly introduced capability of characterizing niches through multimodal GPs).
- Removed several parts of the analysis on **Fig. 2**.
- Shortened the description of differences between methods in the benchmarking results related to **Fig. 3**.

Under the light that reviewer 3 highly appreciated the breadth of our data exploration effort (see **R3C14**), we hope that you find these adaptations sufficient and well chosen.

2. The NicheCompass model uses a set of hyperparameters to balance the factors (Methods), along with many others. How to set these parameters seems vague and confusing. For the entire loss function in Equation (18) of the Methods section, how to do the hyper-parameter selection for the weighting factors of each loss component is unclear. This is related to the previous comment on the lack of well-designed simulations in the work to carefully assess the model parameters and their influence on the analysis results.

R1C2: Thank you for this constructive comment. We have considered this point by treating the loss weights of Equation (18) as major hyperparameters covered in our ablation studies described in the previous comment (see **R1C1**). The results are reported in **Supplementary Fig. 15** and in **Supplementary Note 5**, where we guide the user how to set hyperparameters.

In summary, we found that it is crucial to find a balance between the gene expression and edge reconstruction losses to obtain good NID and GPR and have set the default hyperparameters in the NicheCompass package accordingly. As described in **Supplementary Note 5**, we do not recommend users to change these. In terms of gene expression regularization, we found that it is important to regularize the weights of *de novo* GPs to improve niche identification performance and correctly retrieve *de novo* genes. We have set the default hyperparameters in the NicheCompass package accordingly.

3. Conceptually, the authors did not clearly articulate the novelty of the niche that NicheCompass is focused on. Throughout the paper, it seems like the authors often conflate cell type and niche in their analysis. Do the metrics for scoring different methods actually make sense in the context of niche identification?

R1C3.1: Thank you very much for your thoughtful feedback and we apologize if we have not clearly articulated our niche concept. We have now added an explicit definition in the **Introduction** section in **line 53**: “...niches, defined as communities of spatially localized cells with coordinated functions”). We want to highlight that, as described in the **Introduction** section, these niches are largely shaped by cell interactions and, therefore, NicheCompass is explicitly designed to take into account cellular interactions to identify and characterize such niches quantitatively based on cell interaction activity. We consider this the main conceptual novelty and have revised the **Abstract** and **Introduction** sections to better highlight this. For instance, in **lines 64-65**, we now explicitly mention “*there is a lack of computational approaches to simultaneously identify and characterize niches based on the cellular interaction mechanisms shaping*”

them”, and in **lines 72-76** we describe that NicheCompass fills this gap: “Here, we present NicheCompass (Niche Identification based on Cellular Graph Embeddings of Communication Programs Aligned across Spatial Samples), a graph deep learning approach to simultaneously identify and quantitatively characterize niches based on learned cell embeddings of signaling events, encoding activities of spatial gene programs (GPs)”. We also want to point out that such niches can consist of homogeneous cell populations, i.e. spatially localized cells of the same cell type, or heterogeneous cell populations. We have added such a clarification in **lines 53-54**.

Moreover, to better separate cell types and niches in our analysis, we have implemented the following improvements:

- We have considerably revised the mouse organogenesis analysis presented in **Fig. 2** which we believe was the major cause for this conflation as during early embryonic development cell type annotations often coincide with niche annotations due to colocation of homogenous cells (in region-specific progenitor state cell types) forming spatially consistent niches⁴. We also want to point out that our cell type annotations (**Fig. 2b**) come from the original study (see **Fig. 2b** in Lohoff et al. 2022 Nat Biotech.⁴) and have probably contributed to this conflation as cell types are sometimes labeled based on their anatomical region. We have hence changed the label of **Fig. 2b** from “Cell Types” to “Original Annotations”.
- In addition, we have tried to make it more clear across all analyses that the purpose of NicheCompass is to identify niches and not cell types. Cell types are now merely used to investigate the composition of niches in our analyses. For instance, in **line 199**, we have removed the analysis of the embedding space in terms of cell types, and, in **line 604**, we have revised a statement that might suggest that cluster labels coincide with cell types.
- Further points that might have contributed to conflation are related to individual metrics and are clarified below. Most notably, we believe that we have mistakenly described some spatial conservation/consistency metrics as supervised (with cell type labels), which caused confusion when in reality these metrics are not measuring cell types but spatial consistency, with cell types only as auxiliary labels. We have corrected this in the **Methods** section in **lines 1284-1285**.

We hope that this addresses existing concerns about conflation.

Finally, we want to emphasize that we have designed our suite of metrics to evaluate methods in an unsupervised scenario where no (high quality) ground truth niche labels are available. We have now additionally evaluated all benchmarked methods on our simulated data for which we have ground truth niches available. For this, we have used the prediction accuracy metrics from the SDMBench³ package. The results are reported in **Supplementary Fig. 14** and show high alignment between the ground truth-based prediction accuracy metrics from the SDMBench package and our metrics, both of which show NicheCompass and BANKSY as top performing methods. This should increase the confidence in our metrics. Additionally, we have clarified concerns related to some of the individual metrics below. We believe that we might have not been clear enough in describing what exactly they measure and that this might have led to misunderstandings that also contributed to a conflation of cell types and niches.

e Metrics

Model	CAS	MLAMI	CLISIS	GCS	NASW	CNMI	Overall Score	Model	NARI	NNMI	HOM	COM	Ground Truth Prediction Score
NicheCompass Light	0.729	0.579	0.932	0.876	0.770	0.435	0.866	NicheCompass Light	0.930	0.929	0.930	0.928	0.929
BANKSY	0.609	0.580	0.938	0.890	0.657	0.431	0.755	BANKSY	0.924	0.925	0.925	0.926	0.925
NicheCompass	0.173	0.588	0.906	0.858	0.736	0.415	0.708	NicheCompass	0.891	0.903	0.902	0.903	0.900
GraphST	0.241	0.547	0.916	0.914	0.564	0.406	0.584	GraphST	0.874	0.878	0.878	0.879	0.877
STACI	0.299	0.351	0.912	0.852	0.628	0.284	0.456	CellCharter	0.792	0.889	0.868	0.912	0.865
CellCharter	0.102	0.529	0.683	0.773	0.616	0.479	0.444	DeepLinc	0.725	0.727	0.723	0.730	0.726
DeepLinc	0.148	0.446	0.891	0.796	0.519	0.262	0.314	STACI	0.368	0.533	0.511	0.556	0.492

Extract Supplementary Fig. 14 | Data simulation. d, The reference-based simulated tissue colored by the predicted niches of each method. **e**, On the left side, the metrics from the NicheCompass benchmarking suite and, on the right side, four metrics that measure the performance of the predicted niches compared to the ground truth. The overall score and ground truth prediction score are computed by min-max normalization and subsequent aggregation of the individual metrics in the tables. The ranking of methods is largely consistent between the two metrics suites (see full version in **Supplementary Fig. 14**).

Specifically:

- CAS was designed to work on cell types. Do we expect that it should do well on niches that were identified by each method?

R1C3.2: Thank you for this question. We want to clarify that while the original cell type affinity metric was designed to measure the enrichment of given cell types in the neighborhood of other cell types, the cell type affinity similarity (CAS) is our adaptation of this metric (and not the originally designed metric) which compares how similar the cell type affinity is between the embedding space of the model and the physical (tissue) space (see **Methods** section under the title **Spatial Consistency Metrics**). As such, it is purely a measure for spatial consistency of the embedding space of a model and takes the cell type annotation only as an auxiliary label to evaluate this spatial consistency on a more global level (i.e. it doesn't measure exact spatial consistency on the cellular level such as the GCS but does it on a semantic aggregate level using cell types). In our opinion, it is therefore a very useful metric to evaluate cellular embeddings in the context of niche identification, which requires spatial consistency of the embedding space. We have

reworked the **Methods** section to clarify this point in **lines 1290-1291**. Besides, we have relabeled the “*spatial conversation*” score as “*spatial consistency*” score throughout the manuscript which we believe better highlights why these metrics are important for niche identification.

- For CLISI, are the cell type annotations for the calculation coming from the original paper?

R1C3.3: Yes, all cell type annotations used for the computation of metrics come from the datasets’ original publications. We have added this information in the **Methods** section in **lines 1372-1373**. Thank you for helping us to clarify this for future readers.

- GCS is measuring how close the latent embeddings are of physically-proximal cells.

Since the niche embedding for each cell is incorporating its neighbors within the embedding, these metrics are all likely to score NicheCompass higher than competitors that don’t rely as highly on neighbor information. This is unfair and somewhat misleading.

R1C3.4: Thank you for this assessment. We agree that NicheCompass is expected to perform well on this metric due to the incorporation of the neighborhood in the cell embedding. Nonetheless, we believe that this is an important metric to guarantee spatial consistency and is exactly why we have designed NicheCompass in this way. It is very important to note that all other methods from our benchmarking experiments also highly rely on neighborhood information and NicheCompass is on certain datasets outperformed by these competing methods (STACI, GraphST, BANKSY) on this particular metric (Supplementary Fig. 18a & 19a). STACI and DeepLinc use similar edge reconstruction objectives as NicheCompass and are therefore equally expected to perform well on the GCS. BANKSY and Cellcharter also explicitly use neighborhood information as described by the authors:

- BANKSY: “*We captured gene expression variations in the spatial neighborhoods of cells using a weighted average of the expressions of neighboring cells and a kernel inspired by the Gabor filter.*”, “*We used this to compute feature-wise z-scaled versions of two matrices: an average neighborhood expression matrix and an AGF matrix. These matrices are then scaled on the basis of a mixing parameter λ , which controls their relative weighting, and concatenated with the original gene–cell expression matrix to construct two neighbor-augmented matrices, one for cell typing and another for domain segmentation*” (Singhal et al. 2024 Nat Genet¹).
- CellCharter: “*Neighborhood aggregation. Incorporating the features of neighboring nodes into a given spot or cell is achieved through neighborhood aggregation. This technique involves concatenating the features of a spot/cell with those aggregated from its neighbors*” (Varrone et al. 2024 Nat Genet⁵).

Due to this consistent integration of neighborhood information by all benchmarked methods, we believe that this is a fair metric (which is supported by the fact that NicheCompass is not the best performing method on this metric), and the metric’s alignment with the design of all methods strengthens its case as an unsupervised evaluation metric. We hope that you can agree with this based on the supplied information.

At the bare minimum, the scores should be normalized, since the improvement of NicheCompass over prior methods is listed as a percent improvement over the next-best tool over these metrics, which is not very meaningful, since they are on different scales.

R1C3.5: Thank you for this valuable feedback. We apologize that we have not clearly presented our normalization procedure. In **Fig. 3f,g**, where we report percent improvements of the metric categories, all scores are aggregates of normalized (min-max scaled) individual metrics across methods and runs (i.e. the best run across all methods received a score of 1 and the worst run across all methods received a score of 0 for each individual metric). The reason that the normalized scales are not immediately visible from the plots in **Fig. 3f,g** are the following:

- As the metric categories are aggregates of individual 0-1 scaled metrics, the final range of the metric category might not exactly be between 0 and 1.
- The bars only display the mean values across 8 runs (incl. confidence interval).
- We have reduced the axes to remove areas of the 0-1 scale where no bar was located (to save space).

Our normalization procedure is aligned with previous benchmarking studies in the single-cell field (e.g. Luecken et al. 2022 Nat Methods⁶), where aggregate metrics have been equally computed from scaled individual metrics. However, we agree that percent improvements are not the most meaningful measure to indicate differences between methods. As a result, we have changed the improvement indication to (average) absolute value improvements instead of percent improvements. Due to the consistent scale for all metrics, this should represent a meaningful difference measure. We have revised **Fig. 3** accordingly. We also want to note that, in contrast to **Fig. 3f,g**, we have not normalized the metrics in **Supplementary Fig. 18** and **Supplementary Fig. 19**, which might have contributed to this lack of clarity (in these plots we don't report any improvements but instead wanted to keep the raw metrics to enable future comparison of new methods directly on the metrics level). We have added corresponding clarifications in the **Methods** section in **lines 1358-1362** and in the caption of **Fig. 3**. We hope that you are happy with these modifications and appreciate the feedback that helped us to clarify and improve the method comparison.

Indeed, **Fig. 4a** shows an instance where the embedding from NicheCompass would likely not score well on the supervised metrics for spatial conservation, since the niches captured in the breast cancer dataset are composed of multiple different cell types.

R1C3.6: Thank you for raising this doubt. Importantly, as previously mentioned, we are not using the originally proposed CLIS and cell type affinity scores but rather have adapted these metrics to measure their similarity in the embedding space of a model and in the physical tissue space. With these adaptations, our metrics purely measure spatial consistency and use cell types barely as auxiliary labels. Accordingly, we have relabeled these metrics in the **Methods** section in **lines 1284-1285** to remove their classification as “*supervised metrics*” and hope that this prevents misunderstandings that we used cell types as a ground truth supervised label.

Moreover, we have computed the metrics on this dataset (CLIS: 0.913 & CAS: 0.210). Both metrics are within reasonable ranges when compared to **Supplementary Fig. 18a** and **Supplementary Fig. 19a**. It is important to note that while the metrics may be on the lower end of the spectrum, this is because it is in general challenging for models to obtain high spatial consistency if niches are very diverse and hence

models cannot rely that strongly on gene expression of the “self” cell but are more reliant on the neighborhood. However, our metrics for spatial conservation/consistency do not require niches to be composed of individual cell types and can handle multiple different cell types. Instead of requiring niches to be composed of just one cell type, our metrics measure how cell type composition is preserved in the embedding space of a model compared to the physical tissue space, i.e. if cell types are intermingled in a specific pattern in physical tissue space, this pattern must be reflected in the embedding space of a model to obtain a high score on these metrics. It is therefore possible to obtain high scores with niches composed of multiple different cell types if cellular neighborhoods in the embedding space are similar to those in physical tissue space. Vice versa, low scores can be obtained with niches composed of a single cell type if cells in physical tissue space are neighbored by diverse cell types.

4. The authors claim that the NicheCompass is novel in its ability to “quantitatively characterize tissue niches”. While the method covers many use cases and multiple applications are demonstrated in the Results section, it is unclear what is the quantitative characterization of niches. Is it the niches (spatial domains) recognized by clustering or cell encoding with spatial information integrated? This is quite confusing and is not well defined.

R1C4: Thank you for appreciating the many use cases and applications that we demonstrated and for pointing out this confusion about the quantitative characterization of niches. We agree that the quantitative characterization of niches can be defined in a clearer way. We have explicitly designed the architecture of NicheCompass to model cellular communication and learn an embedding space where each embedding feature encodes a specific signaling pathway (spatial gene program). As a result, the values of the embedding can be understood as activities of these programs and therefore offer a quantitative characterization of a cell in terms of cell signaling events. As tissue niches are driven by communalities in underlying cell interactions, clustering of these cell embeddings reveals tissue niches while subsequent differential testing of the embeddings on the cluster-level reveals the spatial gene programs and hence signaling events driving those niches. This contrasts with existing methods, which focus on the identification of niches (spatial domains) but rely on post-hoc analysis to characterize these. On our simulated data, we also added a new analysis to show that integrating GPs during model training leads to better recovery of ground truth signaling events compared to post-hoc analysis with two alternative workflows (workflow 1: niche identification with BANKSY followed by gene set enrichment analysis per niche; workflow 2: niche identification with BANKSY followed by cell-cell communication inference with LIANA). As a result, NicheCompass can provide more accurate characterization of niches. Details for this analysis can be found in the **Methods** section under the title **GP inference comparison**.

Extract Supplementary Fig. 14 | Data simulation. f, F1 scores between inferred and ground truth upregulated GPs for three different workflows to infer niche-specific GPs. Displayed are eight training runs with different random seeds and a kNN neighborhood graph with $k=6$ (the ground truth cell interaction range). Boxplot elements are defined as: center line, median; box limits, upper and lower quartiles; whiskers, 1.5x interquartile range. NicheCompass considerably outperforms alternative methods, providing evidence that it is useful to integrate GPs during model training (see full version in **Supplementary Fig. 14**).

We hope that this clarifies our novel ability to quantitatively characterize tissue niches.

With regard to the manuscript, we have made further modifications to make this more clear:

- We have revised the **Abstract** section and in **lines 34-38** now explicitly state: *“NicheCompass learns interpretable cell embeddings of local signaling events that allow to simultaneously identify niches and elucidate the underlying cellular processes constituting each niche. Unlike existing methods, it uniquely enables quantitative characterization of niches based on these cell embeddings which encode cellular activity of diverse communication pathways, represented as spatial gene programs”*.
- We now consistently talk about *“cell embeddings of (local) signaling events”* across the manuscript instead of sometimes referring to *“latent space”*, and other times to *“GP space”* or *“embedding”*.
- We have revised the **Introduction** and **Discussion** sections to make clear that the quantitative characterization of niches refers to the inferred GP activities. For instance, in **lines 72-76**, we now state: *“Here, we present NicheCompass (Niche Identification based on Cellular Graph Embeddings of Communication Programs Aligned across Spatial Samples), a graph deep learning approach to simultaneously identify and quantitatively characterize niches based on learned cell embeddings of signaling events, encoding activities of spatial gene programs (GPs)”*; in **lines 718-720**, we now state: *“We introduced NicheCompass, a graph deep learning approach designed based on the principles of cellular communication to identify and quantitatively characterize tissue niches through spatial GP activity.”*

5. NicheCompass focuses on showing the importance of prior information in identifying cellular niches. But how important is that and whether prior information leads to bias toward known processes heavily?

R1C5: Thank you for this question. We agree that this is an interesting and important consideration. On the one hand, in our ablation studies, we have included a scenario where we do not make use of prior information (see **R1C1**; **Supplementary Fig. 15**). In this specific scenario, we observed comparable performance demonstrating that the prior knowledge does not lead to strong bias on niche identification and does not impact it negatively. However, the absence of prior information leads to absence of interpretability of the cellular embeddings and prevents inherent niche characterization. As presented in **R1C4**, integration of prior information can significantly boost the niche characterization performance compared to post-hoc analysis.

Extract Supplementary Fig. 15 | NicheCompass ablation. Niche identification (NID) and GP recovery (GPR) metrics on the simulated data under different model design choices and hyperparameters. The first row in each panel contains metrics that measure the correspondence between predicted and ground truth niche labels, the second row contains F1 scores between enriched and artificially incremented GPs and member genes in ground truth niches. Boxplot elements are defined as: center line, median; box limits, upper and lower quartiles; whiskers, 1.5x interquartile range. The red dotted line indicates the baseline of mean F1 scores between the artificially incremented GPs and member genes and randomly selected candidate GPs and genes (the number was matched). Ablations were performed on **g**, a model design without prior GPs and different embedding sizes, indicating good NID performance at the cost of GP gene recovery compared to a model design with prior GPs (see full version in **Supplementary Fig. 15**).

Additionally, it is true that the use of prior information can lead to biases toward known processes, which we have acknowledged in the Discussion section in **line 737**; however, this is an essential component to make cell embeddings interpretable in terms of known biological processes. To mitigate the bias from prior information, we have integrated the concept of spatial *de novo* GPs that can partially address this as outlined in the analysis on **Fig. 4**.

Finally, we have specifically investigated the impact on niche identification and GP inference when only including certain types of prior GP sets in the seqFISH mouse organogenesis analysis. The results are reported in **Supplementary Fig. 10**. We observed that the identified niches were robust against the selected prior GP sets while each prior GP set elucidated different underlying biological processes.

Extract Supplementary Fig. 10 | Niche and GP inference with different prior GP sets. a, Brain and gut niches identified when only using a specific set of prior GPs. All niches from the analysis in **Fig. 2** can be recovered independently of the used set of prior GPs; however, different prior GPs elucidate different aspects of underlying niche biology, highlighted with a characterizing GP and its most important gene in the Midbrain niche (see full version in **Supplementary Fig. 10**).

I think the design of identifying potential de novo gene programs is a bit awkward in NicheCompass as it requires users to define the number. But more importantly, how are de novo gene programs analyzed for biological significance? Is it easy to differentiate between de novo gene programs that are biologically meaningful and those that are due to simply noise within the data? For example, for the breast cancer dataset, it seems like there are only 5 out of at least 86 de novo gene programs that seem to encode biological knowledge.

R1C6: Thank you for this question. We want to clarify the process of how *de novo* GPs have been selected for biological significance. First, it is important to note that only the maximum number of *de novo* gene programs is specified by the user. Throughout our analyses (and also in the ablation experiments presented in comment **R1C1**), we found that ~100 is a sufficient number in combination with the default prior gene programs from Omnipath, NicheNet and MEBOCOST and specified this as default for this hyperparameter which does not have to be changed by the user. We have added this to the user guidance in **Supplementary Note 5**. The reason why we kept it as a hyperparameter is that the user might want to change this based on the supplied prior GPs (either default GPs or GPs from an additional database or custom GPs). For instance, if only few prior GPs are provided to NicheCompass, a higher number of *de novo* GPs is reasonable. All specified GPs (including *de novo* ones) are then pruned during model training as described in the **Methods** section under the title **Gene program pruning**. Enrichment analysis of the remaining GPs returned 35 (out of the 100) GPs to be enriched in any niche. Out of the 35 GPs, we identified 5 GPs as characteristic GPs of niches based on correlation between GP activities and the expression of the GP's important genes (as described in the **Methods** section under the title **Selection of characterizing gene programs for each niche**).

In addition, we have now updated the package and the figure with a new way to visualize the GPs (**Fig. 4g,h**), allowing the user to interpret the biological significance of a GP using the gene importances assigned by NicheCompass more clearly. While there are more *de novo* GPs associated with it, we focus on the ones that, as users, we found to be more informative. However, NicheCompass enables the users to explore these to identify new biology.

Extract Fig. 4 | NicheCompass identifies meaningful niches and *de novo* gene programs in human breast cancer. Sunburst plots of GPs and their member genes for *de novo* 37 GP (**g**) highlighting the importance of the keratin gene family and the involvement of an uncharacterized gene (*C5orf36*), and *de novo* 86 GP

(h) displaying a more complex KRT8-driven GP that includes elements of fatty acid metabolism (*FASN*, *ABCC1*), and *ELF3* as a potential gene regulator. The scale represents the gene weights inferred by NicheCompass (see full version in **Fig. 4**).

6. In Line 152 of the Methods section, the graph decoder module is used to reconstruct the entire adjacency matrix *A* through computing the cosine similarities of pairwise latent feature vectors. However, the adjacency matrix *A* is extracted from a disconnected graph composed of all sample-specific symmetric *k*-nearest neighbor subgraphs. I have doubts about encouraging recovering the adjacent elements between samples to be 0, since there may be spatial relationships between cells from different samples (e.g., adjacent slices in a spatial atlas of the whole mouse brain).

R1C7: Thank you for raising this doubt. We apologize that we have not clearly described that this is not the case. Our method does not reconstruct the entire adjacency matrix *A* but instead just the part where nodes of an edge belong to the same sample (for exactly the valid reason mentioned by you). While we described this in **line 147** of the **Results** section and in **lines 979-983** and **1036-1039** of the **Methods** section, we have now also added it to **line 929** in the **Methods** section and the caption of **Fig. 1** (“*A graph decoder reconstructs the sample-specific input adjacencies*”), which likely produced this misunderstanding. We hope this makes it clear now and fully agree with your conclusion, which is exactly why we have chosen this design.

7. The work lacks comparisons with some of the more recent niche embedding learning methods such as COVET (Haviv et al. 2024 Nat Biotech) and BANKSY (Singhal et al. 2024 Nat Genet). For example, in Line 342, the author mentioned that only NicheCompass could identify four separable cortical layer niches. It would be important to see whether other methods can identify the cortical layer by trying different cluster numbers.

R1C8: Thank you, we agree with this suggestion. Although it is difficult to include all available niche embedding learning methods given their plentiful amount and rapid pace of development, we have added CellCharter just before our submission as a new niche embedding learning method published in 2024 (Varrone et al. 2024 Nat Genet⁵). As suggested, we have now additionally added BANKSY¹, which also postulated niche/spatial domain identification as one of its main goals.

As a result of including BANKSY, we have:

- updated **Fig. 3c-f** to integrate BANKSY in both the single sample and sample integration scenarios (including renormalization of the metrics of all methods).
- updated **Supplementary Fig. 12b** with the BANKSY tissue niche composition
- updated **Supplementary Fig. 13** (added tissue annotated with BANKSY in panel **a**, added BANKSY metrics in panel **b**, renormalized all methods in panel **b** with BANKSY included).

In single sample scenarios, BANKSY ranked second on most datasets (after NicheCompass); however, in sample integration scenarios, it unveiled important deficiencies in data integration (**Fig 3**).

Extract Fig. 3 | Benchmarking NicheCompass across various scenarios. f, g, Performance summary metrics of NicheCompass and similar methods on four single-sample (**f**) and three multi-sample (**g**) datasets. The bars display the mean across eight training runs for each method and dataset with varying numbers of neighbors. The error bars display the 95% confidence interval. A score change accentuates the difference between the overall score of NicheCompass and the second-best method. Missing bars reflect unsuccessful model training due to memory constraints (see full version in **Fig. 3**).

In the case of COVET⁷, our understanding is that its main goal is not the identification of niches in the sense of spatial domains but the integration of scRNA-seq and spatial transcriptomics data and imputation of genes (see here for the only example where the authors showcase the method); therefore, we have not included it.

We also appreciate the suggestion to try out different cluster numbers and have modified our method comparison analysis as a result (**Fig. 3c**). Specifically, to make our clustering approach less dependent on the clustering resolution and fairer across methods, we have spent considerable effort not to fix the number of clusters ad-hoc but instead incrementally increase the resolution for each method individually until niches became too scattered and noisy (i.e. not spatially adjacent anymore, intermixed with other niches and not anatomically meaningful). In our opinion, this approach is reflective of what an analyst would use to identify more fine-grained niches in a method-agnostic manner and should therefore be well suited for unbiased method comparison. After changing our analysis approach, it became apparent that NicheCompass was not the only method that could identify four separable cortical layer niches. We therefore want to thank the reviewer for pointing this out and have removed the claim from the **Results** section. We also changed the clustering workflow of CellCharter from the native mclust to Leiden clustering for comparability, which significantly improved the obtained clusters from this method. Nonetheless, we find that after thorough fine-tuning of the clustering resolution for each method, NicheCompass was the only method that could identify all anatomically relevant niches while not producing spurious artifacts. The overall result of NicheCompass' superiority in terms of identified niches therefore remains intact and the updated results can be found in **Fig. 3** and the corresponding text. We have updated the **Methods** section with our new approach under the title **SlideSeqV2 mouse hippocampus**. We have also updated **Supplementary Fig. 13, 17, 18** and **19** as a result.

Extract Fig. 3 | Benchmarking NicheCompass across various scenarios. **c**, The mouse hippocampus tissue colored by niches identified through clustering of the embedding spaces of four similar methods. The color of clusters match with **a**. Below it, the dendrograms computed on average embeddings of the respective method, showing artifact niches and incomplete anatomical hierarchies. **d**, Six metrics across the two categories spatial consistency and niche coherence display the performance of each method on this dataset. The overall score is computed by min-max normalization and subsequent aggregation of the individual metrics into categories, followed by equal weighting of categories (see full version in **Fig. 3**).

== Other major concerns on the data analysis ==

8. In Fig. 3a, it seems that the assignment of the identified CA1 niche is unreasonable since it is connected with the identified V3 niche. However, in fact, from the Allen reference atlas (in Fig. 3a), these two niches are separated from each other, and other competitor methods successfully separate these two niches.

R1C9: Thank you for highlighting this important observation which we overlooked in our initial analysis. As mentioned in **R1C8**, we have now increased the clustering resolution and were able to separate the CA1 niche from another newly detected niche adjacent to the V3 niche. This new niche represents the Fasciola cinerea (see here). We have updated **Fig. 3a,b** accordingly.

Extract Fig. 3 | Benchmarking NicheCompass across various scenarios. **a**, A coronal mouse brain image from the Allen Reference Atlas showing anatomical subcomponents in the isocortex, hippocampal region,

thalamus, fiber tracts, and ventricular systems. Next to it, a mouse hippocampus tissue, sequenced with the SlideSeqV2 technology, highlighting corresponding niches obtained with NicheCompass (see full version in **Fig. 3**).

9. In Figure 3c, it seems like the intent is to show that other methods do not create clusters in the isocortex that are close together in their embedding space. This comparison seems to be done in the UMAP space, which is not quite meaningful. It is possible that given different UMAP parameters, the other methods would place the isocortex clusters all closer together. This analysis should be done in the original latent spaces, using the distance between cluster centers.

R1C10: Thank you for this constructive feedback. We agree with the suggestion that this analysis should be performed in the original embedding space. After changing our analysis approach as suggested by you and described in comment **R1C8**, other methods created clusters in the isocortex that were close together in the embedding space, which is why we have removed this claim entirely from the **Results** section. In addition, we have removed the UMAPs from **Fig. 3c** and, guided by your comment, have instead added the dendrograms from hierarchical clustering (using ward linkage) to better represent distances between clusters (based on embedding space). We have used these dendrograms to describe differences in the identified niches and their hierarchy (**Fig. 3c**, see also **R1C8**). While our original claim of the Isocortex niche separation did not hold anymore, we observed that NicheCompass was the only method that could recover all anatomical niches without spatially scattered artifact clusters while BANKSY and GraphST, the two closest competitors, could not recover a meaningful anatomical niche hierarchy based on the dendrograms. We have updated the **Results** section accordingly (**lines 335-337**). We are very grateful for this suggestion that has really improved our methodology.

10. In Figure 6, there is a comparison of the original clusters from the spatial ATAC-RNA dataset paper to the niches from NicheCompass. This comparison relies on the original clustering from the paper, which identified a specific number of clusters based on the parameters given to the clustering method. It would be much more meaningful to compare the actual underlying representations that were given to the clustering algorithm to compare the ability to identify these substructures without the influence of the clustering algorithm. It is possible that under different parameters, the original data was sufficient to separate the brain regions. Looking at the UMAP from the original paper, it does seem like the Corpus Callosum and Anterior Commissure can be easily separated and correctly identified.

R1C11: Thank you for this suggestion. We agree that this comparison was reliant on the original clustering from the paper. Based on this and congruent feedback from reviewer 3 suggesting that a separation of the highlighted niches does not require multimodal data (see **R3C1**), we have removed this part entirely from the **Results** section.

11. It is reasonable that highly heterogeneous tumor cells form their own niches (e.g., Fig 5a), but when the goal is to understand the microenvironment and immune cell infiltration, a user may be interested in tolerating more diverse tumor cells in a niche. How will NicheCompass deal with this?

R1C12: Thank you for this question. In our presented analysis on the NSCLC dataset, we do not believe that tumor cell heterogeneity prevents analysis and identification of shared microenvironment and

stromal structures between donors. In NicheCompass, we define niches as “communities of spatially localized cells with coordinated functions” (we have added this definition in the **Introduction** section in **line 53**). To achieve spatial colocalization of cells within a niche, we consider only immediate neighbors when building the neighborhood graph (**Methods**; $k=4$ in this case). Since tumor cells form tight aggregates in this dataset (see below and new **Supplementary Fig. 25**), tumor cells are mostly connected to other tumor cells. As a consequence, 92% of tumor cells were detected in tumor exclusive-niches, having relatively little impact in the definition of stromal niches (we have updated the **Results** section in lines **529-530** to better describe this). At the stromal level, only neutrophils have tumor cells within their closest neighbors. In this case, we do identify a tumor-infiltrating-neutrophils dominated niche with a relatively high percentage of tumor cells (niche 6; **Fig. 5e**; we have relabeled niche 6 as “*Tumor-infiltrating neutrophils*” to better highlight that it is composed by infiltrating neutrophils and also tumor cells (**line 532**)), and the related niche of tumor cells interacting with neutrophils (niche 3). As highlighted in the text, this niche is mostly composed of tumor cells from donor 9, but tumor cells from donor 12 that are close to neutrophils and produce higher levels of CXCL1 are also included in the niche (**Fig. 5d & Supplementary Fig. 26**). This suggests that gene expression of tumor cells is not so dominant that it impedes identifying equivalent spatial structures when they do exist. Although the niches are generally tumor-exclusive, when there is stroma infiltration, we identify in both the stroma niches and the tumor niches an over-activation of the relevant GPs (as described in the above case of the CXCL1 GP activated in infiltrating neutrophils and tumor cells, or the SPP1 GP activated in infiltrating macrophages and tumor cells). In this case, it is interesting to perform downstream niche-niche communication analysis to solidify these findings (as exemplified in **Fig. 5m**).

The boxplots show, per cell type, the distribution of the number of cells from each cell type in their spatial neighborhood (with a symmetric spatial neighborhood graph and $n=4$). All tumor cells have predominantly tumor cells as closest spatial neighbors.

Extract Supplementary Fig. 25 | Tumor niches neighborhood composition. Cell type composition in the spatial neighborhood of all cells in tumor niches 1 to 5, using a symmetric neighborhood graph with $c, 4$

neighbors. In this dataset, tumor niches consist of spatially segregated tumor cells, reflected by the identification of pure tumor niches where cells only have tumor cells in their spatial neighborhood. Boxplot elements are defined as: center line, median; box limits, upper and lower quartiles; whiskers, 1.5x interquartile range (see full version in **Supplementary Fig. 25**).

In addition, we want to highlight that if it were the case that tumor cells were spatially mixing with stromal cells and tumor cell gene expression differences would dominate niche identification, the user could, in downstream analyses separate tumor cells from stromal cells, and perform differential gene program testing for each cell type separately. With this approach the GP mediating tumor-stroma interaction would be embedded in our embedding space, but the tumor-specific GPs would not prevent shared GP identification in the stroma. We also want to note that, depending on the biological question of interest, the clustering resolution is an important parameter to identify coarser or more fine-grained niches, analogous to the identification of coarser and more fine-grained cell types described in single-cell analysis⁸. We have added this point in the **Discussion** section in lines **789-791**.

12. Line 527 (Fig. 5): A reason donor 6 (see Table S6 in the original publication) has very different niches might be because it is squamous cell carcinoma (all others are adenocarcinoma), so their surroundings (i.e. the sample collected) may be different in the first place. This is unclear.

R1C13: Thank you for pointing this out. We agree that this might play a role in differences in spatial organization. We have added a clarification in the **Results** section when introducing the dataset (**line 520**) and in the **Methods** section under the title **nanoString CosMx human non-small cell lung cancer**), as well as in the comparison between reference and query niches (**lines 623-624**).

13. Supp. Note 4: It appears to me that while NicheCompass scales to a large number of cells, more genes significantly impact the running time (note the log-scale y-axis in Supp. Fig. 17). Why is this the case?

R1C14: Thank you for this question, we appreciate this important observation. The number of genes significantly impacts the run time due to multiple factors:

- Only GPs are included for which genes are present in the dataset. As a consequence, when using the default prior GPs, a higher number of genes usually implies a higher number of GPs, which corresponds to a bigger model embedding size as each embedding feature represents a GP (inclusion of all GPs leads to an embedding space dimensionality of ~2000).
- Our model architecture has a fully connected layer in the decoder and an increasing number of genes not only increases the number of neurons in the output layer but also the number of connections between the hidden layer and output layer as GPs will consist of more genes.
- While the previous two points increase the number of model parameters and hence lead to slower run times per se, we have optimized the batch size for each model run to use our available GPU memory (40GB). As a result, we were able to use bigger batch sizes for datasets with fewer genes which led to a significant speed up.

We have added two panels to **Supplementary Fig. 20** to explore the impact of the number of genes on the number of GPs and number of model parameters, respectively. In addition, we have updated **Supplementary Note 6** with a summary of the above explanation.

c NicheCompass: Single Sample Model Sizes

Dataset	Number of Genes	Number of GPs	Number of Params
SlideSeqV2 Mouse Hippocampus	4000	1529	55802739
seqFISH Mouse Organogenesis	351	1368	8802846
MERFISH Mouse Liver	347	1388	8826986
Nanostring CosMx Human NSCLC	960	1325	23854930

d NicheCompass: Sample Integration Model Sizes

Dataset	Number of Genes	Number of GPs	Number of Params
seqFISH Mouse Organogenesis	351	1368	8804970
seqFISH Mouse Organogenesis (Imputed)	3000	1489	48939157
Nanostring CosMx Human NSCLC	960	1504	26958473

Extract Supplementary Fig. 20 | Benchmarking runtimes and model sizes. **c**, Overview of NicheCompass model sizes for datasets used for single sample method benchmarking, showing the impact of the number of genes on the number of GPs and model parameters. **d**, Same as **c** but for datasets used for sample integration benchmarking (see full version in **Supplementary Fig. 20**).

14. The manuscript overall is quite dense and can be significantly shortened to highlight the most important novelty. The figures are not made in high quality and are often difficult to read.

R1C15: Thank you for this important feedback. We have revised all figures to improve readability. To retain high quality and resolution, we have spent significant effort to export all graphics in vector format, have removed the PDF compression that we used for the original submission, and have inserted all figures into the final MS Word documents (as opposed to pasting them into Google Docs which caused the quality to degrade significantly).

Additionally, we have significantly shortened various segments of the manuscript without compromising on the breadth of our data exploration effort which was emphasized as setting this manuscript apart by reviewer 3 (see **R3C14**) :

- In our analysis on the seqFISH mouse organogenesis data presented in **Fig. 2**, we have removed various sentences that were not essential to deliver the main points.
- In our benchmarking experiments presented in **Fig. 3**, we have removed various segments describing differences between niches.
- We have removed vast parts of the analysis related to **Fig. 6**, and converted **Fig. 6** into **Supplementary Fig. 30**, also based on the feedback from reviewer 3 (see **R3C1**).
- Across the entire manuscript, we have removed information that is not essential to deliver our key messages.

We are grateful for this feedback and believe that the paper has become more accessible for the reader.

- In Line 122 of the Methods section, I found that the latent vector z_i , which should be the output of the graph encoder module, is not introduced in the graph encoder module part. I guess the latent vector z_i is just the variational posterior mean vector μ_i .

R1C16: Thank you, this is a very helpful and correct observation. We appreciate the detailed and attentive readthrough of the **Methods** section and have addressed this by adding a note in **lines 909-910** that introduces this equality.

- The y-axis labels for Fig 3f are somewhat confusing.

R1C17: Thank you for this comment. We have revised **Fig. 3f** and have added a “*Dataset*” label to make clear that the y-axis shows different datasets. We hope that this clarifies the confusion.

- Fig 4j isn’t clearly discussed in the main text. In general Fig 4f-j is a large number of panels and it would be helpful to label which sentence each of them corresponds to.

R1C18: Thank you for helping us improve the readability of our manuscript. We have now rewritten the corresponding sections and clearly referred to figure labels in **Fig. 4**. Specifically, we have:

- described in **lines 466-468** the Ptpcr combined interaction GP as enriched in the CD4+T niche, which has been described as a biomarker for cancer prognosis and as an interaction marker between breast cancer stem cells (BCSCs) and CD4+ T cells with a reference to **Fig. 4f**.
- used **Fig. 4g** to discuss *de novo* GP 37 in **lines 472-478**, and **Fig. 4h** to discuss *de novo* GP 86 in **lines 478-484**.
- addressed in lines **484-485** how to make the interpretation of *de novo* GPs more accessible, with a reference to **Fig. 4g,h**.
- described in **lines 486-490** and **Fig 4i,j** how the *de novo* GPs can be linked to histological structures.

Response to Reviewer 2

Remarks to the Author:

NicheCompass is a novel and promising tool for spatial niche mapping based on gene programs (GP), particularly GPs for cell-cell interactions. The biological assumption indicated in the manuscript is that nearby cells that express enriched gene programs for cell-cell interactions likely form a niche. This is a valid assumption.

Thank you, we highly appreciate that you acknowledge the validity of our method’s assumption and like your summary.

The paper presents a well-developed variational graph neural network model with several innovative components, including the multi-module decoder for preserving spatial neighborhood information, neighbor sampling for memory-efficient training, and weight-restricted fine-tuning for querying unseen data. NicheCompass models confounding factors and batch effects by including covariates vectors. This allows for the analysis of multiple samples and this is a very important feature that is needed in the field. The method has been thoroughly evaluated across multiple biological systems, technologies, and sample sizes, demonstrating its potential for diverse and broad applications. The manuscript is well-written and the technical information and formula are clearly described.

We are grateful that you emphasize NicheCompass’ innovative approach and important features, and are happy that you perceive the manuscript is well-written and the technical information clearly described. We agree that data integration of spatial omics data is a very important feature and together with our interpretable approach allows for batch-effect-free characterization of niches.

The detailed software documentation and tutorials, the well-organized code (including code to reproduce figures) are complemented. NicheCompass leverages functionalities in common software platforms such as AnnData, scanpy and squidpy and would be implementable by many users. This reviewer has installed the software and run through the key functionalities smoothly.

We highly appreciate the effort and time you spent installing the software and testing all key functionalities and are glad that everything ran smoothly. Reproducibility and use by the community are very important goals for us.

Major comments:

- As the paper emphasizes cell-cell interactions as a key input and focus of NicheCompass, the authors should discuss the limitations of using panel-based technologies which can only measure a subset of cell-cell interaction genes, missing many in the ligand-receptor gene databases.

R2C1: Thank you for this suggestion. We agree that this is a very important point to consider. We had shortly mentioned such a limitation in the **Discussion** section but have made it more prominent now in **lines 759-767**:

“Further, to fully utilize the potential of prior GPs from cellular interaction databases, it is necessary that datasets are not limited by small gene panels that only capture a subset of relevant genes like ligands and receptors. Many currently available experimental technologies are subject to such limitations or show uneven coverage of genes due to high sparsity and noise (as was the case in our spatial ATAC-RNA-seq mouse brain analysis). This can lead to poor performance in niche identification and GP inference. NicheCompass will hence benefit from expected experimental innovations leading to high-resolution, high-throughput, high-quality spatial readouts¹²⁵. Future database enhancements and the detection of new interaction pathways will further strengthen NicheCompass’ capabilities”.

We hope this modification adequately addresses your comment.

In addition, we want to point out that in cases where gene panels have limited availability of genes in our prior knowledge databases, NicheCompass can also rely on spatial *de novo* GPs, which we have introduced for this reason among other reasons (see **Results** section in **lines 121-124**). We have illustrated the usefulness of *de novo* GPs in the context of limited gene panels in our analysis on the Xenium human breast cancer data presented in **Fig. 4**.

- How would NicheCompass perform differently when applying for two types of data with different resolutions and number of genes. The first type is with hundreds of thousands of cells like Xenium and CosMX but with fewer genes. The second data type has few spots/grids, but more genes (e.g. Visium, DBiT).

R2C2: Thank you for bringing this up, we agree that this is an interesting question worth exploring and have addressed it in multiple ways. First, based on the suggestion of reviewer 3 (see **R3C11**), we have created a spot resolution version of our simulated data by binning gene expression counts into circular bins of diameter 55µm (as in 10x Visium). We then compared the niche identification and GP recovery metrics of NicheCompass models trained on this version of the data and the original (unbinned) version. The results are reported in **Supplementary Fig. 16** and can be compared with **Supplementary Fig. 14**.

Overall, while NicheCompass still shows good niche identification performance (albeit lower compared to the unbinned dataset), GP recovery performance is lower. This allowed us to have a direct comparison of the impact of resolution using the same underlying dataset. The results are expected since NicheCompass' conceptual idea of "self" and "neighborhood" modeling makes single-cell resolution data the primary use case. We have added a corresponding note in the **Discussion** section in **lines 768-769**.

Second, while we have covered many datasets with hundreds of thousands of cells and few genes throughout our analyses and benchmarking experiments, sequencing-based data with few spots/grids and more genes was limited to the SlideSeqV2 mouse brain analysis presented in **Fig. 3** and the spatial ATAC-RNA-seq multimodal analysis originally presented in **Fig. 6**, now in **Supplementary Fig. 30**. We have therefore introduced a new dataset from the SDMBench study³ in our benchmarking experiments (StereoSeq technology). We have trained multiple NicheCompass models as well as other models included in our benchmarking study on this dataset and have computed the niche prediction accuracy metrics proposed in SDMBench (as this dataset has ground truth niche labels) as well as the NicheCompass metrics. The results are presented in **Supplementary Fig. 16** and show that NicheCompass is also among the top-performing methods on this spot-resolution StereoSeq dataset.

Supplementary Fig. 16 | Additional ablation and benchmarking on spot-level resolution data. **a**, Ground truth niches of the single-cell resolution simulated data that has been binned into spots with 55µm diameter. **b**, Distribution of the number of cells per bin/spot. **c**, Niches identified by a NicheCompass model with a misclassification at the periphery of Niche 8. **d**, Niche identification and GP recovery metrics on the binned simulation data for different encoder layer and neighborhood graph KNN hyperparameter combinations. For each combination, eight models have been trained. Boxplot elements are defined as: center line, median; box limits, upper and lower quartiles; whiskers, 1.5x interquartile range. **e**, Ground truth niches of the spot-level resolution Stereo-seq mouse embryo dataset. **f**, Mean NicheCompass and ground truth prediction metrics across eight training runs for each method, ranking NicheCompass first and second, respectively. STACI did run out of memory on our 40GB GPU and DeepLinc did not converge; both were hence excluded. The overall and ground truth prediction scores are computed by aggregating the min-max-normalized individual metrics.

Third, we have included a new analysis where we integrate datasets across different technologies, specifically STARmap PLUS and MERFISH (see also **R3C12**). In this case we observed that NicheCompass was able to integrate data from these two experimental modalities and identified congruent anatomic niches across both datasets (**Supplementary Fig. 36**).

Supplementary Fig. 36 | NicheCompass integrates tissue samples across different spatial transcriptomics technologies. **a**, UMAP representation of the NicheCompass GP embedding space after integrating the MERFISH mouse brain and STARmap PLUS mouse CNS datasets, colored by dataset/sequencing technology. **b**, Composition of niches in terms of cells from each of the two technologies, showing that all niches except niche 9 were present in both datasets. Only niches with more than 100,000 cells are displayed. **c**, Two example tissues of the same brain region, one from the MERFISH mouse brain dataset and the other one from the STARmap PLUS mouse CNS dataset, highlighting consistent anatomical niches. **d**, Zoom in on four specific niches that emphasize the consistency in niche identification across technologies.

Finally, we want to note that we do not recommend an integration of modalities from very different resolutions such as Xenium and 10x Visium as the underlying data distributions are too diverse to be efficiently captured by NicheCompass.

- The authors should provide more detailed information on the model training process and its sensitivity to data size and composition. The authors should clarify/discuss: How much data is required to optimize model parameters? For example, were all three embryo tissues used for training the model in Figure 1 and would the model trained on two samples be able to map niches as expected when tested on a third, leave-out sample?

R2C3: Thank you for this valuable suggestion. For the original analysis, we trained NicheCompass on all three embryo tissues. However, we have now performed an experiment on the mouse embryo data where we trained two NicheCompass models on only two of the embryos while leaving one embryo sample out (**Supplementary Fig. 8c,d**).

Indeed, we observed that the identified niches and inferred GP activities in embryo 2 were similar between a scenario where embryo 2 was left out during reference training and mapped as query compared to the model training on the full dataset and the model training with leaving embryo 3 out.

Extract Supplementary Fig. 8 | Niche and GP inference reproducibility and generalizability. c, Embryo 2 niches identified by NicheCompass when leaving out embryo 3 during reference model training. Next to it, the inferred GP activity for the characterizing GPs from the main analysis in **Fig. 2. d,** Same as **c** but when leaving out embryo 2 during reference model training. Overall, there is high robustness of inferred niches and GP activities providing evidence for generalizability (see full version in **Supplementary Fig. 8**).

We also want to highlight that, as part of our benchmarking efforts, we have already tested the sensitivity to and impact of different data sizes (**Supplementary Fig. 18,19**). In the smaller data regime, methods such as CellCharter and BANKSY work better than NicheCompass, which we indicated in **Supplementary Note 5**. This can be explained by the data-hungry nature of deep learning-based models.

In addition, we have now performed an analysis of the seqFISH mouse organogenesis data in a spatial reference mapping scenario by sequentially removing parts of the reference while retaining embryo 3 as query. Specifically, we have evaluated the niche identification and integration performance when including only embryo 2 or 50%, 25%, 10%, 5% and 1% of the embryo 1 and 2 data as reference. We observed that performance continuously decreases with lower reference size with a bigger performance drop when the reference becomes smaller than the query.

Extract Supplementary Fig. 38 | seqFISH mouse organogenesis spatial reference mapping. b, Metrics from the scenarios in **a** per number of cells in the reference. NMI significantly reduces at a size of ~80,000 reference cells (see full version in **Supplementary Fig. 38**).

- For the NSCLC analysis, would the authors suggest that the training using 4 donors with two replicates could train a model generalisable enough to find niches in unseen data with high inter-sample heterogeneity? If so, additional testing on independent dataset would be needed. Could the authors discuss about the power calculation and sample size?

R2C4: Thank you for this comment. We have performed additional testing on the seqFISH mouse organogenesis dataset. While samples in this dataset are quite homogeneous, varying positioning of the tissue slices (**Supplementary Fig. 2**) provided an ideal scenario where one niche (Presomitic Mesoderm) was only present in Embryo 1 and 3 but not in Embryo 2 (**Fig. 2a**). We therefore tested whether this niche could be successfully detected with a reference that only included Embryo 2. We found that this was indeed possible, providing evidence that niches unseen in the reference can be successfully detected as novel niches (**Supplementary Fig. 38a,c**). Moreover, the low prediction probability of our niche label transfer kNN classifier highlighted that the model was uncertain when performing niche label transfer to cells in this niche. This means that this niche would also be highlighted in a scenario of niche label transfer when the reference data is not accessible but only a kNN classifier trained on the reference data.

Nonetheless, we also observed that a characterization of this niche through the characterizing GPs from the full analysis presented in **Fig. 2** was not possible as these GPs and their important member genes were only relevant in this niche and hence did not play a role during reference training. For effective characterization, it is therefore important that a reference atlas is comprehensive and comprises diverse niches with shared niche-relevant GPs compared to the query. We have added this as a limitation in the **Discussion** section (**lines 777-781**).

In addition, as mentioned in the previous comment (see **R2C3**), we have performed a power analysis on this dataset, subsequently leaving out a bigger number of cells while comparing the integration performance and niche identification performance based on the niche labels from the full analysis. Here, we observed that an increasing number of cells in the reference led to consistently better performance with a significant drop when the reference dataset was smaller than the query (**Supplementary Fig. 38a**).

Extract Supplementary Fig. 38 | seqFISH mouse organogenesis spatial reference mapping. a, Power analysis using different dataset proportions of the mouse embryos 1 and 2 as reference while holding out embryo 3 as query. Embryo 3 is mapped onto the reference using weight-restricted fine-tuning (**Methods**). UMAPs represent the integrated GP embedding space. BLISI quantifies the integration performance. Label transfer from reference to query is performed via a kNN classifier trained on the reference. The prediction probability of this kNN classifier quantifies uncertainty in niche label transfer. The NMI quantifies niche prediction performance based on niche labels from the full analysis in **Fig. 3. c**,

Comparison of niche detection of the Presomitic Mesoderm niche in scenarios 1 and 2. In Scenario 1, this niche is seen in the reference, and we recover the same characterizing GPs as in the analysis on the full dataset, supported by expression of the respective ligand genes. In Scenario 2, this niche is not seen in the reference, yet it is detected as a novel niche; however, the same GPs could not be recovered as these GPs were not relevant during reference training (see full version in **Supplementary Fig. 38**).

- The GP pruning step is an interesting feature that reduces the dependency on specific GP selection. However, further analysis of the effects of using different prior GP sets (e.g., ligand-receptor vs. metabolite-sensor vs. TF-target GPs) on niche identification would be useful to provide insights into the relative importance of different interaction types. The authors should also discuss whether using more GPs or combined interaction GPs improves niche identification.

R2C5: Thank you for this constructive suggestion. We have performed experiments on the seqFISH mouse organogenesis data in which we have trained multiple NicheCompass models while using different subsets of prior GP sets (ligand-receptor GPs only, metabolite-sensor GPs only, transcriptional regulation GPs only, and combined interaction GPs only). We have evaluated the impact on niche identification performance quantitatively using the metrics from our benchmarking suite as well as qualitatively by visualizing the niches in the tissue. The results are reported in **Supplementary Fig. 10**. We observed that niche identification is robust across different sets of prior GPs; however, different biology can be elucidated based on the used set of prior GPs. Users should thus select prior GPs according to the biological question of interest. We have added this to our user recommendations in **Supplementary Note 5**.

Extract Supplementary Fig. 10 | Niche and GP inference with different prior GP sets. a, Brain and gut niches identified when only using a specific set of prior GPs. All niches from the analysis in **Fig. 2** can be recovered independently of the used set of prior GPs; however, different prior GPs elucidate different aspects of underlying niche biology, highlighted with a characterizing GP and its most important gene in the Midbrain niche (see full version in **Supplementary Fig. 10**).

- While the chosen performance metrics are comprehensive and appropriate, giving equal weight to spatial conservation scores and niche coherence scores might not be optimal in all scenarios. The authors should consider discussing the potential for weighting these metrics differently depending on the specific biological question and the relative importance of spatial structure versus cell-cell interaction signatures.

R2C6: Thank you for this idea. We have added this information in **Supplementary Note 4**, which describes the metric suite in detail (**lines 431-433**).

- The paper should state whether batch correction was applied to the data in Figure 5c, which shows a large separation of cells by samples.

R2C7: We apologize that this was not clearly stated. We have now clarified in **lines 525-526** that we include covariates in NicheCompass training to reduce batch effects. However, as discussed in **lines 533-535**, given the high degree of tumor heterogeneity, particularly in terms of tumor cell transcriptional profiles and microenvironment organization and composition, a complete integration would mean removing the biological signal. In line with this, good integration is observed across technical replicates, where the biological signal is shared. This should give good confidence in the integration.

- To increase confidence in the cell type and niche mapping results, the authors could consider incorporating annotation and assessment by pathologists and visualising more gene markers. For example, the Spp1 Macrophage cluster would need a plot to show the distinct expression of Spp1 and macrophage markers for this cluster.

R2C8: We apologize if the annotation process was unclear. Cell type annotations are taken from the source dataset (as mentioned in **lines 1080-1081** of the **Methods** section). Niche annotation is based on cell type composition as well as spatial organization. However, to increase confidence in the annotations and clarify the process, for stromal niches we have substituted **Supplementary Fig. 22f** (which showed the top 5 differentially expressed genes, which we realize was not relevant for niches with mixed cell types) with new **Supplementary Fig. 24**, summarizing the cell type composition of that niche, and showing the expression in that niche of key markers for each cell type. Since tumor niches are almost exclusively constituted by homogeneous tumor cells, for them we still conserved **Supplementary Fig. 22f**, showing top differentially expressed genes.

Supplementary Fig. 24 | Characterization of stromal niches. Each row represents a niche. The bar plots on the left represent cell proportions for the most abundant cell types in that niche (i.e. more than 10% of the cells in the niche). The color and length of the bar is proportional to the cell abundance in the niche. The heatmaps show mean expression of selected gene markers across cell types in each niche separately, with color representing mean gene expression. Shown are selected marker genes per cell type that are differential in that cell type compared to the rest, considering all the niches together. Indicated at the top are the cell types represented by each set of markers.

Regarding SPP1+ Macrophages, for which our transcriptional characterization goes beyond the original publication's annotation, we previously showed and described that both the gene and its receptor are over-expressed in the macrophage and the associated tumor niche, supporting this communication pathway. Unfortunately, based on the feedback of an expert pathologist that we consulted, it is not possible to distinguish this macrophage phenotype from others in histological samples. To address your comment, we have now added an additional **Supplementary Fig. 29**, showing that these macrophages have a profibrotic phenotype, as extensively reported in literature (we have added multiple references to the manuscript), and express known markers of this subpopulation (**lines 631-643**).

Extract Supplementary Fig. 29 | Characterization of infiltrating macrophages in Donor 13 as SPP1+ macrophages. a, UMAP highlighting the macrophages in the integrated GP embedding space, colored by cell type, data source and niche as identified by NicheCompass, **b**, Expression levels of genes characteristic of SPP1+ macrophages and its profibrotic phenotype, significantly overexpressed in donor 13 macrophages compared to macrophages from other donors (see full version in **Supplementary Fig. 29**).

- Some *de novo* GPs do not show a clear pattern (e.g., Sfrp1 GP as in Figure S30). Could the authors suggest down-stream analyses or post-hoc tests to gain the confidence of the *de novo* GPs predicted by NicheCompass?

R2C9: Thank you for highlighting this. We have removed **Supplementary Fig. 30** based on this feedback and the feedback of reviewer 3 (see **R3C1**). We want to highlight that one way to gain confidence in the identified *de novo* GPs is to filter them based on correlation with the expression of important member genes. We define GPs with a high correlation as characterizing GPs as described in the **Methods** section under the title **Selection of characterizing gene programs for each niche**. We used this procedure to determine characterizing prior GPs in our analyses but it can be equally applied to *de novo* GPs.

In addition, we have added a capability to the NicheCompass package to visualize GPs in the biological context based on gene weights of member genes learned by NicheCompass (**Fig. 4g,h**). This should help users to evaluate the biological significance of *de novo* GPs.

Extract Fig. 4 | NicheCompass identifies meaningful niches and *de novo* gene programs in human breast cancer. Sunburst plots of GPs and their member genes for *de novo* 37 GP (**g**) highlighting the importance of the keratin gene family and the involvement of an uncharacterized gene (*C5orf36*), and *de novo* 86 GP (**h**) displaying a more complex KRT8-driven GP that includes elements of fatty acid metabolism (*FASN*, *ABCC1*), and *ELF3* as a potential gene regulator. The scale represents the gene weights inferred by NicheCompass (see full version in **Fig. 4**).

Minor comments:

- The authors should provide a clear definition of a tissue niche at the beginning of the paper.

R2C10: Thank you for this valuable suggestion. We have added a clear definition of a tissue niche in the introduction (**lines 53-55**: “niches, defined as communities of spatially localized cells with coordinated functions. Such niches can consist of homogeneous or heterogeneous cell populations and are mainly shaped by local cell interactions”). We hope you find this definition reasonable.

- Lines 74, 790: The claim that NicheCompass is the first work to characterize niches via cell-cell interactions needs to be revised, as other methods to characterize cell-cell interaction in the tissue microenvironment exist.

R2C11: We apologize for missing this and have revised the mentioned text segments to not accidentally undermine any existing methods (**lines 74, 718**: “...a graph deep learning approach” instead of “the first approach”). We also want to emphasize that, in contrast to existing methods, our method explicitly considers cell interactions in the identification of a niche vs “mere” post-hoc characterization of previously identified niches. We have added a new analysis on our simulation data that shows that this approach can better recover ground truth GPs than alternative workflows relying on chaining together niche identification tools and niche characterization tools.

Extract Supplementary Fig. 14 | Data simulation. f, F1 scores between inferred and ground truth upregulated GPs for three different workflows to infer niche-specific GPs. Displayed are eight training runs with different random seeds and a kNN neighborhood graph with k=6 (the ground truth cell interaction range). Boxplot elements are defined as: center line, median; box limits, upper and lower quartiles; whiskers, 1.5x interquartile range. NicheCompass considerably outperforms alternative methods, providing evidence that it is useful to integrate GPs during model training (see full version in **Supplementary Fig. 14**).

- The marker plot for the niches identified in small cell lung cancer samples (Figure 22f) would need more revision because some groups are not convincingly distinct (e.g. the lymphoid rich niche 10).

R2C12: Thank you again for noticing the annotation process was not clear enough. As explained in **R2C8**, we have improved this by substituting **Supplementary Fig, 22f** for **Supplementary Fig. 24** for stromal niches.

- It would be useful for users that the authors discuss on whether NicheCompass Light is recommended. The results shown in Figures S15-S17 show similar performance compared to the NicheCompass with attention weights.

R2C13: Thank you for this constructive comment. Besides the existing benchmarking experiments, we have included a comparison between NicheCompass and NicheCompass Light on our simulated data (**Supplementary Fig. 14**). While NicheCompass Light can achieve similar and sometimes even better niche identification performance compared to NicheCompass (**Supplementary Fig. 14-19**), NicheCompass showed better performance in the recovery of GPs (**Supplementary Fig. 15**). Since the GP-based characterization of niches is the main feature of our method, we would recommend NicheCompass in most scenarios. However, we would recommend NicheCompass Light if runtime is a bottleneck. We have added such a discussion in **Supplementary Note 5**.

Extract Supplementary Fig. 15 | NicheCompass ablation. Niche identification (NID) and GP recovery (GPR) metrics on the simulated data under different model design choices and hyperparameters. The first row in each panel contains metrics that measure the correspondence between predicted and ground truth niche labels, the second row contains F1 scores between enriched and artificially incremented GPs and member genes in ground truth niches. Boxplot elements are defined as: center line, median; box limits, upper and lower quartiles; whiskers, 1.5x interquartile range. The red dotted line indicates the baseline of mean F1 scores between the artificially incremented GPs and member genes and randomly selected candidate GPs and genes (the number was matched). Ablations were performed on **c**, the encoder architecture, indicating that NicheCompass Light performs better on NID while NicheCompass has superior GPR (see full version in **Supplementary Fig. 15**)

Response to Reviewer 3

Remarks to the Author:

A. Summary of the key results:

In this manuscript, the authors describe NicheCompass, a computational framework and software package for identifying and characterizing cellular niches in spatially resolved omics measurements. NicheCompass' performance is described through application to several relevant datasets and comparison to related analysis packages. An admirable effort has been taken to apply NicheCompass to publicly-available datasets and interpret the results in biological context.

Thank you for this very kind and nice summary and for appreciating our analysis efforts.

B. Originality and significance:

This work presents a novel framework for integrating spatial domain detection with gene program prior information. Importantly, their implementation scales to handle large datasets in challenging configurations that will have broad applicability to large spatial transcriptomic datasets.

Thank you very much for pointing out the scalability and broad applicability of our method.

C. Data and Methodology:

The incorporation of publicly-available data and its subsequent processing and presentation is generally excellent. Because of this, the analysis of the mouse spatial ATAC data greatly weakens the manuscript:

As the authors point out, the major island of Calleja is readily distinguished from the surrounding tissue by its expression of the D3 dopamine receptor. Expression of this one (one!) gene is a good marker for the Islands of Calleja, so discrimination of this region from the nearby structures does not require more data beyond gene expression (as the authors suggest), but rather less and/or better data.

R3C1.1: Thank you very much for describing our general analysis approach as excellent. We highly appreciate your expert assessment in the context of the spatial ATAC-RNA-seq mouse brain data and have therefore removed all statements that suggest that a discrimination of this region benefits from or requires multi-modal integration from the manuscript.

Unfortunately, the exact placement of the dataset used shows only the major Island of Calleja and not the others in the ventral striatum that may have helped the authors recognize the structures in the context of niche detection. Supplemental Figure 28 shows likely root cause of this confusion: extremely sparse detection of *Drd3* (contrast to Allen ISH atlas or recent BICCN MERFISH datasets). This may help explain why the rest of the section shows rather haphazard correspondence to anatomical structures, leading me to conclude that the interesting results from the white matter structures and Islands of Calleja were the exception, rather than the rule.

R3C1.2: Thank you for pointing this out. Our understanding of your assessment is that in this case this is not a methodological problem of NicheCompass but rather a data quality problem of the publicly available data. Based on your guidance, in the **Results** section of the manuscript we now caution the readers that *“Notably, this dataset suffered from sparse detection of well-known marker genes (Supplementary Fig. 30a), posing a significant challenge for analysis” (lines 659-660)*, and have further removed vast parts of the analysis as described below and aligned with feedback from reviewer 1 (see **R1C14**) to shorten the manuscript.

The distinction between the anterior commissure and corpus callosum is similarly questionable- all of the biological interpretation of enriched GPs confirming NicheCompass performance (e.g. starting at line 709) surely also apply to the same cell types doing the same thing in the anterior commissure. Given the poor detection of *Drd3* in a similarly sized structure and the lower detection of myelin basic protein (*Mbp*) transcripts in the anterior commissure vs corpus callosum in Supplemental Figure 31, I suspect that differences between these two structures is primarily due to poor detection efficiency. Furthermore, Figures 6f and S31 all show GPs present in both white matter tracts, a fact that is glossed over in the text, which says that these GPs are “meaningful in the context of the Corpus Callosum niche”. Figure S31 shows something quite different from that: these GPs are common to both white matter tracts.

R3C1.3: Guided by your comment, we have removed this part of the analysis entirely from the manuscript. We agree that in this case it is hard to differentiate between real biological differences and differences due to poor detection efficiency.

These problems and the poor correspondence between other niches and anatomy in Fig 6 leads me to conclude that NicheCompass applied to this dataset produced a few reasonable results that were picked for further analysis. I suggest that this dataset and analysis only be included as cautionary tale in the supplemental section, illustrating the limits of NicheCompass in the face of sparse detection.

R3C1.4: Thank you for this constructive conclusion. We apologize that this analysis did not match the quality of the other analyses that we have presented. Our interpretation of your conclusion is that there seem to be major problems directly linked to the quality of the underlying dataset. We have decided to follow your expert suggestion and have removed vast parts of the analysis and **Fig. 6** entirely from the manuscript. Instead of focusing on a delineation between the Corpus Callosum and Anterior Commissure niches and claiming that multi-modal integration enabled fine-grained niche identification in this specific dataset, we now reduced the analysis to barely show that NicheCompass has a general capability for multi-modal characterization of niches.

In addition, we picked up your point of sparse detection as a limit of NicheCompass in the **Discussion** section in **lines 761-766**:

“...Many currently available experimental technologies are subject to such limitations or show uneven coverage of genes due to high sparsity and noise (as was the case in our spatial ATAC-RNA-seq mouse brain analysis). This can lead to poor performance in niche identification and GP inference. NicheCompass will hence benefit from expected experimental innovations leading to high-resolution, high-throughput, high-quality spatial readouts”.

We hope that our revised analysis addresses all your points and is now a valuable contribution to the manuscript.

D. Appropriate use of statistics and treatment of uncertainties

The authors do include some assessment of variability in their benchmarking against other computational approaches, but in other parts of the manuscript, this is absent. Are the NicheCompass results that are presented in biological context robust against the variation in neighborhood size discussed in the benchmarking? See comment 2 in F below.

R3C2: We appreciate this valuable suggestion. We have included an ablation of the neighborhood size and its effect on the recovery of ground truth gene programs in our new ablation experiments on simulated data, described in **R1C1**. Specifically, in these ablation experiments, we have performed eight runs with different random seeds across different neighborhood sizes, respectively. We observed that GP recovery performance improves with increasing k while niche identification performance decreases with increasing k (**Supplementary Fig. 15** and **R1C1**). We have also added these findings and recommendations in the user guidance provided in **Supplementary Note 5**.

Extract Supplementary Fig. 15 | NicheCompass ablation. Niche identification (NID) and GP recovery (GPR) metrics on the simulated data under different model design choices and hyperparameters. The first row in each panel contains metrics that measure the correspondence between predicted and ground truth niche labels, the second row contains F1 scores between enriched and artificially incremented GPs and member genes in ground truth niches. Boxplot elements are defined as: center line, median; box limits, upper and lower quartiles; whiskers, 1.5x interquartile range. The red dotted line indicates the baseline of mean F1 scores between the artificially incremented GPs and member genes and randomly selected candidate GPs and genes (the number was matched). Ablations were performed on **d**, the neighborhood size indicating better NID and retrieval of *de novo* genes for lower *k* and better GPR for higher *k* (see full version in **Supplementary Fig. 15**).

In addition, to evaluate a specific biological context, we have trained two NicheCompass models with varying neighborhood sizes on the imputed seqFISH mouse organogenesis dataset presented in **Fig. 3** (**Supplementary Fig. 9a,b**). We have compared the identified niches and GPs qualitatively with those from **Fig. 2** and observed high alignment between identified niches and inferred GP activities, with significant differences restricted to the GPs identified in the Hindbrain niche.

Extract Supplementary Fig. 9 | Niche and GP inference neighborhood graph robustness. **a**, Embryo 2 niches and GP activities inferred by NicheCompass with a longer-range kNN neighborhood graph ($k=12$). Displayed are characterizing GPs from the main analysis in **Fig. 2**. **b**, Same as **a** but with a shorter-range

kNN neighborhood graph (k=4). Missing GPs were filtered by GP pruning in the respective model. Overall, there is good robustness of inferred niches and GP activities across neighborhood graphs; however, there are also minor differences, most pronounced in the Hindbrain niche (see full version in **Supplementary Fig. 9**).

Taken together, these results indicate good robustness of the identified niches across different neighborhood sizes. The major GPs driving niche identity presented in biological context largely remain consistent with different K while the first analysis indicates this might not be the case across all GPs. To conclude, we would still recommend the user to carefully pick K based on the expected range of interactions in the tissue. We describe this in the user guidance in **Supplementary Note 5**.

E. Conclusions:

There are some biological results presented, but it's unclear to me exactly which biological conclusions are exclusively available through NicheCompass versus simpler analyses. In this way, some aspects of NicheCompass' design and implementation have questionable value. I have included a few specifics below.

R3C3: Thank you, we agree that this is an important aspect. To address it, we want to mention various points.

(1) We have now added quantitative ablation experiments on simulated data to evaluate all model design choices and hyperparameters of NicheCompass rigorously under the aspects of ground truth niche identification performance and recovery of ground truth gene programs (see **R1C1**).

(2) We also used this simulated data to compare NicheCompass' inherent niche characterization with two alternative workflows that consist of chaining together non-interpretable niche identification and characterization tools. We observed that NicheCompass could better recover ground truth GPs/signaling events compared to the best alternative niche identification method, BANKSY, followed by gene set enrichment analysis or cell-cell communication inference with LIANA. Details for the analysis can be found in the **Methods** section under the title **GP inference comparison**.

Extract Supplementary Fig. 14 | Data simulation. f, F1 scores between inferred and ground truth upregulated GPs for three different workflows to infer niche-specific GPs. Displayed are eight training runs with different random seeds and a kNN neighborhood graph with k=6 (the ground truth cell interaction

range). Boxplot elements are defined as: center line, median; box limits, upper and lower quartiles; whiskers, 1.5x interquartile range. NicheCompass considerably outperforms alternative methods, providing evidence that it is useful to integrate GPs during model training (see full version in **Supplementary Fig. 14**).

At the same time, we acknowledge that individual simpler analyses might reveal some common conclusions. For instance, cell-cell communication tools might recover some of the same interactions that have a clear gene expression signature. We used such clear interactions to provide evidence that the NicheCompass model architecture is working as intended. However, it is especially the strength of NicheCompass to combine the inference of signaling events with niche identification and do this at scale by integrating diverse tissues, via query-reference mapping, and potentially in multimodal scenarios, all in one tool. As shown in our extensive benchmarking efforts and with the newly added simulated data (**Supplementary Fig. 14**), NicheCompass is able to outperform existing tools for niche identification while at the same time providing superior niche characterization capabilities with minimal extra effort.

Moreover, we have provided further specific answers below.

F. Suggested Improvements.

Beyond the issue of the mouse ATAC data above, I have 2 main concerns about this manuscript and a few technical questions for the authors that may lead to clearer text:

1. The NicheCompass approach is presented as if there are no alternative approaches to arrive at the biological conclusions. At the same time, it's unclear which aspects of NicheCompass contribute to the final quality and interpretability of the results.

R3C4: Thank you for this constructive feedback. To more clearly outline which aspects of NicheCompass contribute to the final quality and interpretability of the results, we have performed extensive ablation studies on different model components using simulated data with ground truth niches and ground truth gene program activity (see **R1C1 & Supplementary Fig. 15**). We have evaluated these experiments with metrics for niche identification as well as GP recovery.

We included the following design choices and hyperparameters in our ablation study:

8. Edge reconstruction and gene expression reconstruction loss weights
9. Gene expression decoder weight regularization loss weights
10. Encoder architecture
11. Neighborhood graph size
12. Number of *de novo* GPs
13. GP selection via active GP threshold (GP pruning)
14. Use of prior GPs versus fully connected decoder (i.e. uninterpretable embeddings)

We found that:

8. Finding a balance between gene expression and edge reconstruction is a key element for good niche identification (NID) and GP recovery (GPR) performance.
9. Regularization of *de novo* GP weights is essential for GPR and especially the correct retrieval of important *de novo* genes.

10. The NicheCompass Light variation with a GCN layer could achieve better NID on this dataset while the GATv2 layer led to better GPR.
11. A smaller number of neighbors was more efficient in NID while a larger number of neighbors facilitated GPR.
12. The inclusion of *de novo* GPs was crucial for GPR with a default number of 100 *de novo* GPs leading to the best results.
13. GP pruning slightly improved NID and GPR while reducing the embedding size.
14. Comparable NID performance was achievable without prior GPs but this came at the cost of GP gene recovery preventing an inherent niche characterization.

In addition, as described in **R3C3**, NicheCompass' inherent niche characterization can more accurately recover ground truth signaling events in our simulated data.

In their own examples, the authors end up referring to the spatial distribution of known marker genes as an achievement of NicheCompass. In these cases where there are small sets of marker genes that readily define a spatial region or subset of cells, what exactly is NicheCompass bringing to the table beyond a gene-expression-based connection to a GP database? This kind of issue is mentioned in passing in the Supplementary Notes (“...necessitating future method development to investigate the disentanglement of intrinsic and spatially-induced variation.”), but this topic is a critical aspect of the work that should be addressed quantitatively or discussed (see below).

Examples of specific comparisons that could be quantitatively assessed:

Do GPs have to be integrated during training?

R3C5: Thank you for this feedback, we agree that this is a very important question to explore. As part of our ablation studies on simulated data (see **R1C1**), we have therefore trained a version of NicheCompass without the use of prior GPs (i.e. a model that uses fully connected layers and hence does not incentivize the embeddings to represent specific spatial gene programs). One important finding was that, on our simulated data, the use of prior GPs was not necessary to reach improved niche identification performance but prevented inherent niche characterization. As illustrated in **R3C3**, the use of prior GPs during training leads to better GP recovery compared to post-hoc characterization workflows.

We also want to highlight that the use of prior GPs enables a batch-effect-free characterization of signaling events across tissue samples through differential testing in the embedding space. This would not be possible through mere differential gene expression analysis between niches post-hoc. In addition, methods that rely on external data integration via for example Harmony, such as BANKSY, show suboptimal niche identification performance in data integration settings which we show in our benchmarking experiments (**Fig. 3**). Including prior GPs during data integration therefore allows for end-to-end analysis and niche characterization.

In the end, space of prior GPs is whittled down to “...two characteristic GPs per niche based on correlation between GP activities and the expression of the GP's important target genes and ligand and receptor or enzyme and sensor genes“. Could this same exercise determine the “characteristic GPs” of GP-agnostic spatial domains?

R3C6: Thank you for this question, we hope we interpret it correctly. Our model design does not enforce spatial co-expression of ligand and receptor or target genes (as gene pruning might turn either source or target genes off if co-expression is not given in the dataset). As a result, many GPs only pick up individual or a few co-expressed target genes. We have added a new section in the **Discussion** section to describe this (**lines 746-748**). For this reason, we have introduced the filter of characterizing GPs to enforce that co-expression of source and target genes is given and thus false negative signaling events are largely filtered. However, it would be entirely possible to characterize spatial domains/niches that are agnostic of signaling events through GPs that pick up, for example, only marker genes instead of signaling events. This could, for instance, be achieved by computing correlation between GP activity and individual target genes and determine characterizing GPs in this way. We also want to highlight that it is necessary to keep such GPs during model training to obtain superior niche identification performance as signaling events are not the only driver of niche identity. In this context, we also want to clarify that niche identification with NicheCompass relies on the full embedding space of prior GPs (and not only characterizing GPs). We have selected characterizing GPs only to perform niche characterization with a higher likelihood for true positive signaling events.

We have updated the **Results** section in **lines 215-221** to better describe that we use the full set of GPs for clustering and that while characterizing GPs have clear niche signatures, they are not sufficient to obtain optimal niche identification performance:

*“To investigate the preservation of global spatial organization by NicheCompass, we applied hierarchical clustering on the GP embeddings and grouped individual niches into higher-order functional components (**Methods; Fig. 2e**). This revealed a higher-level tissue organization consistent with anatomy, with the Midbrain, Forebrain, Floor Plate, Hindbrain and Spinal Cord niches constituting a CNS-cluster, and the Dorsal and Ventral Gut niches forming a Gut Tube cluster (**Fig. 2e**). The activities of the subset of characterizing GPs of each niche further supported this hierarchy, differentiating individual niches (**Fig. 2f**)”.*

In the reduced space of final GPs, how many genes are represented? What is the overlap between these genes and highly-variable genes from non-spatial clustering of the same data?

R3C7: Thank you for your comment. To answer this question, we have conducted an analysis on the NicheCompass model trained on the seqFISH mouse organogenesis dataset presented in **Fig. 2**. (1) For all enriched GPs and each of the two characterizing GPs in each niche, respectively, we identified the top 10 genes based on their importance. (2) We computed the overlap between these top GP genes and differentially expressed genes (DEGs) from non-spatial clustering. To determine DEGs, we followed a standard preprocessing workflow including normalization and dimensionality reduction of the raw gene expression with PCA, followed by Leiden clustering. We then considered the top 10 and top 20 DEGs for each cluster. (3) We compared the aggregated set of DEGs as well as a random baseline (matching in number with DEGs and selected across 10 random seeds) with the top GP genes and found that while there is an enrichment of DEGs within top GP genes, many top GP genes cannot be recovered from non-spatial analysis. This highlights a value-add of NicheCompass over non-spatial analysis.

Supplementary Fig. 37 | Overlap between GP genes and differentially expressed genes (DEGs) from non-spatial analysis in the seqFISH mouse organogenesis analysis. Venn diagrams illustrate the overlap between GP genes with an importance score greater than 0.05 and the aggregated top DEGs after clustering the PCA-reduced gene expression space. **a,b**, Overlap between enriched GP genes (log Bayes factor > 2.3) and top 10 DEGs (**a**) or top 20 DEGs (**b**). **c,d**, Overlap between characterizing GP genes and top 10 DEGs (**c**) or top 20 DEGs (**d**). Additionally included is a baseline of randomly selected genes across ten random seeds, matching in number the top DEGs. Number labels in the Venn diagram indicate averages across the ten random seeds. While the average overlap of randomly selected genes with GP genes is significantly lower than the overlap of DEGs with GP genes, many GP genes are not retrieved with non-spatial analysis.

We have also added a corresponding discussion around this topic in the **Discussion** section in **lines 750-754**.

2. The paper claims to produce a “quantitative definition of a niche”, but such a thing is difficult to find in the manuscript. For example, the Methods section pronounces that the underlying graph G “... can be defined with a fixed neighborhood-radius threshold if different numbers of neighboring observations ... are desired. Equally, a weighted version of G ... can be used to reflect differences in Euclidean distances

between neighboring observations.” This kind of flexibility is intriguing- what is the impact of these kinds of changes? Are the results quantitatively identical? I strongly doubt it. Similarly, spatial domain detection methods produce sometimes surprising variability based on random seeds, implementation details of low-level operations, etc. (see e.g. the BANKSY paper <https://doi.org/10.1038/s41588-024-01664-3> [doi.org]) The authors do present evidence of variability in metrics across training runs with different numbers of neighbors, but the impacts of this variation on spatial domains or downstream analysis is unclear. Does this variation result in small changes at the borders of niches? Or entirely different classification of GPs?

R3C8: Thank you for this comment. We acknowledge that “*quantitative definition of a niche*” might be interpreted in different ways. Based on your guidance and similar suggestions from reviewer 1 (see **R1C4**), we now have rephrased this to be clearer in the manuscript. Instead of “*quantitative definition of a niche*”, we now only refer to “*quantitative characterization of a niche based on signaling events*”. What we mean here is that GP embeddings represent the activity of spatial gene programs, and hence, differential testing on the niche-level reveals quantitative differences between signaling events driving those niches. Specifically, we have made various modifications to make this clearer:

- We have revised the **Abstract** section and in **lines 34-38** now explicitly state: “*NicheCompass learns interpretable cell embeddings of local signaling events that allow to simultaneously identify niches and elucidate the underlying cellular processes constituting each niche. Unlike existing methods, it uniquely enables quantitative characterization of niches based on these cell embeddings which encode cellular activity of diverse communication pathways, represented as spatial gene programs*”.
- We now consistently talk about “*cell embeddings of signaling events*” across the manuscript instead of sometimes referring to “*latent space*”, and other times to “*GP space*” or “*embedding*”.
- We have revised the **Introduction** and **Discussion** sections to make clear that the quantitative characterization of niches refers to the inferred GP activities. For instance, in **lines 72-76**, we now state: “*Here, we present NicheCompass (Niche Identification based on Cellular Graph Embeddings of Communication Programs Aligned across Spatial Samples), a graph deep learning approach to simultaneously identify and quantitatively characterize niches based on learned cell embeddings of signaling events, encoding activities of spatial gene programs (GPs)*”; in **lines 718-720**, we now state: “*We introduced NicheCompass, a graph deep learning approach designed based on the principles of cellular communication to identify and quantitatively characterize tissue niches through spatial GP activity.*”

Moreover, as outlined in **R3C2**, we have quantitatively evaluated the impact of the variation in the number of neighbors on spatial domain/niche identification and GP inference using our simulated data (**Supplementary Fig. 15**) and the seqFISH mouse organogenesis data (**Supplementary Fig. 8**). In the case of simulated data, we observed better performance on niche identification when using fewer neighbors while GP inference was better with bigger neighborhood sizes. In the case of the seqFISH mouse organogenesis data, we could not observe major differences in the identified niches and inferred GP activities with respect to the niches that we analyzed in **Fig. 2**.

Besides evaluating the impact of the variation in the number of neighbors, we also tested a scenario with a radius-based neighborhood graph (the radius was set to obtain an average number of neighbors of ~ 9). In this case we also did not observe major differences (**Supplementary Fig. 9**).

Extract Supplementary Fig. 9 | Niche and GP inference neighborhood graph robustness. **a**, Embryo 2 niches and GP activities inferred by NicheCompass with a longer-range kNN neighborhood graph ($k=12$). Displayed are characterizing GPs from the main analysis in **Fig. 2**. **b**, Same as **a** but with a shorter-range kNN neighborhood graph ($k=4$). **c**, Embryo 2 niches and GP activities inferred by NicheCompass with a radius-based neighborhood graph (average number of neighbors ~ 9). Missing GPs were filtered by GP pruning in the respective model. Overall, there is good robustness of inferred niches and GP activities across neighborhood graphs; however, there are also minor differences, most pronounced in the Hindbrain niche (see full version in **Supplementary Fig. 9**).

We have also added an explicit analysis investigating the effects of different random seeds in biological context (**Supplementary Fig. 8a,b**). On the inferred niches and GPs presented in our analysis in **Fig. 2**, we observed good reproducibility. In addition, we trained multiple models with different random seeds in our ablation experiments presented in **R1C1**. Like other methods, such as BANKSY, we also find some variability based on random seeds (as expected for a deep learning model).

Extract Supplementary Fig. 8 | Niche and GP inference reproducibility and generalizability. a,b, Embryo 2 niches and GP activities inferred by NicheCompass with different random seeds. Displayed are characterizing GPs from the main analysis in **Fig. 2**. Missing GPs were filtered by GP pruning in the respective model. Overall, there is good robustness of inferred niches and GP activities across random seeds; however, there are also minor differences, most pronounced in the Hindbrain niche (see full version in **Supplementary Fig. 8**).

Are there other parameters or data characteristics that have predictable impact on the results?

R3C9: Thank you for this question. We have also evaluated the impact of other major hyperparameters and model design choices in our ablation studies on simulated data presented in **R1C1** and **R3C4** (**Supplementary Fig. 15**).

Inclusion of this kind of information greatly increases the real-world usability of computational tools and allows researchers to establish realistic expectations of NicheCompass for application to their work.

Thank you for pointing this out. We agree with this statement and are grateful for your feedback that we believe has drastically improved the real-world usability of our work. We have added **Supplementary Note 5** where we highlight the impact of different design choices and provide user guidance.

Technical questions:

- I'm curious about some of the outcomes of the GP pruning and regularization. Can these procedures remove genes from the GPs that are present in the data? How many genes are present in the set of GPs after the 'warm up'? and after the completed training?

R3C10: Thank you for this question. On the one hand, based on the predefined set of prior GPs, the GP pruning removes GPs that are not relevant for the dataset under consideration (via the active GP threshold). On the other hand, the gene regularization removes irrelevant genes from specific GPs (there are two separate regularization hyperparameters for genes in prior GPs and genes in *de novo* GPs, respectively). These mechanisms interoperate during model training (see **Methods** section under the titles **Gene program pruning** and **Gene program regularization**).

To address your question, we have analyzed the number of active GPs and active source and target genes (1) at the start of model training (in columns labeled "Total"), (2) after the warm up epochs (in columns labeled "(W)") and (3) after completed model training (in columns labeled "(F)") in our new ablation study on simulated data. The results are reported in **Supplementary Fig. 15h**. On this dataset, before model training there were 440 GPs with genes present in the data. Across all these 440 GPs, there were ~120k source and target gene members. Depending on the specific hyperparameter configuration, we observed that after warm-up between 51.36% and 98.64% of GPs, between 1.26% and 89.31% of source genes, and between 24.12% and 93.64% of target genes were still active. After completed model training, we observed that between 47.27% and 73.18% of GPs, between 0.13% and 15.33% of source genes, and between 23.22% and 35.57% of target genes were still active.

h

Active GP Thresh	Prior GP Reg	De novo GP Reg	Total GPs	Active GPs (W)	Active GPs (F)
0.01	0.0	3.0	440	434	322
0.01	0.0	30.0	440	319	302
0.01	0.0	300.0	440	301	299
0.01	3.0	3.0	440	416	303
0.01	30.0	30.0	440	303	276
0.01	300.0	300.0	440	292	275
0.1	0.0	3.0	440	292	260
0.1	0.0	30.0	440	250	246
0.1	0.0	300.0	440	248	245
0.1	3.0	3.0	440	288	248
0.1	30.0	30.0	440	226	208
0.1	300.0	300.0	440	227	225

Total Sources	Active Sources (W)	Active Sources (F)	Total Targets	Active Targets (W)	Active Targets (F)
119547	106762	18107	119687	112072	42372
119547	17076	4509	119687	41839	32251
119547	3039	749	119687	30444	28739
119547	105950	18322	119687	111288	42568
119547	20304	4092	119687	44485	32101
119547	3507	336	119687	30951	28698
119547	35828	13332	119687	55549	37809
119547	5522	3710	119687	32155	30646
119547	1622	509	119687	28864	28071
119547	42053	13209	119687	60469	37534
119547	6095	3467	119687	32181	28490
119547	1510	159	119687	28865	27797

Extract Supplementary Fig. 15 | NicheCompass ablation. h, Number of total and active GPs, GP source, and GP target genes for different GP pruning and gene regularization hyperparameters. (W): Warm-up, (F): Final.

- NicheCompass framework is the same for spatial data at cellular resolution and also at lower resolution. How does this variation in scale impact the results? Testing NicheCompass on spatially-binned versions of high-resolution data would be one way to help understand how spatial resolution impacts the niche designations.

R3C11: Thank you for this suggestion. To investigate the impact of the variation in scale, we followed your proposed approach and have created a spatially binned version of our newly added simulation data. Specifically, we have created bins of size 55 μ m (as in 10x Visium) and aggregated the expression of all cells falling into each bin. We then compared the niche identification and GP recovery metrics of NicheCompass models trained on this version of the data and the original (unbinned) version. The results are reported in **Supplementary Fig. 16** and can be compared with **Supplementary Fig. 14**. Overall, while NicheCompass still shows good niche identification performance (albeit lower compared to the unbinned dataset), GP recovery performance is lower. This is expected as NicheCompass' conceptual idea of "self" and "neighborhood" modeling makes single-cell resolution data the primary use case. We have added a corresponding note in the **Discussion** section in **lines 768-769**.

In addition, we have trained various NicheCompass models and models from the other five benchmarked methods on a new spot-level resolution StereoSeq mouse embryo dataset. Here, NicheCompass was among the two best performing methods together with CellCharter. This provides evidence that while the primary use case of NicheCompass is single-cell resolution data, it can still be very valuable for niche identification on spot-level resolution data (**Supplementary Fig. 16**).

Supplementary Fig. 16 | Additional ablation and benchmarking on spot-level resolution data. **a**, Ground truth niches of the single-cell resolution simulated data that has been binned into spots with 55 μ m diameter. **b**, Distribution of the number of cells per bin/spot. **c**, Niches identified by a NicheCompass model with a misclassification at the periphery of Niche 8. **d**, Niche identification and GP recovery metrics on the binned simulation data for different encoder layer and neighborhood graph KNN hyperparameter combinations. For each combination, eight models have been trained. Boxplot elements are defined as: center line, median; box limits, upper and lower quartiles; whiskers, 1.5x interquartile range. **e**, Ground truth niches of the spot-level resolution Stereo-seq mouse embryo dataset. **f**, Mean NicheCompass and ground truth prediction metrics across eight training runs for each method, ranking NicheCompass first and second, respectively. STACI did run out of memory on our 40GB GPU and DeepLinc did not converge; both were hence excluded. The overall and ground truth prediction scores are computed by aggregating the min-max-normalized individual metrics.

- NicheCompass produces biologically relevant parcellations of the mouse brain in the case of STARMAP+ and MERFISH data, but they seem to be at varying resolution. Can NicheCompass integrate data from these 2 experimental modalities? What drives the difference in resolution between these 2 datasets? Spatial integration and congruent niche definition across modalities would be a noteworthy accomplishment and extremely useful to the field.

R3C12: We agree that integration across experimental modalities is an important feature for the field and are grateful for this suggestion. We have trained a new NicheCompass model jointly on the STARmap PLUS and MERFISH datasets. NicheCompass was able to integrate data from these two experimental modalities and identified congruent anatomic niches across both datasets (**Supplementary Fig. 36**).

Supplementary Fig. 36 | NicheCompass integrates tissue samples across different spatial transcriptomics technologies. a, UMAP representation of the NicheCompass GP embedding space after integrating the MERFISH mouse brain and STARmap PLUS mouse CNS datasets, colored by dataset/sequencing technology. **b,** Composition of niches in terms of cells from each of the two technologies, showing that all niches except niche 9 were present in both datasets. Only niches with more than 100,000 cells are displayed. **c,** Two example tissues of the same brain region, one from the MERFISH

mouse brain dataset and the other one from the STARmap PLUS mouse CNS dataset, highlighting consistent anatomical niches. **d**, Zoom in on four specific niches that emphasize the consistency in niche identification across technologies.

G. References:

The literature is adequately cited, although the collection of spatial domain detection methods has no doubt grown since this manuscript was submitted.

R3C13: Thank you for acknowledging that the literature is adequately cited. We agree that the collection of methods has grown since our submission. While it is hard to include all methods and keep up with the pace of newly arriving methods, based on feedback from reviewer 1 (see **R1C8**), we have additionally added BANKSY to the methods included in our benchmarking efforts (**Fig. 3**).

As a result of including BANKSY, we have:

- updated **Fig. 3c-f** to integrate BANKSY in both the single sample and sample integration scenarios (including renormalization of the metrics of all methods).
- updated **Supplementary Fig. 12b** with the BANKSY tissue niche composition
- updated **Supplementary Fig. 13** (added tissue annotated with BANKSY in panel **a**, added BANKSY metrics in panel **b**, renormalized all methods in panel **b** with BANKSY included).

H. Clarity and Context:

The overall organization of the manuscript is clear, the writing is good and the figures clearly communicate the main points of the paper. The breadth of the data exploration effort and clarity of presentation set this manuscript apart from the large pool of similar projects.

R3C14: Thank you for the positive feedback. We are very grateful that you emphasized the breadth of our data exploration effort in which we invested a lot of time. We are also happy that you found the presentation clear and liked our overall organization, writing and figures. Based on suggestions from reviewer 1 (see **R1C1**), we have further improved the presentation by shortening certain segments and improving the quality of figures.

We are thankful to all your comments which we believe significantly improved the paper and its presentation.

References

1. Singhal, V. et al. BANKSY unifies cell typing and tissue domain segmentation for scalable spatial omics data analysis. *Nat. Genet.* (2024) doi:10.1038/s41588-024-01664-3.
2. Zhu, J., Shang, L. & Zhou, X. SRTsim: spatial pattern preserving simulations for spatially resolved transcriptomics. *Genome Biol.* 24, 39 (2023).
3. Yuan, Z. et al. Benchmarking spatial clustering methods with spatially resolved transcriptomics data. *Nat. Methods* (2024) doi:10.1038/s41592-024-02215-8.
4. Lohoff, T. et al. Integration of spatial and single-cell transcriptomic data elucidates mouse

- organogenesis. *Nat. Biotechnol.* 40, 74–85 (2022).
5. Varrone, M., Tavernari, D., Santamaria-Martínez, A., Walsh, L. A. & Ciriello, G. CellCharter reveals spatial cell niches associated with tissue remodeling and cell plasticity. *Nat. Genet.* 56, 74–84 (2024).
 6. Luecken, M. D. et al. Benchmarking atlas-level data integration in single-cell genomics. *Nat. Methods* 19, 41–50 (2022).
 7. Haviv, D. et al. The covariance environment defines cellular niches for spatial inference. *Nat. Biotechnol.* (2024) doi:10.1038/s41587-024-02193-4.
 8. Luecken, M. D. & Theis, F. J. Current best practices in single-cell RNA-seq analysis: a tutorial. *Mol. Syst. Biol.* 15, e8746 (2019).
 9. Kanemaru, K. et al. Spatially resolved multiomics of human cardiac niches. *Nature* 619, 801–810 (2023).
 10. Scadden, D. T. The stem-cell niche as an entity of action. *Nature* 441, 1075–1079 (2006).

NG-TR65030 Quantitative characterization of cell niches in spatial atlases

2nd Response to Reviewers

We would like to thank the reviewers and are delighted that all reviewers found the revision substantially improved and satisfied with how we addressed their comments. We are grateful to reviewers and editors for the second round of comments and suggestions, which have again improved the quality of our manuscript. In short, guided by the editor, we mainly addressed the following points, as then described in the answer letter below:

- **Additional evaluation and guidance on hyperparameter choices:**
 - We performed additional ablation studies on real ST data, evaluating our model on niche identification performance using ground truth labels provided by the authors of the original study.
 - We added a user guide to our package documentation based on the results of this ablation study and our previous ablation study on simulated data. We also included guidance on the model selection and when to use NicheCompass Light and potential trade-offs from the user's perspective.
- **Clarifications for the ablation study on simulated data**
 - We provided an interpretation for the obtained F1 scores on *de novo* GP genes.
 - We discussed the trade-offs between the NID and GPR metrics based on different hyperparameters.
 - We clarified how the baseline F1 score is obtained.
- **Clarifications and additional discussion on the concept of niches compared to cell types**
 - We provided a more elaborate conceptual discussion on the concept of "niches" compared to "cell types".
 - We emphasized the requirement for spatial collocation of cells in our definition of "niche".
- **Corrections, clarifications, extensions and improved descriptions of existing analyses**
 - We corrected the labeling of cell types in **Supplementary Fig. 25**.
 - We corrected a wrong label in **Fig. 3b**.
 - We provided additional clarifications on how we created the sunburn visualizations of *de novo* GPs in **Fig. 4g-h**.
 - We have relabeled the region V3 in **Fig. 3** to V3/CP.
 - We have corrected a typo in the gene name in the caption of **Fig. 3**.
 - We clarified how hierarchical clustering was performed.
 - We strengthened our conclusions on different (sub)niches in **Fig. 3** by analyzing gene expression of marker genes from the Allen brain atlas.

- We added an additional panel to **Supplementary Fig. 25** to illustrate more in detail which non-tumor niches are shared across donors, and which are donor-specific. We also improved the visualizations in **Fig. 4** to differentiate this more clearly.
- We improved the description of existing methods in the **Introduction** section.
- We reworked the text related to the analysis presented in **Fig. 3a-c** where we now describe and explain differences obtained by NicheCompass and other methods compared to the anatomy, rather than highlighting how NicheCompass is better.
- We added visualizations of additional MERFISH and STARmap plus slices in **Supplementary Fig. 37** to strengthen our conclusion of successful integration across technologies.

In the following, reviewers' comments appear in blue, our point-by-point answers appear in green. Text and figures we have added or altered in the manuscript are highlighted in green in the manuscript.

Response to Reviewer 1

Reviewer #1 (Remarks to the Author)

I would like to thank the authors for their effort to address my comments and revise the manuscript. The manuscript has indeed improved in several aspects. However, there are still some major issues that require further attention. These include confusing aspects in the simulation study, potential confounding effects of the identified niches in the model, the need for greater clarity in the method description, and additional analysis to support for the identified niches in the brain.

Thank you for this positive and constructive feedback and all the valuable comments from the first round of review. We are happy that you perceived that the manuscript has improved in several aspects. We address the remaining points below.

My additional comments are listed below:

1. R1C1: While most scores appear to be unimodal or stable, it is difficult to discern a clear trend in the F1 for De-Novo GPs. How should this be interpreted?

R1C19: We appreciate this observation. Our interpretation of the obtained results is that the F1 score for *de novo* GP genes highly relies on appropriate regularization of *de novo* GP gene weights in the decoder, which is visible in **Supplementary Fig. 15b**: In this ablation experiment the differences are almost an order of magnitude bigger than in all other ablation experiments of different hyperparameters. With weak regularization, the F1 scores can clearly surpass a score of 0.1 and even 0.2 (for target genes), which is not achieved in any other ablation experiment. The reason for the low scores in all other ablation experiments is that we used the initial default regularization, which was no regularization (again see low scores for “no regularization” in **Supplementary Fig. 15b**). The variation is proportionally high as most other hyperparameters seem not to be as relevant as the *de novo* GP gene weight regularization hyperparameter. In our opinion, this also makes intuitive sense as the model can in general pick any gene not included in prior knowledge as a potential gene in a *de novo* GP (whereas the prior GPs are already constrained and therefore rely less on regularization). Without *de novo* GP gene weight regularization, it seems to be difficult for the model to discard genes that have not been upregulated in the simulation, whereas the regularization helps the model to achieve that. Because of this ablation experiment, we have changed the default regularization hyperparameter to “low regularization” as opposed to “no regularization” as discussed previously and want to thank you again for this important suggestion in the first round of review.

We also want to note that the two other hyperparameters that have relevant effects in this case are the number of *de novo* GPs and the neighborhood size. This can be seen in **Supplementary Fig. 15g**, where a high embedding size leads to significant drops in performance, and **Supplementary Fig. 15d**, where a high

neighborhood size equally leads to significant performance drops. Both cases have, in our opinion, intuitive explanations. First, if the number of *de novo* GPs is higher than the number of ground truth interactions, the F1 score decreases as more false positives, e.g. cell-type-specific genes, are picked up (**Methods** under title **Data Simulation; Supplementary Fig. 15g**). Additionally, the model can split effects of interacting genes across multiple GPs, making the signals picked up by individual GPs weaker. Second, with an increasing neighborhood size, one *de novo* GP can capture mixed signals as multiple diverse interactions and multiple different cell types can be present in the “receptive field” of a cell. This can blur the effect of individual important interactions picked up by a *de novo* GP (again, the *de novo* GPs have more freedom than the prior GPs and can therefore easily pick up genes from diverse biological processes).

We have added a shorter form of this interpretation in **Supplementary Note 5 (lines 472-478)**.

2. R1C2: The authors highlight the importance of the balance between the gene expression reconstruction loss and the edge reconstruction loss. They set the default hyperparameters in the NicheCompass package based on ablation studies conducted on simulated data and advise users against altering these settings. However, there is a notable gap between simulated data and real ST data, especially regarding specific spatial patterns and expression sensitivity which are significantly different. Therefore, this overall approach doesn't make sense. The authors need to explore metrics for determining the weight based solely on real ST data. A similar justification is also needed for other hyperparameters.

R1C20: Thank you for this valuable suggestion. To address it, we have performed additional ablation studies on real ST data, specifically a slice from the STARmap PLUS mouse brain dataset introduced in the analysis in **Supplementary Fig. 36 & 37 (Methods)**. This data was well suited for our experiments as the original authors provided niche/region labels. We therefore used these niche labels to compute the NMI and ARI across different hyperparameter configurations using scib-metrics¹. The specific procedure is described in the **Methods** section under the title **Ablation on real data**.

We described our results in detail in **Supplementary Note 5 (lines 497-510)** and the newly added **Supplementary Fig. 16**. For most hyperparameters we found good agreement between optimal configurations in the simulated and real ST data, showing the robustness of choices across specific spatial patterns and expression sensitivity. In summary, like the experiments on simulated data, the experiments on real ST data clearly show (1) the advantage of finding a balance between gene expression and edge reconstruction, (2) improvements when using GP gene weight regularization, (3) better niche identification with lower neighborhood sizes, and (4) the possibility for performance boosts while reducing embedding size with GP pruning. In contrast, with real ST data, we observed better niche identification with a GATv2Conv layer encoder instead of a GCNConv layer encoder; however, this is in line with our previous guidance as we had drawn a similar conclusion from our benchmarking efforts where NicheCompass (GATv2Conv layer encoder) outperformed NicheCompass Light (GCNConv layer encoder) on more complex datasets. This is reasonable as more expressivity in the encoder can help to model more complex spatial structures than the one present in our simulation study. Additionally, there was a clear performance boost when using prior GPs compared to a scenario without prior GPs.

We have also provided all our insights from both ablation studies in a User Guide in our package documentation.

Supplementary Fig. 16 | NicheCompass ablation on real data. Niche identification (NID) metrics on the real data under different model design choices and hyperparameters. Metrics measure the correspondence between predicted and ground truth niche labels provided by the authors in the original study. Boxplot elements are defined as: center line, median; box limits, upper and lower quartiles; whiskers, 1.5x interquartile range. Ablations were performed on **a**, the weighting of the reconstruction loss hyperparameters indicating that both edge and gene expression reconstruction are essential components, **b**, the weighting of the regularization loss hyperparameters, indicating that regularization improves NID. **c**, the encoder architecture, indicating that NicheCompass performs better on NID than NicheCompass Light, **d**, the neighborhood size indicating better NID for lower k. **e**, the number of de novo GPs, indicating that inclusion of de novo GPs slightly improves NID, **f**, GP pruning via the active GP threshold, indicating that GP pruning can slightly boost NID while reducing the model embedding size, and **g**, the use of prior GPs, showing better NID when prior GPs are used.

3. R1C3: It appears there are performance trade-offs between niche identification (NID) and gene program recovery (GPR) when selecting different GNN layers and varying the number of neighbors. Given the authors' definition of niches (communities of spatially localized cells with coordinated functions), synergy, rather than trade-off, may be expected. This confusion should be addressed carefully.

R1C21: We appreciate that you raised this point and agree that our definition of niches suggests a synergy between the general goals of NID and GPR. This is particularly true for our simulated data where we know that there are existent ground truth GPs underlying each niche. However, in our opinion, one cannot

deduct from this that each hyperparameter of the model needs to have positively correlated effects on both metrics. In terms of neighborhood size, we find it intuitive that a lower number of neighbors leads to better niche identification as especially at the niche boundaries the model has less impact from cells that come from different niches, which through the edge reconstruction loss are incentivized to have more similar embeddings. We believe that the performance increase in the F1 score of prior GPs with bigger neighborhood sizes is due to the design of the simulation dataset as we have replicated the same cell interactions across the niche with rather high levels of noise and sparsity. The model therefore benefits from a higher neighborhood size which helps to strengthen the signal of prior GPs. In contrast, this is not the case for *de novo* GPs which do not have similar constraints as prior GPs and instead are free to mix different interaction signals, which seems to lead to lower F1 scores for *de novo* GPs as the neighborhood size increases. We have added a corresponding interpretation in **Supplementary Note 5 (lines 482-486)**.

Overall, based on our results, we guide users to pick neighborhood sizes based on expected interaction ranges but not exceeding neighborhood sizes of 12 neighbors, which empirically did not seem to work well on real data in both our ablation and benchmarking experiments.

We also want to emphasize again the new experiment on real data which clearly shows advantages in niche identification performance when using prior GPs, which supports this synergy of goals mentioned by you (**Supplementary Fig. 16g**).

Extract Supplementary Fig. 16 | NicheCompass ablation on real data. Niche identification (NID) metrics on the real data under different model design choices and hyperparameters. Metrics measure the correspondence between predicted and ground truth niche labels provided by the authors in the original study. Boxplot elements are defined as: center line, median; box limits, upper and lower quartiles; whiskers, 1.5x interquartile range. Ablations were performed on **g**, the use of prior GPs, showing better NID when prior GPs are used (see full version in **Supplementary Fig. 16**).

4. I do not fully understand the statement, “The red dotted line indicates the baseline of mean F1 scores between the artificially incremented GPs and member genes and randomly selected candidate GPs and genes (with matching numbers).”

R1C22: We apologize that this description was not clear. The red dotted line marks a baseline that was computed as follows: (1) for each NicheCompass model training run, we selected random candidate GPs

and genes that matched the number of GPs and genes identified as upregulated by NicheCompass in that run (to account for differences in F1 scores just based on the number of selected GPs/genes; “candidate” means that only GPs and genes that could have been selected by NicheCompass as prior or *de novo* GP/genes, respectively, were eligible); (2) we then computed the F1 scores between these randomly selected GPs/genes and the artificially incremented GPs and genes in the simulation, i.e. the GPs and genes which we have upregulated based on the procedure described in the **Methods** section under **Data Simulation**; (3) we computed the mean scores across all model training runs.

To make this clearer for other readers, we have revised the figure caption, which now says:

“Supplementary Fig. 15 | NicheCompass ablation on simulated data. Niche identification (NID) and GP recovery (GPR) metrics on the simulated data under different model design choices and hyperparameters. The first row in each panel contains metrics that measure the correspondence between predicted and ground truth niche labels; the second row contains F1 scores between enriched, i.e. identified as enriched by NicheCompass (Methods), and artificially incremented GPs and member genes, i.e. GPs and genes that have been upregulated in the simulation, in ground truth niches. Boxplot elements are defined as: center line, median; box limits, upper and lower quartiles; whiskers, 1.5x interquartile range. The red dotted line indicates the baseline of F1 scores between the artificially incremented GPs and member genes and randomly selected candidate GPs and genes (Methods; only GPs and genes that could have been selected by NicheCompass as prior or de novo GPs/genes, respectively, qualified as “candidates”; to avoid differences in F1 scores due to the number of GPs and genes, the number of random GPs and genes was matched for each run with the number of GPs and member genes identified by NicheCompass and the average across all runs was reported).”

Moreover, the exact procedure is described in the **Methods** section under the title **Ablation on simulated data**, to which we now also refer in the figure caption.

We hope this clarifies what the red dotted line shows and thank you for suggesting providing clarification.

5. R1C3 & R1C4: On a related note, the claim that NicheCompass is designed to identify niches and not cell types seems to rest heavily on the weighting of gene expression and edge reconstruction. The model did not take steps to remove the effect of cell type. This suggests that the two concepts – niches and cell types – are closely related. Additional analysis is needed to develop a general principle on their relationship, which would strengthen the work. For example, when two cells of the same type should be in different niches, and when two cells of different types should be considered part of the same niche.

R1C23: Thank you for this assessment. We want to acknowledge that it is true that we did not take explicit steps to remove the effect of cell type. However, this is intentional because we think that cell type (or rather cell type specific gene expression variation because neither did we use cell types as a supervised label) is a crucial factor in the determination of a niche as we have defined it (“niches, defined as communities of spatially localized cells with coordinated functions”). Coordinated function is often achieved through a community of cells of the same cell type being spatially colocalized and working together to support tissue homeostasis², potentially in conjunction with cells of another cell type. In that regard, in our opinion, if two cells of the same cell type are spatially colocalized they should be part of the same niche as it would be hard to argue that they perform vastly different functions. However, while there

is a relationship (and clear correlation) between the two concepts, there are clear differences as per our definition. For example, in our view, a niche needs to be spatially colocalized and cannot be distributed across a tissue. A comparison between **Fig. 3a** and **Supplementary Fig. 12d**, for instance, clearly shows this difference. On the one hand, niches often have a majority cell type dominating their function (it would therefore not be good to remove the effect of cell type); on the other hand, many cell types can be found in different niches (e.g. the microglia). There are existing approaches that attempt to remove the effect of cell types to isolate spatial effects³; however, we consider this a different goal/problem and such algorithms have shown limited success in identifying niches (as we define it) based on spatial effects only. Nonetheless, we think that this approach of isolating cell type independent spatial effects is another interesting direction to pursue in future work and we mention this in the **Discussion** section, which we have now updated with more detail (**lines 769-771**).

In addition, while we appreciate the point raised, we respectfully disagree with the statement that the identification of niches versus cell types heavily relies on the weighting of the gene expression and edge reconstruction losses. This is because our gene expression reconstruction loss is a spatially aware loss that reconstructs not only the gene expression of a cell itself but also that of its neighborhood. In our experiments we can therefore see that even without an edge reconstruction loss, our model can identify niches and not cell types (see **Supplementary Fig. 15a,e**; a method could not achieve an NMI > 0.7 on niche identification if it would identify cell types; our method with only gene expression reconstruction performs comparably to DeepLinc which still quite reasonably identifies niches). Nonetheless, as per our ablation experiments recommended by you in the first round of review and the new ablation experiments on real data recommended in the second round of review, the edge reconstruction loss improved the identification of niches especially in more heterogeneous tissue regions where spatial gene expression reconstruction alone might not be sufficient to preserve spatial adjacencies.

6. R1C6: The visualization is helpful. It would also be important to provide more details in the Methods section regarding how the genes are characterized. Additionally, [minor] is “C5orf36” a typo for “C5orf46” (as shown in the figure)?

R1C24: We are pleased that you found our visualization useful. The genes for this plot are extracted from the GPs learned by NicheCompass and used for characterization into “Pathway” and “Gene Family” using the Python implementation of BioMart (<https://pypi.org/project/biomart/>). We then used this classification dictionary to build the sunburst plot linking the hierarchy of the genes into the gene family and pathway, and colored genes based on the NicheCompass gene weights. A notebook is provided in our reproducibility repo for users to reproduce this. In addition, to take advantage of the available Large Language Models, we have generated a standardized prompt for ChatGPT 4o to feed the genes and get the hierarchical classification of the genes: *“Please classify the genes by their broad categories and subcategories based on their biological functions and roles. Use two layers of classification: categories and subcategories, for example, Immune system genes will be a category and Cytokines will be a subcategory. Don't use Miscellaneous or Other classification. Use up to nine categories for the first layers of classification. Output the results in the data frame: Category - subcategory - gene. Here is the list of genes to classify”*.

We have added a short form of this description in the **Methods** section under **Data Visualization (lines 1187-1192)**.

We also apologize for overlooking the typo. The correct gene name is indeed *C5orf46*, as shown in the figure. The text has been corrected accordingly in **line 523**.

7. R1C10: The authors need to clarify in the Methods whether the underlying hierarchical clustering for the dendrogram was performed in the embedding space.

R1C25: We apologize that we have not clearly described that the hierarchical clustering was performed in the embedding space. We have now added this to the Methods section in lines 1197-1198: *“Second, hierarchical clustering is computed on the embedding space of a model with niche labels as input, using the scanpy tl.dendrogram() functionality”*.

8. R1C9: The authors increased the clustering resolution parameter, leading to further separation of a *Fasciola cinerea*-like niche, distinguishing it from the V3-like and CA1-like niches. While the authors have provided cell type proportions for each identified niche, it would be more reliable and convincing to show the association between identified niches and brain regions by displaying the expression levels of known marker genes for brain regions, as readily available in databases such as the Allen Brain Atlas.

R1C26: Thank you for this valuable suggestion. We agree that this is a good way to provide additional support for the identified niches beyond anatomic location and cell type composition. We therefore have added plots of marker gene expression in **Supplementary Fig. 12f**. The expression of these marker genes clearly delineates the identified niches and thus strengthens our annotation.

f

Niche Marker Gene Expression

Extract Supplementary Fig. 12 | SlideSeqV2 mouse hippocampus benchmarking. f, Marker gene expression of three adjacent niches shows a clear delineation between the niches beyond anatomic location and cell type composition. Each row corresponds to a specific niche. The bar charts on the left display the proportions of the most abundant cell types in each niche (those making up more than 10% of the niche's cells). The color and length of each bar reflect the relative cell abundance. The heat maps illustrate the average expression levels of specific gene markers across different cell types within each niche, with color intensity indicating the mean gene expression. Shown are niche-specific marker genes that are differentially expressed in that niche compared to all other niches combined (see full version in **Supplementary Fig. 12**).

Response to Reviewer 2

Reviewer #2 (Remarks to the Author):

The authors have made significant efforts to address each comment from reviewers, resulting in substantial improvements to the manuscript, especially regarding the accuracy and consistency of the figures and the text describing them. This is challenging, given that there is large biological and technical variation of the analysis results across samples, disease types, technologies, and parameter settings in

different methods. The additional analyses and figure updates and new supplementary figures provide a comprehensive assessment of the method across various considerations, which will be beneficial for broader audience and applications.

Thank you so much for this positive feedback. We are happy that you think that we have made substantial improvements to the manuscript and are grateful for all the valuable suggestions received in the first round of review.

Several remaining, minor comments are:

- Cell type signatures for many niches shown in the Supplementary Fig. 24a are not as expected, e.g., all or most cell types in niches 6, 7, and 12 don't have their corresponding markers highly expressed, but display markers of other cell types. This raises the concern about how reliable and consistent this cell-type specific gene marker analysis is.

R2C14: Thank you for pointing this out. We realized that we accidentally integrated an erroneous plot with wrongly ordered rows in the manuscript and missed this in our own review. We are very sorry for this and have updated the figure with the correct ordering of rows corresponding to the matching cell type labels, which should now make sense (**Supplementary Fig. 25a**). We highly appreciate your feedback here which allowed us to correct this mistake.

Extract Supplementary Fig. 25 | Characterization of stromal niches. a, Each row represents a niche. The bar plots on the left represent cell proportions for the most abundant cell types in that niche (i.e. more than 10% of the cells in the niche). The color and length of the bar is proportional to the cell abundance in the niche. The heatmaps show mean expression of selected gene markers across cell types in each niche separately, with color representing mean gene expression. Shown are selected marker genes per cell type that are differential in that cell type compared to the rest, considering all the niches together. Indicated at the top are the cell types represented by each set of markers (see full version in **Supplementary Fig. 25**).

- Regarding batch effect, this reviewer agrees with the authors regarding the expected separation of tumour clusters across patients due to heterogeneity, however Figure 5c appears to show clear batch effect in multiple niches that are not just tumour niches. For examples, niches 2 (tumor) and 12 (macrophage enriched) seem to be exclusively in patient 6, while niches 11 (lymphoid aggregate) and 7 (neutrophil expansion) are for patient 5. Similarly for Figure 5e “Shared and donor-specific stromal niches”, the niches 9, 11, 6, 7 appear to be specific to donors, but not shared. The Figure 5 will benefit from highlighting more about the shared niches. Finding consistent niches across patients is a question that is attracting a lot of attention in the field.

R2C15: Thank you for this feedback. We agree that **Fig. 5e** was not clear enough and that shared niches were not sufficiently highlighted. We have addressed this via multiple improvements:

- We have added a new panel **b** to **Supplementary Fig. 25** in which we classify all stromal niches as donor-shared or donor-specific depending on whether they are present (i.e. more than 5% of the cells in the niche) in one or more donors. There, we also show the cell type composition in that niche per donor to ensure that when structures from two samples are assigned to the same niche, it is because the structure is shared and not because of over-integration. Although most niches are dominated by a sample, when the structure is present in an additional sample, we observe that in general we correctly map it to a shared niche.
- We also highlight this in **Fig. 5e** in niche 9, 11 and 6, which are dominant in a sample but also present in others. We have clarified this panel by adding in the legend which niches are shared and which are donor-specific. We have also added another case of shared niches (niche 6 in donor 9 and 12).
- We have improved **Fig. 5c** by shuffling the cells before plotting them so that they are not plotted ordered by sample, which hid mixed niches in the UMAP.
- We have updated the text to indicate in which donor niches are present (**lines 572-576**).

Extract Supplementary Fig. 25 | Characterization of stromal niches. b, Niche cell type composition for all the samples where the niche is present (i.e. more than 5% of the cells in the niche are from that sample). Top bar plots show the cell type composition and bottom bar plots show the proportion of the cells from each niche in each of the samples (see full version in **Supplementary Fig. 25**).

Extract Fig. 5 | NicheCompass enables spatial reference mapping to contextualize new donors with a spatial atlas, revealing emergent niches. e, Spatial visualization of tissue sections colored by niche and cell type, highlighting shared and donor-specific stromal structures identified across donors (see full version in **Fig. 5**).

Extract Fig. 5 | NicheCompass enables spatial reference mapping to contextualize new donors with a spatial atlas, revealing emergent niches. a-c, UMAP representations of the NicheCompass GP embedding space after integrating six lung samples from NSCLC patients, colored by identified and annotated niches using NicheCompass (a), pre-annotated cell types (b), and sample donor and replicate (c) (see full version in Fig. 5).

- NicheCompass light seems to perform not as good as other methods (Figure S16). The authors may add discussion about the usage case for this option to guide users on when they should apply NicheCompass light.

R2C16: Thank you for this feedback. We have added this observation and additional guidance on when we think NicheCompass Light should and should not be used in **Supplementary Note 5 (lines 506-508)**. In addition, we have now added a page called **User Guide** in our package documentation, where we also include comments and guidance on NicheCompass Light from **Supplementary Note 5**.

Response to Reviewer 3

Reviewer #3 (Remarks to the Author):

The authors have greatly improved this manuscript in response to peer review.

The detailed assessment of the model under different parameters and conditions and the testing on simulated data generally support the robustness of the method and validity of the NicheCompass design. The successful integration of two independent mouse brain spatial datasets suggests that NicheCompass will be useful in atlas creation (as the authors point out) but also as a more general-purpose tool for data integration.

Thank you for this positive assessment and all the important points raised in the previous round of review. We are especially grateful for the suggestion to integrate datasets from different technologies, which we agree demonstrates important capabilities for spatial atlas creation by the community.

I have one major issue on biological interpretation and presentation of the results:

Figure 3a-d and related text:

a. the "V3" (third ventricle) region of the brain is open space, not tissue. The choroid plexus is a distinct structure inside the third ventricle and likely the source of mRNA sampled in that region. Unless there are specific marker genes or other information to disambiguate, I suggest labeling this region V3/CP to clarify.

R3C15: Thank you, we highly appreciate your expertise in brain biology and have relabeled this niche/region as V3/CP (lines 327-328). We have also updated the labels in **Fig. 3** and **Supplementary Fig. 12** accordingly.

In this process, we also noticed that there was a small labeling mistake in **Fig. 3b** where cluster 2 was labeled as "Hippocampal Region" and cluster 3 as "Isocortex" when it should have been the other way round. We fixed this now by swapping the labels.

Extract Fig. 3 | Benchmarking NicheCompass across various scenarios. b, A dendrogram computed on average GP activities, uncovering a hierarchy of anatomically and molecularly similar niches (see full version in **Fig. 3**).

b. When comparing these results to the anatomy of the mouse brain, there is very little justification for taking NicheCompass over the other results here.

They are all imperfect:

- none of these methods produce a convincing L6b (which is much thinner than the laminar regions shown)
- BANKSY doesn't differentiate thalamic nuclei
- GRAPH ST fails to report laminar structure in cortex
- Cell Charter and DeepLinc smear the pyramidal layer in CA1-3 well beyond their cytoarchitectural boundaries.

Furthermore, the 'artifacts' present in BANKSY, GraphST and CellCharter almost certainly correspond to inhibitory neuron populations that are detected as distinct spatial domains (compare to Gad2 expression in Allen ISH atlas), and the 'artifact' in DeepLinc may correspond to glial cells along the boundary of the thalamus (compare to Aqp4 expression in Allen ISH atlas). In what way is NicheCompass better for not detecting these spatially-localized variations in gene expression?

Similarly, the hierarchy matching of niches to anatomy is different among the methods, but I don't think it's accurate to claim that NicheCompass' is better matched than others.

Despite these shortcomings, the analysis presented in the figure can be a valuable contribution if the writing is improved. The authors should explain how NicheCompass' design leads to the results in 3a-b, describing points of agreement and disagreement with the anatomy as well as with other methods. NicheCompass is one of many tools in this space and the comparisons here are useful in showing how NicheCompass performs in a spatially and transcriptomically complex dataset. This kind of transparent comparison is more valuable than stretching to make NicheCompass seem "better".

R3C16: We are grateful for these valuable and specific suggestions on how to improve our comparison with other methods related to the capture of mouse brain anatomy. We agree with this approach and, in accordance, have revised the text related to **Fig. 3**. Specifically, we removed all claims that NicheCompass is better matched than other methods and instead focused on describing differences between methods compared to the anatomy. We also highlighted how the design of the loss function in NicheCompass contributes to the identification of spatially contiguous niches/domains. We highly appreciate the specific observations you have provided for different methods and have directly integrated them into the manuscript. We have also relabeled all the "Artifact" clusters that captured spatially contiguous domains as either "*Thalamus Boundary*" or "*Lateral Ventricle*" (**Fig. 4c**). In the case of "Artifact 1" in BANKSY and GraphST and "Artifact 2" in CellCharter, we have kept the label as the corresponding cluster is not spatially contiguous (i.e. it consists of individual cells that are not proximal in physical space) and therefore does not fulfill our definition of a niche, which now emphasizes the requirement for spatial colocation: "*niches, defined as communities of spatially colocalized cells with coordinated functions*" (**line 53**). However, in the text we make it clear that the identification of these spatially non-contiguous "Artifact" clusters is not a failure of these methods but is also biologically interesting as they mostly capture Inhibitory neuron populations (although cell type composition is also mixed; see **Supplementary Fig. 12b**). We have also reformulated the comparison of the identified niche hierarchies between methods in a neutral way.

The complete text now reads:

“Among methods that trained successfully, NicheCompass identified only spatially contiguous niches in line with its design of joint spatial gene expression and edge reconstruction losses (**Methods**) and our concept of niches as communities of spatially colocalized cells. Contrarily, other methods including BANKSY, GraphST and CellCharter also identified clusters that were not spatially contiguous. We labeled these clusters as “Artifacts” in **Fig. 4** as they do not fulfill our definition of niche; however, they are also biologically interesting as they mostly capture inhibitory neuron populations (Supplementary Fig. 12b; compare also to *Gad2* expression in Allen ISH atlas). Besides, all methods struggled to produce a convincing L6b layer, which is much thinner than the identified laminar regions. BANKSY could not differentiate the thalamic nuclei, GraphST failed to report laminar structure in the cortex, and CellCharter and DeepLinc smeared the pyramidal layer in CA1-3 well beyond their cytoarchitectural boundaries. Hierarchical clustering of the identified niches further showed slight differences between methods in comparison to the anatomical hierarchy although all methods could recover reasonable structures (**Methods; Fig. 3c & Supplementary Fig. 12b**)” (lines 335-346).

Regarding modifications to figures, we have updated the labels in **Fig. 3** and **Supplementary Fig. 12** as well as the figure captions:

- The cluster previously labeled “Artifact 1” identified by CellCharter and DeepLinc is now labeled “Thalamus Boundary”.
- The cluster previously labeled “Artifact 2” identified by CellCharter is now labeled “Lateral Ventricle” (it consists of ependymal cells adjacent to the corpus callosum).
- We have added a color legend for all clusters not identified by NicheCompass.

Extract Fig. 3 | Benchmarking NicheCompass across various scenarios. **c**, The mouse hippocampus tissue colored by niches identified through clustering of the embedding spaces of four similar methods. The cluster colors match with **a**. Below it, the dendrograms, computed on average embeddings of the

respective method. Clusters/niches not present in NicheCompass have an additional color legend (see full version in **Fig. 3**).

Minor Notes:

1. L69-70. Existing methods also incorporate local spatial organization of gene expression

R3C17: Thank you for this suggestion which we agree is an important component of existing methods. We have updated **lines 69-70** accordingly: *“Existing approaches define and identify niches by grouping cells based on their tissue histology or by incorporating local spatial organization of gene expression”*.

2. The supplemental note 5 on model components and hyperparameter tuning will be helpful for users and some version of it should be included in the package documentation

R3C18: We appreciate this idea and have created a new page called **User Guide** in our package documentation. This page includes the information about model components and hyperparameters from our ablation studies summarized in **Supplementary Note 5**. Based on additional user feedback in the future, we plan to extend this page further. We have also updated the **Index** page to include this new section.

3. In supplemental figure 36 the mouse brain should be displayed with the dorsal side up. Showing more side-by-side comparisons of niches across modalities would also strengthen the conclusions about successful integration.

R3C19: Thank you for these suggestions. We have adjusted **Supplementary Fig. 36** (now **Supplementary Fig. 37**) to show the mouse brains with the dorsal side up. In addition, we have added 2 more diverse pairs of matching slices (sagittal) between the STARmap PLUS and MERFISH datasets for side-by-side comparison. Again, we find vast agreement between the identified niches across technologies (**Supplementary Fig. 37e**). This further strengthens our conclusions.

Supplementary Fig. 37 | NicheCompass integrates tissue samples across different spatial transcriptomics technologies. **a**, UMAP representation of the NicheCompass GP embedding space after integrating the MERFISH mouse brain and STARmap PLUS mouse CNS datasets, colored by dataset/sequencing technology. **b**, Composition of niches in terms of cells from each of the two technologies, showing that all niches except niche 9 were present in both datasets. Only niches with more than 100,000 cells are displayed. **c**, Two example tissue slices of the same brain region, one from the MERFISH mouse brain dataset and the other one from the STARmap PLUS mouse CNS dataset, highlighting consistent anatomical niches. **d**, Zoom in on four specific niches that emphasize the consistency in niche

identification across technologies. **e**, Two additional pairs of tissue slices showing consistent NicheCompass niches across the two technologies.

References

1. Luecken, M. D. et al. Benchmarking atlas-level data integration in single-cell genomics. *Nat. Methods* 19, 41–50 (2022).
2. Meizlish, M. L., Franklin, R. A., Zhou, X. & Medzhitov, R. Tissue homeostasis and inflammation. *Annu. Rev. Immunol.* 39, 557–581 (2021).
3. Dong, M., Kluger, H., Fan, R. & Kluger, Y. SIMVI reveals intrinsic and spatial-induced states in spatial omics data. *bioRxiv* 2023.08.28.554970 (2023) doi:10.1101/2023.08.28.554970.

NG-TR65030 Quantitative characterization of cell niches in spatial atlases

3rd Response to Reviewers

We would like to thank the reviewers for all their suggestions that have greatly improved this manuscript. We are very happy that all reviewers found the 2nd revision addressed remaining concerns. In this revision, we focused on additional efforts to streamline the text and improve accessibility. Specifically, we addressed the following points, as then described in the answer letter below:

- **Text streamlining:**
 - We significantly shortened the main article, mostly by elimination of less important details and more succinct wording. We also created 5 additional Supplementary Notes by moving text from the main article to the Supplementary Material:
 - Supplementary Note 1: Description of prior program categories
 - Supplementary Note 4: Communication potential scores and communication strengths
 - Supplementary Note 7: Extended benchmarking on simulated data
 - Supplementary Note 9: Spot resolution benchmarking
 - Supplementary Note 10: Extended NanoString CosMx human NSCLC benchmarking
 - We significantly shortened the methods section, mostly by elimination of less important details and more succinct wording. We also created a Supplementary Methods section.
 - We rewrote the Abstract and Introduction sections to be shorter and more accessible based on feedback from external researchers.
- **Accessibility of figures:**
 - We spent significant effort to format all figures by e.g. adjusting font sizes, adding scale bars, adding UMAP axes, and recreating all graphics as vector graphics. In some cases, we split up figures for better readability:
 - Supplementary Fig. 15 was split up into Supplementary Figs. 10-12
 - Supplementary Fig. 19 was split up into Supplementary Figs. 15-19
 - Supplementary Fig. 20 was split up into Supplementary Figs. 20-23

In the following, reviewers' comments appear in blue, our point-by-point answers appear in green. Text and figures we have added or altered in the manuscript are highlighted in green in the manuscript.

Response to Reviewer 1

Reviewer #1 (Remarks to the Author)

The authors have satisfactorily addressed most of the concerns raised in the previous round of review regarding evaluation metrics, hyperparameter selection strategies, the definition of niches, and other components of the paper. The inclusion of ablation studies on real spatial data strengthens the manuscript.

Thank you again for your constructive suggestions that have significantly improved this manuscript. We are happy that you find that most concerns have been satisfactorily addressed. We agree that the inclusion of ablation studies on real spatial data strengthens the manuscript and are especially grateful for this idea.

While the responses are thorough, the manuscript may benefit from additional efforts to streamline the text and improve accessibility as it seems not only too long but some of the more technical descriptions could move to the supplement. But I will leave this to the editors and authors to decide.

Thank you for this assessment. We have significantly shortened both the main article (previously 9042 words, now 3929 words) as well as the Methods section (previously 8652 words, now 4034 words). As mentioned in the summary above, we have created 5 new Supplementary Notes as well as a Supplementary Methods section and have moved more technical descriptions to the Supplementary Information. Based on feedback from the editors, we have also reworked the Abstract and Introduction sections to make them more accessible to external researchers. Lastly, we have also moved technical descriptions from figure captions to the Methods or Supplementary Methods sections which we believe further improved accessibility.

Overall, I believe this work represents a valuable contribution to the literature on spatial transcriptomics analysis methods.

Thank you very much for this positive conclusion and all your suggestions that positively impacted the contribution of this work.

Response to Reviewer 3

Reviewer #3 (Remarks to the Author):

The authors have responded to reviewers' concerns and returned an improved manuscript that balances technical detail, applications of the method, biological exploration and helpful information for eventual users of NicheCompass. I have no further comments on the manuscript and support its publication.

Thank you very much for all your constructive suggestions that have greatly improved this manuscript.
We are happy that you support publication.